Corrected: Publisher correction

# Differences in age-specific mortality between wild-caught and captive-born Asian elephants

Mirkka Lahdenperä [1], Khyne U. Mar[1], Alexandre Courtiol [2] & Virpi Lummaa [1]

Wild-capture of numerous species is common for diverse purposes, including medical experiments, conservation, veterinary interventions and research, but little objective data exists on its consequences. We use exceptional demographic records on Asian elephants from timber camps in Myanmar to investigate the long-term consequences of wild-capture during 1951–2000 on their mortality ($N = 5150$). We show that captured elephants have increased mortality compared to captive-born elephants, regardless of their capture method. These detrimental effects of capture are similar for both sexes but differ substantially according to age. Elephants captured and tamed at older ages show a higher increase in mortality after capture than elephants captured and tamed young. Moreover, the increased mortality risk following capture and taming is still perceived several years after capture. Our results are timely given the continued capture of elephants and other wild animals to supplement captive populations despite the alarming declines of wild populations globally.

[1] Department of Biology, University of Turku, 20014 Turku, Finland. [2] Department of Evolutionary Genetics, Leibniz Institute for Zoo and Wildlife Research, Alfred-Kowalke-Strasse, Berlin 10315, Germany. These authors contributed equally: Mirkka Lahdenperä, Khyne U. Mar, Alexandre Courtiol, Virpi Lummaa. Correspondence and requests for materials should be addressed to M.L. (email: Mirkka.Lahdenpera@utu.fi)

Animals are captured from the wild in vast numbers each year for various purposes, including well-accepted causes such as conservation, population management and research[1,2]. The resulting captive populations offer attractive research opportunities for multidisciplinary studies with easier observation and experimentation, and they are often used for general inferences concerning the entire species (for example see ref. [3–5,6,7]). Despite the acknowledged limitations of captive environments, such populations are also used as the reference group for optimum predation-free life-history or maximum longevity in evolutionary ecological studies (for example see ref. [8,9]). While some species are indeed healthier, longer-lived and more fecund in captivity than their free-living conspecifics, others perform less well[10,11]. Although studies on the effects of captivity on life-history are accumulating[2,11–13], less is known about the long-term differences between captive-born and wild-caught animals, despite a substantial proportion of captive individuals being obtained from the wild rather than born in captivity. Most of the existing research on the outcomes of wild-capture is concerned with the short-term elevated mortality risk that such operations may pose[14–18]. By contrast, evidence of the long-term consequences of capture and the resulting subsequent differences in experiences on mortality is less common[14–16], and the effects on different age groups and sexes remain poorly understood. This is an important shortcoming, because any long-term differences between captive-born and wild-captured animals are currently not considered in research and conservation programs.

The capture of wild animals may have long-term consequences on life-history for three main reasons. First, capture can alter behavior, physiology, and immunity[19–21] through chronic stress or sustained injuries, leading to reductions in survival rates over the subsequent months or years[16,22,23]. Second, even brief disruptions to early developmental conditions can cause considerable reductions in later-life health, reproduction and survival[24] that can persist for decades in long-lived species[25]. Third, interactions with humans, taming/breaking, changes in social systems and dynamics, interspecies competition, and social isolation within the captive environment can have further effects on post-capture lifespan[26]. For example, some wild-captured animals may be subject to differences in management compared to captive-born animals for continued periods following capture, for example, due to differences in behavior[27–30]. Finally, there may be pre-existing differences between wild-captured and captive-born animals due to inherent differences (e.g. captured individuals can experience a more natural environment before being captured), selective capture of certain type of animals[26,31] and selective survival during and after capture process[32].

One of the most striking examples of frequent wild-capture still happening today[23,33] in a species that often performs poorly in captivity is elephants, yet the long-term effects of capturing elephants are largely unknown. First, compared to wild or semi-captive populations, both African (*Loxodonta africana*) and Asian elephants (*Elephas maximus*) suffer considerably higher mortality rates in zoos[34]. However, such comparisons do not reveal the effects of origin per se, but instead illustrate management differences because the diet, social environment, exercise possibilities and disease patterns in zoo populations are vastly different to the wild[12,13]. For example, although not studied in detail, stress levels of captive elephants are reported to increase when interacting with humans[35]. Moreover, a lack of multigenerational family groups in most zoos means that the early maternal environment of those born in captivity is typically different from wild-captured animals. This prevents the differentiation of early parental effects[36] from capture effects among wild-born and captive-born zoo residents. Different zoos also present heterogeneous living conditions, breeding possibilities and climate, and

even small variation in factors such as within-year fluctuations in temperature can double the mortality risk of Asian elephants in range countries[37].

Second, in Asian captive populations wild-born elephants suffer from increased mortality compared to captive-born individuals (but vice versa in zoos, see discussion for more details)[31,34,38] but no studies currently compare the age-specific survival of males and females after capture, the effects of different capture methods, or investigate how long the adverse effects of capture persist for individuals of different ages, sex and capture method. Thus, we lack detailed comparisons of mortality patterns between wild-captured and captive-born individuals that otherwise live in a similar environment with shared food and disease source, similar social interactions and breeding patterns. Unfortunately, such data rarely exist for any species, preventing the comparison of the long-term survival of wild-caught and captive-born animals, despite the urgent need of animal welfare specialists, veterinarians and ecologists to identify such effects for the success of the individuals and consequently populations.

Here we take advantage of exceptional studbook data recorded on timber elephants in Myanmar (Burma) by the local government for a century, to demonstrate the adverse age-specific mortality effects of capture from the wild and subsequent taming in a long-lived mammal. Live-capture of wild Asian elephants to replenish captive populations has been practiced for more than 3000 years[30,39,40]. Consequently, a third (~15,000) of the remaining, endangered global population of Asian elephants now lives in captivity in range countries[41] (including captive facilities, temples, private owners and working camps). The largest such captive population exists in Myanmar, with ~5000 captive elephants mainly employed in the timber logging industry[41]. The capture of wild elephants to supplement the timber elephant population has been performed using three different methods; (i) stockade for whole group, (ii) immobilization by sedation, and (iii) milarshikar (lasso) of chosen individuals (see results and methods for details). Capture is controlled by the government: offtake numbers have varied between 22 and 283 individuals captured per year from 1970–1993, with >2000 individuals captured in total by the government over the 22-year period[42], however these values may be underreported[43]. Few restrictions on capture existed before 1968, but since then a minimum shoulder height of 1.37 m (~5 years age) has been enforced. The immediate mortality rate during the capturing process and taming is unknown to us and not included in our study. However, estimates are high, varying between 5 and 30% or potentially even higher in the same time period depending on the capture method[40,42]. Although wild-capture in Myanmar was banned in the 90s[44], smaller-scale capture primarily focused on individuals involved in human-elephant conflict continues.

Here we produce a detailed analysis of the age-specific effects of wild-capture in Myanmar and investigate whether (i) capture increases the mortality risk of elephants compared to captive born individuals, (ii) the effect of capture on mortality depends on the age at capture, (iii) the effect of capture on mortality depends on the time since capture, (iv) there is variation in mortality based on the capture method used, and (v) there is a sex difference in the effect of capture on mortality. Our dataset enables us to answer such questions. First, it includes longitudinal repeated measures of mortality risk for 2930 females over 50,054 elephant-year observations and for 2220 males over 32,972 elephant-year observations. Of these elephants 2072 were wild-captured during 1951–2000 and 3078 were captive-born during 1925–1999. Consequently, the dataset offers a robust sample size to test the effects of capture over entire life-spans. Second, wild-caught and captive-born elephants live, forage and work alongside one another, and the same governmental regulations apply for both

types of elephants concerning data recording, workload and rest periods. The elephants are not provisioned, but instead forage unsupervised in forests at night, and the same basic veterinary care is available to all individuals. Their social environment includes genetically related and unrelated animals of both sexes and all ages that they can choose to interact with during nights and free-time unsupervised in the forest, including allomothers and grandmothers known to increase calf survival in the population[45]. Both wild-caught and captive-born elephants undergo taming based on similar principles and methods before entering workforce[31,46]. Calves born in captivity spend their first 4–5 years in relative freedom unused to human handling and commands, and are then separated from their mother and trained to work. Wild-caught elephants follow similar procedure after capture, but are likely to be exposed to harsher treatment depending on their sex, age and personality. Thus part of the effects between captive-born and wild-caught elephants can be due to differences during taming period. Finally, detailed information exists on the age and time of capture for wild-captured or birth date for captive-born elephants as well as living region for both, enabling us to control for confounding factors such as geographic and temporal variation in living conditions and mortality. Our results demonstrate a considerable cost of capturing elephants from the wild on both their short and long-term mortality, regardless of the capture method used. The practice of capturing elephants for sustaining captive populations is not only detrimental because it reduces wild populations of this endangered species; it also fails to provide a viable solution for sustaining captive populations. The long-term differences between captive-born and wild-captured elephants shown by our study are currently rarely considered in research and conservation programs.

## Results

**Baseline mortality strongly depends on age and sex.** We first characterized the general mortality pattern of the studied elephant population, which we used in the following analyses to investigate the effects of capture from the wild on subsequent mortality risk. The elephants followed a typical bathtub age-specific mortality (Fig. 1), with mortality risk being higher at young and old age. To study mortality precisely, we relied on multi-model inference[47] based on an extension of Siler's mortality function[48] applied to 83,026 elephant-year observations from 5150 elephants. Our approach accounts for uncertainty in survival of elephants before their capture (left censorship), after their follow up (right censorship) as well as for uncertainty concerning the best model choice (see methods). For elephants aged between 0 and 55 years, maximal mortality was predicted during the first year of life with a mortality rate of 10.74% for males and 8.17% for females (model averaged prediction at age = 0, for the best living region and birth cohort; Supplementary Tables 1–3). Mortality then decreased by half every 2.28 years for males and 2.00 years for females, leading to the lowest mortality rate predicted at 14.07 years for males and 13.95 for females. After that, mortality increased slowly with age and doubled every 10.64 years for males and every 9.05 years for females. In addition to age-varying mortality effects, the Siler's function contains a parameter that models a mortality term that is independent from age. This was estimated to nearly zero for the birth cohort and living region associated with lowest mortality (i.e., Magway 1960; see Supplementary Tables 1–3). Departure from this 'optimal condition' resulted in an increase in yearly mortality of up to 0.83% depending on the region, and up to 0.43% depending on the birth cohort (Supplementary Tables 1–3; note that Fig. 1 represents the most common condition in our dataset—Sagaing 1980—and not Magway 1960). Overall, males showed higher mortality

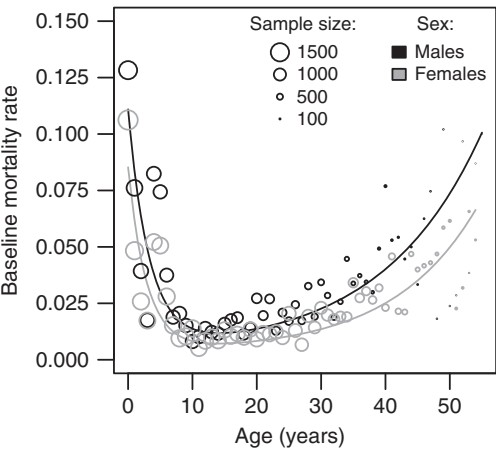

**Fig. 1** Baseline yearly mortality rates of males and females in Myanmar timber elephant population. Lines depict predictions from model averaging of all model fits presented in Table 1. Predictions were made for an average male (black) and female (gray) that would have been born in captivity from the birth cohort 1980 at Sagaing (the time and place accounting for the highest proportion of our data). Other conditions do not impact the general bathtub shape of the mortality function. Dots represent yearly mortality rates for each sex directly computed using raw data without accounting for capture status, birth cohort or living region. The diameter of the dots is proportional to the number of yearly elephant observations (Total = 83,026)

than females at all ages (Fig. 1), resulting in a predicted median lifespan of 30.81 years for males against 44.73 years for females for captive elephants living in 'optimal conditions' (Supplementary Table 4). More simple models of the general mortality pattern considering a single type of mortality or no effect of sex showed very poor explanatory power (see models 14–17 in Table 1).

**Capture and taming increase mortality for several years.** Second, we quantified how being captured from the wild between 1951 and 2000 (2072 individuals, born during 1902–1991; $N_{females} = 1340$, $N_{males} = 732$) modifies the general age-specific mortality pattern in each sex compared to 3078 ($N_{females} = 1590$, $N_{males} = 1488$) captive-born elephants born during 1925–1999. We found strong evidence that both males and females captured from the wild experienced an increased mortality compared to captive-born elephants. To show this, we estimated parameters precisely describing how capture (or any other effect correlated with it) triggered the mortality rate of elephants to depart from the baseline curve we just described (see Methods). Irrespective of how elephants were captured, we found that capture increased mortality (beyond the known immediate increase in mortality associated with the capture operation itself[40,42], which is not included in our dataset). This increase in mortality increased with the age of the elephant at capture (Fig. 2a) and was maximal during the year of capture (Fig. 2b). The best model fit does not consider differences between capture methods (model 10; Table 1) and predicts an increase in mortality in the first year following capture of 2.72% beyond the baseline mortality if the elephant is captured at age 5, 3.19% at age 10, 4.13% at age 20, or 5.99% at age 40. Once captured, the increase in mortality risk decreases with time. Based on our best model fit, we estimate that the excess in mortality associated with being captured reduced by half every 1.92 years. Consequently, the excess mortality per year associated with being captured reduces below 0.1% 9.15 years

**Table 1 Outcome of the model comparison.**

| Model | logLik | K | AIC | w | w_sum | w1 | b1 | w2 | b2 | w3 | w4 | w5 | b4 | b5 |
|---|---|---|---|---|---|---|---|---|---|---|---|---|---|---|
| 10 | −9698.6 | 27 | 19451.2 | 0.37936 | 0.37936 | s | s | s | s | s + t + l | 1 | 1 | 1 | 0 |
| 2 | −9692.8 | 33 | 19451.7 | 0.29885 | 0.67821 | s | s | s | s | s + t + l | c | c | c | 0 |
| 9 | −9698.6 | 28 | 19453.2 | 0.13956 | 0.81777 | s | s | s | s | s + t + l | 1 | 1 | 1 | 1 |
| 6 | −9697.7 | 30 | 19455.3 | 0.0484 | 0.86617 | s | s | s | s | s + t + l | s | s | s | 0 |
| 3 | −9697.7 | 30 | 19455.4 | 0.04761 | 0.91377 | s | s | s | s | s + t + l | c | 0 | c | 0 |
| 11 | −9701.9 | 26 | 19455.7 | 0.03949 | 0.95326 | s | s | s | s | s + t + l | 1 | 0 | 1 | 0 |
| 1 | −9692.8 | 36 | 19457.7 | 0.01488 | 0.96814 | s | s | s | s | s + t + l | c | c | c | c |
| 12 | −9701.9 | 27 | 19457.7 | 0.01453 | 0.98267 | s | s | s | s | s + t + l | 1 | 0 | 1 | 1 |
| 7 | −9701.5 | 28 | 19459.1 | 0.00741 | 0.99008 | s | s | s | s | s + t + l | s | 0 | s | 0 |
| 5 | −9697.7 | 32 | 19459.3 | 0.00655 | 0.99663 | s | s | s | s | s + t + l | s | s | s | s |
| 4 | −9697.7 | 33 | 19461.4 | 0.00237 | 0.999 | s | s | s | s | s + t + l | c | 0 | c | c |
| 8 | −9701.5 | 30 | 19463.1 | 0.001 | 1 | s | s | s | s | s + t + l | s | 0 | s | s |
| 13 | −9791.3 | 24 | 19630.5 | 4.3E−40 | 1 | s | s | s | s | s + t + l | 0 | 0 | 0 | 0 |
| 14 | −9825.6 | 18 | 19687.1 | 2.2E−52 | 1 | 1 | 1 | 1 | 1 | 0 + t + l | 0 | 0 | 0 | 0 |
| 15 | −9907 | 16 | 19845.9 | 7.3E−87 | 1 | 1 | 1 | 0 | 0 | 0 + t + l | 0 | 0 | 0 | 0 |
| 17 | −10244 | 14 | 20516.1 | 2E−232 | 1 | 0 | 0 | 0 | 0 | 0 + t + l | 0 | 0 | 0 | 0 |
| 16 | −10244 | 16 | 20520.1 | 3E−233 | 1 | 0 | 0 | 1 | 1 | 0 + t + l | 0 | 0 | 0 | 0 |

The 17 models we fitted are here sorted by increasing AIC. The best model is thus at the top (Model 10). For the "Model fit" part of the table (columns 2 to 6), columns show the log likelihood, the number of estimated parameters ($K$), the AIC value, the AIC weight of the model and the cumulative weight. AIC weights estimate the probability for a given model to be the best among the candidate models fitted. The "Model Structure" part of the table (columns 7 to 15) indicates which model parameters have been estimated. Model parameters are defined in methods. The symbol "0" indicates that the model parameter has not been estimated and was thus assumed to be null. The symbol "1" indicates that a single parameter value was estimated for a given model meta-parameter (see Methods). The symbol "s" indicates that one parameter value for each sex was estimated (hence the value of the model parameter can change according to the sex). The symbol "c" indicates that one parameter value for each capture method was estimated. For the model parameter $w3$, we consider that it can depend on the sex ("s"), birth cohort ("t") and living region ("l") but we did not consider interactions, leading to 14 estimates for $w3$ for "$t + l$" and 16 for "$s$" $+ t + l$". The model averaged estimates are provided in Supplementary Table 1 and all parameter estimates for all 17 models are provided in Supplementary Table 2 and their SE in Supplementary Table 3

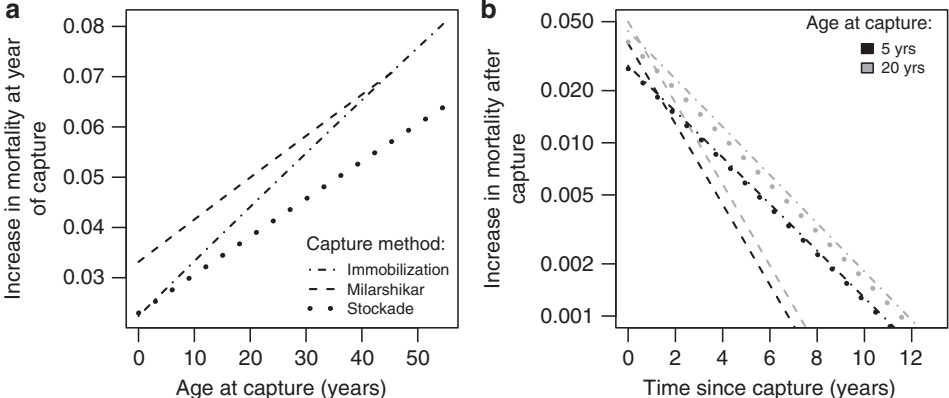

**Fig. 2** Additional mortality related to capture. This panel represents the short-term (**a**) and long-term (**b**) increase in mortality following capture for the three capture methods. Lines depict predictions from model averaging of all model fits presented in Table 1. Here, predictions would be identical for different birth cohorts and living regions, and although predictions are here computed for males they are virtually identical for both sexes (not shown). Short-term effect refers to the increase in mortality from the baseline level for the year following capture. This does not consider the immediate mortality during capture which is not recorded in our dataset but it could include effects caused by the taming procedure. Long-term effect refers to the increase in mortality from the baseline level as a function of the time elapsed since capture. We show the long-term effect of capture for an average elephant captured at 5 years old (black) and at 20 years old (gray)

after capture for those elephants captured at age 5, 9.59 years after capture for those captured at age 10, 10.31 years after capture for those captured at age 20, and 11.33 years after a capture for those captured at age 40. The pace at which the mortality rate of elephants returns to the baseline level the years following capture did not depend noticeably on the elephant age (Supplementary Tables 1–3; note also that black and gray lines are almost parallel in Fig. 2b). Similarly, although males have higher mortality in general (Fig. 1), the sex of elephants seems to exert a negligible influence on the effect size of the mortality increase in the first year following capture and on the long-term recovery time from the effects associated with capture (Table 1, Supplementary Tables 1–3). Together with already very high immediate mortality rates associated with capture operations[40,42], our results

demonstrate a considerable cost of capturing and taming elephants from the wild.

**The method of capture impacts little the mortality cost.** We found some weak evidence that the clear increase in mortality associated with capture and taming actually differs depending on the capture method; this suggests that whichever method was used to capture an elephant had little influence on its survival past the event of capture and that all methods were associated with a similar long-term mortality cost. In our dataset, 378 males and 805 females were captured by a stockade method (trapping all elephants in the same area) during 1951–1999 between ages 1 and 50 (mean ± SD: 14.32 ± 9.43 years; median:

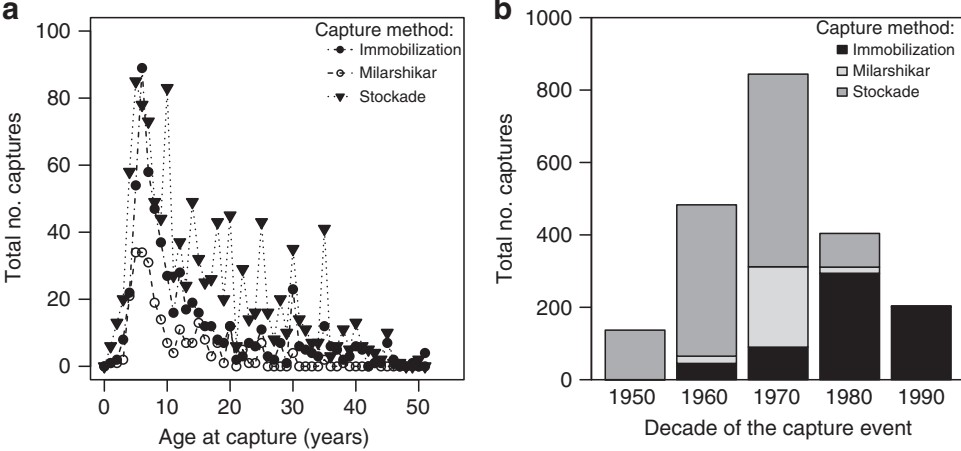

**Fig. 3** Capture frequencies by elephant age at capture and time period differentiating the three capture methods. **a** The number of 1 to 51-year-old elephants (males and females combined) captured by the immobilization, milarshikar and stockade methods. **b** The number of captured elephants in our dataset 1951–2000 by the immobilization, milarshikar and stockade methods. The cohort "1990" includes also 3 elephants captured by immobilization in 2000. $N = 5150$ elephants

12.00 years); 253 males and 377 females were immobilized using sedatives during 1963–2000 between ages 1 and 51 (mean ± SD: 12.54 ± 9.14 years; median: 9.00 years); and 101 males and 158 females were captured using milarshikar ("lasso") during 1968–1981 at ages 1–43 (mean ± SD: 9.99 ± 6.25 years; median: 7.00 years) (Fig. 3a, b). Whereas the stockade method is unspecific and generally results in many more adult elephants being captured with extremely high mortality rate during capture and taming period (not assessed in our study but estimated to be 30.1%[40]), immobilization (9.7% immediate mortality[40]) and milarshikar (4.6% immediate mortality[40]) techniques are more targeted. Immobilization is often used to capture elephants that are involved in conflict with people, and milarshikar for targeting young, apparently healthy individuals. Our model comparison reveals an important model selection uncertainty which prevents us from concluding unambiguously about the possible long-term differences between the capture methods in affecting mortality risk. On the one hand, the best model fit did not support any difference between the capture methods in increasing mortality risk (i.e., all were associated with an increased long-term mortality risk). On the other hand, the model with the lowest AIC presents a probability of selection only 1.27 (evidence ratio) times higher than the second best fit (ΔAIC = 0.48), suggesting that a replication of our analysis on another dataset from the same population could lead to a swap in the models fitting the data best and second best (Table 1). This second fit does suggest that the methods of capture influence differentially mortality (Supplementary Tables 2–3). Similar inconclusive results were also obtained by running the analysis as a survival analysis based on logistic regression or as a Cox survival analysis, which we do not present due to the limitation of these alternative approaches when it comes to studying precisely the effect of capture. In any case, model predictions—which we produced by model averaging so to precisely capture this model selection uncertainty—suggest that the different capture methods have relatively similar effects on long-term mortality (Fig. 2, Supplementary Table 4). Stockade and immobilization methods of capture appear to have very similar effects for the ages at which most elephants are captured. Milarshikar may lead to a slightly higher increase in mortality in the period following capture than other capture methods (Fig. 2a) but this is also the method for which the recovery after capture is the fastest (Fig. 2b). Again, our data clearly support

that all capture methods are associated with increased long-term mortality in both males and females in a similar way (Table 1).

To fully assess the influence of capture conditions (captive born or three capture methods) on elephant lifespan, we first predicted the median lifespan of elephants for these different conditions. Irrespective of the capture method, the age at capture, and the sex of the elephant, capture resulted in the reduction of the median lifespan by several years (Supplementary Table 4). Even in the most optimal combination of conditions (5 years old females living at Magway, born in 1960 and captured by milarshikar), the median lifespan was 3.17 years lower than for the captive counterparts. In less favorable conditions, the median lifespan of captured individuals was reduced by more than 7 years. To represent the overall effect of capture on survival, we predicted the probability of survival associated to each single elephant-year observation and averaged them so to obtain a single predicted survivorship curve for each capture method (Fig. 4). These predictions thus also take the individual differences in birth cohort and living region, capture method and age at capture, into account and differ from other predictions used in this paper which studied the effect of some parameters while holding all the others constant. Immobilization, followed by stockade, was the most detrimental method in terms of overall survival in both sexes. However, in agreement with the results presented above, there is very little difference between the capture methods. Instead, the figure shows that the difference between wild-born and captive-born individuals (irrespective of the method of capture) is as pronounced as the general difference in mortality between males and females in elephants.

None of the above results presented were confounded by general differences in age between wild-captured and captive-born individuals, capture practices or keeping systems in different logging regions of the country, or across the study period, which were controlled for in all models.

## Discussion
We took advantage of data available on wild-captured and captive-born Asian elephants in Myanmar to demonstrate the detailed, age-specific, adverse effects of capture from the wild on mortality in Asian elephants. We found that elephants captured from the wild had higher mortality rates than captive-born elephants at all ages. These effects stemming from the consequence of capture and also

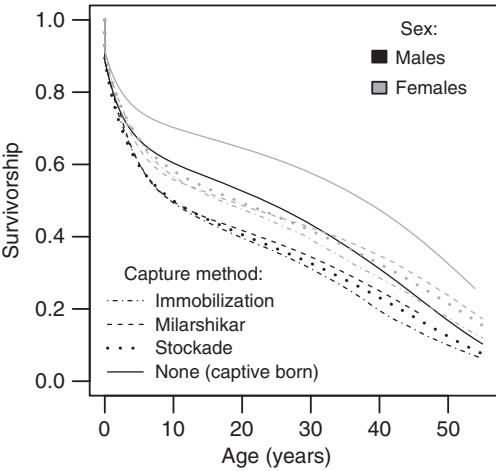

**Fig. 4** Overall impact of capture and sex on elephant survival. The predicted survival curves for captive-born elephants compared to elephants captured from the wild by immobilization, stockade or milarshikar methods for males (black) and females (gray). Lines depict predictions from model averaging of all model fits presented in Table 1. In contrast to Figs. 1 and 2, here the predictions are adjusted by the age, sex, origin and birth cohort of each observation in the dataset and then averaged by sex and capture method. Again, this does not consider the immediate mortality during capture which is not recorded in our dataset but it could include effects caused by the taming procedure. We used this type of prediction to account for the fact that different capture methods target elephants from different ages. For certain young ages for which no observation was available for a given capture method, we assumed that mortality was identical for such observations as the captive-born elephants

possibly from differences in how elephants perceived the taming process resulted in the reduction of (median) lifespan by several years. Second, we found that capture effects were more harmful the older the elephant was at the time of capture. Third, we found that the mortality of wild-captured elephants was the highest during the first year of capture decreasing slowly thereafter over the following years. Despite the paucity of research on the effect of capture and taming, higher mortality among wild-captured animals as compared to captive-born has also been reported in some studies with limited sample size on marine mammals[49,50] and primates[51]. In addition, that age increases this mortality cost has also been reported in killer whales[50] and in macaques[51]. We call for more studies of this kind to assess how general the negative effects of capture are across species.

The highest mortality increase in the year following capture is likely to be mainly related to capture-related injuries and trauma, as well as the subsequent harsh taming and breaking causing some of the recently captured animals that survived the capture-operation itself to die within a short time after entering captivity. However, in addition to acute stress from capture and training, wild-caught elephants are likely to suffer from chronic stress[52], although there are currently no studies directly comparing stress hormone levels or health of wild-captured and captive-born elephants in similar keeping systems. Long-term stress may lead to immune system dysfunction, higher mortality and reduced longevity[53] with reduced subsequent fitness[52]. Such effects of stress are also known to differ according to age[54]. Acclimation of the elephants to new environment presents challenges: in Myanmar, captured elephants are first broken and then trained to walk with hobbles to restrict their movement while night foraging, and the older the elephants at capture, the more difficult it is to move around with hobbles (to which captive-born individuals are used to after their taming around age 5), which is also likely to

restrict their nutritional intake. Also, older elephants take longer to tame than younger elephants and have likely been subjected to harsher taming and breaking process than younger elephants after capture. However, it is noteworthy that in our study population, captive-born elephants, too, go through taming after living comparatively free and unused to human handling and commands for the first five years of life, and similarly to wild-caught elephants, such taming is known to markedly increase their mortality[55]. Another factor potentially affecting long-term success is competition with locals (or unfamiliar food resources), which was hypothesized to be the reason for wild African elephants translocated to new areas suffering higher long-term mortality and having poorer body condition than elephants of the local population[56]. Finally, social disruption and defeat may have severe effects after capture for this highly social species with long-term effects on cognition, behavior and survival[57]. Although capture of very young (under 5 years old) animals is avoided (Fig. 3a), most elephants are captured at a relatively young age, normally when they are still dependent on their mothers, allo-mothers and older herd-mates for protection and guidance, and have not yet attained adult body mass. Such separation from herd-mates early in life is likely to be a powerful adverse experience in elephants[58]. Nevertheless, our results suggest that it was older rather than younger animals that suffered the most detrimental consequences for their survival following capture.

A previous study suggested that the duration of the negative effect of capture on female elephant mortality would last approximately eight years[34]. Also inter-zoo transfers reduce Asian elephant survivorship in the long-term, an effect detected up to four years after the initial transfer[34]. Our results extend previous work, and suggest that capture effects depend on the age at capture and may as well slightly depend on the capture method. In all groups, noticeable negative effect on mortality lasts roughly a decade and reduced the median lifespan by several years. Long-term studies on other species monitoring post-capture mortality rates have investigated the mortality consequences only for a relatively short time after capture, with the long-term effects being defined mostly as 30 days to few months after capture, the maximum follow-up duration being five years[14–16]. To our knowledge, there are no comparable studies investigating capture effects on mortality for decades in any species (but see Saraux et al.[59] on effects of tagging). However, stress levels after capture and translocation indicate that the long-term adverse effects may indeed continue several months after capture also in other large mammals, such as in zebra for 11–18 weeks[60] and rhinoceros for 9 weeks[61]. In contrast, results from some primates[62,63] and carnivores[64] suggest that handling, restraint, and confinement may have no effect on well-being, reproduction and longevity of these animals . Further work is warranted to confirm such patterns in different species and conditions.

Interestingly, there were little differences between capture methods in neither males nor females, and the recovery rate was similar at all ages. Although the differential effect of the capture methods appears weak, surprisingly, milarshikar presented the highest increase in short-term mortality but this method was also associated with the fastest recovery. Part of this effect may stem from selective disappearance that is known to bias survival estimates in demographic studies[32,59,65]; different capture methods may focus on individuals that differ in terms of pre-existing trauma e.g., due to previous human-elephant conflict, or in terms of adaptability to captivity and general health. With milarshikar, the captors in Myanmar usually target young animals in good condition, and it may be better suited to selecting the healthiest-looking individuals than other methods such as the stockade that are less discriminate. Furthermore, animals captured by different methods are subject to different levels of mortality during capture

operations that are likely to remove the weakest individuals from the sample to different degrees, thus affecting how robust the remaining survivors may be. Finally, given the increased mortality rates following capture, it is feasible that only the most robust wild-captured animals or those more adapted to semi-captivity contribute to older ages. Therefore, accurately assessing the negative effects of capture (or comparing capture methods) on long-term mortality, with the potential initial bias towards capturing individuals in good condition and large (presumably) selective mortality following capture, is challenging. Note however that the taming condition are similar irrespective of the capture method in our population. These challenges aside, our study design benefits from a keeping system in which wild-caught and captive-born elephants are subject to similar veterinary care, taming procedure, access to food, working regulations and data recording that allows us to control for many confounding factors. Furthermore, potential biases in sampling patterns would most likely under-estimate (and not over-estimate) the true negative consequences of capture and taming on elephant survival. Ideally, an evaluation of the safest capture method would include quantitative data on the injuries and fatalities occurring during capture operations[40,42], as well as its subsequent consequences on mortality, but only data on the latter were available in this study. The same is true for the taming procedure. These data show convincingly that all capture methods are detrimental both in the short-term or long-term, and capturing and taming wild elephants thus pose a substantial risk on their survival across several years.

Although the use of elephants for logging is not a situation that applies to many other populations and species, capture of elephants continues for legal or illegal purposes (for example see ref. [23,33]), and capture of various other species from the wild is practiced for diverse purposes each year (for example see ref. [1,2,66]). Therefore, our results are timely and have three main implications. First, long-term effects of capture are currently not considered in research design and conservation programs, but our results show that capture and taming can negatively influence animal performance for several years in elephants. Therefore, using wild-captured animals to supplement medical trial populations in some species (for example see ref. [67]), or as reference groups for species-typical parameter values (for example see ref. [3,4]), may lead to erroneous conclusions and both immediate (capture-related) as well as long-term effects of capture should be taken into account in further studies.

Second, our results offer interesting comparisons to welfare in zoo collections. In contrast to the situation in our semi-captive Myanmar population, captive(zoo)-born Asian elephants in European zoos have poorer survivorship than wild-captured animals[34,38]. Although wild-captured female Asian elephants entered zoos at a median estimated age of just 3.4 years, they show better survivorship as adults than zoo-born counterparts. Why do wild-captured individuals fare better than captive-born in zoos, but not in the semi-captive keeping system in Myanmar? Overall, elephants suffer considerably higher mortality rates in zoos when compared to wild or semi-captive populations, such as the timber elephant population studied here[34]. Indeed, captive-born elephants in the Myanmar population show comparable mortality to wild elephant populations[34], and the contrasting performance of wild-captured animals against the captive-born in zoos and timber camps highlights the problems that zoo elephants face. The reasons for the lower performance in zoos should be studied in detail, for instance, the effects of early-life stress and higher nutritional plane of animals, which have been suggested to cause this controversial pattern[38]. Thus, taken together, rich datasets available for diverse elephants together show that early experience can have profound and sometimes unpredictable effects of wild animals kept in captivity.

Finally, ~1000 Asian elephants currently live in captivity in zoos, safari parks, and circuses world-wide[12,41], but these populations are not self-sustaining due to high mortality and low fertility rates[68]. Consequently, 81% of the current European zoo populations were imported from range countries in Asia[12] (75% in North America[68]), 60% being wild-caught and 21% transported from timber camps[12]. Although capturing elephants from the wild may be sometimes necessary e.g., for conservation, veterinary and anti-poaching activities, similar large-scale wild-capture as in our study population to supplement captive population has occurred also elsewhere in Asia, because these captive populations have insufficient reproductive rates to maintain population sizes[43]. Captive-born elephants are regarded by keepers as more intelligent, less aggressive, easier to train, tractable and more reliable in temperament than those captured from the wild[30]. Our study implies that capturing wild individuals in elephants (and potentially among other species with slow life-histories[11,49–51]) is costly for individual longevity and alternative methods should be sought to boost captive populations in order to avoid further capture from endangered wild populations.

## Methods

**Study population**. Ancient Myanmar kings captured and tamed elephants as early as the fifteenth century, primarily for their armies[30,40]. Since the early eighteenth century, successive Myanmar governments have practiced selective logging of teak using elephant draught power in timber extraction[30,40]. Today, half of the captive elephants in Myanmar ($N$ ~2700) are government-owned and used in forest camps as riding, transport and draft animals. At night all elephants forage in forests, as part of their family groups unsupervised. Breeding rates are natural and not managed by humans with many captive-born calves thought to be sired by wild bulls, and calves born in captivity are cared for by their biological and allo-mothers.

The Myanmar government has monitored their elephant population for over one hundred years, fully recording the life-history of captive timber elephants. Our dataset has been collated from elephant log-books and annual extraction reports archived and maintained by the Myanma Timber Enterprise. The state ownership of thousands of elephants has enabled the recording of the following data for all registered individuals: registration number and name; capture status (wild-caught or captive-born); date of birth; mother's registration number and name; method and year of capture (if wild-captured); year or age of taming; living region; dates and identities of all calves born; date of death or last known date alive; and cause of death. The individual elephant log-books are maintained by local veterinarians and regional extraction managers in order to check each elephant's health condition and working ability. While the ages of captive-born elephants are known from precise dates of birth, wild-caught elephants are aged by comparing their height and body condition with captive elephants of known age. The extent of depigmentation (freckles) on trunk, face and temporal areas, and the degree of folding of the upper edge of the ear increase with age, while hairiness of the tail tuft and degree of corrugation or wrinkliness of the skin reduce with increasing age. The Myanmar elephant catchers and trainers take careful consideration of all physical features in estimating age of wild-caught elephants. The error in these estimates is unknown, but is likely to be within a couple of years for young animals that continue to grow[69] (under 20), which form the majority (72%) of those captured ($N = 1497/2072$).

**Capture and taming**. Capturing is usually practiced in the cool season by three alternative methods[39]. First, a stockade method involves driving whole family units or elephants within the same area into a stockade. After capture, one animal at a time is forced out to nearby "cradles" designed for taming procedures[31]. Most elephants captured by stockades are females, including matriarchs, pregnant females, juveniles and mothers with suckling calves, and due to capturing entire families/groups of elephants, the mean capture age is higher than that of the other two methods. Second, an immobilization method has been practiced since 1961 using Etorphine hydrochloride[31,39], an opiate-derived narcotic analgesic producing pharmacologic effects similar to those of morphine, which is used in quantities approximating to 1 mg/400 kg. The sedated elephant is tethered and given diprenorphine as a reversal agent at two times the dosage of Etorphine and then dragged to breaking camp by trained elephants. Finally, a milarshikar method involves chasing the chosen wild elephant using trained elephants and noosing them when isolated from the herd, and then dragging them to the camp as above. Those with tusks, suckling calves, or that are pregnant are avoided, and milarshikar is generally not suitable for capturing mature elephants >2.3 m in height because of higher stress and trauma during the capture and breaking procedures.

All captured elephants undergo a taming and "breaking" procedure immediately after capture (normally at nights to avoid heat stroke) that lasts minimum of one month depending on individual temperament, with older elephants taking longer

to tame than younger[31] or captive-born individuals[46]. The taming undoubtedly induces stress for the animal and compromises the welfare, especially during the first few days. Newly caught elephants and 4–5-years-old captive-born calves are first put into crushes (or cradles[46]). Trainers use food and water as a reward for successful training. Later, trained elephants are brought alongside the crush and fed/handled in view of the captive. During the taming progressing, the elephant is held via the breast band (cradle) or tied to a tree. Elephants commonly resist training and reject food/water for the first few days. Males take longer than females to become tame and are more likely to be traumatized by physical punishment and/ or self-inflicted wounds through struggling[31]. Subsequently, the elephant is trained to respond to verbal commands, such as"stop", "come" or "still". Although captive-born elephants are separated from their mother and tamed at 4–5 years of age using similar methods[46], they grow up having regular contact with their maternal herd and to some extent mahouts (riders). Consequently, the wild-captured elephants are likely to go through rougher taming period and are likely to suffer from more stressful psychological and physical trauma during taming[31], and the initial adaptation to captivity usually exceeds the one month duration used for captive-born calves. Following the initial adaptation there is a less stressful phase that may last more than 10 years, in which elephants adapt to captivity and their position as subordinates to mahouts[31].

After taming, elephants are classified as trained calves and assigned permanent individual registration numbers, mahouts and logbooks for recording biodata (sex, temperament, musth, mating, calving, veterinary intervention etc.), and our study utilizes these records (but excludes events and mortality prior to this point). They are used for light work and transport until the age of 17, at which point they are utilized within workload and rest period restrictions set out by government legislation: all state-owned elephants are subject to the same regulations for hours of work/week, working days/year, and tonnage to extract/elephant. For example, in 2010 all mature elephants (17–55 years of age) worked 3–5 days/week, for 5–6 h/ day (maximum 8 h), with a maximum extraction rate of 400 tons/year. Working females are given rest from mid-pregnancy (11 months in gestation) until the calves reach their first birthday. Mothers are then used for light duties but allowed to nurse the calves on demand[39]. Elephants "retire" at 55, but their records are maintained until death.

**Statistical analyses**. The entire studbook at the time of the analysis included 8006 elephants born (or estimated as born for wild-caught individuals) between 1858 and 2000. Full lifespan is known for 3826 elephants, whilst right-censored lifespan was used for 2975 elephants that were either alive in 2000 or disappeared earlier without an exact date of death. All calves born in 2000 were removed ($N = 69$) from the analyses, given they were censored under the age of one. Elephants born before 1900 ($N = 250$) or captured before 1951 ($N = 1239$) were also excluded because of incomplete records, as were 83 individuals with erroneous death data, 6 calves born with unknown sex and 4 individuals captured older than the maximum age investigated in the survival analysis ($\geq 55$ years). The remaining 5150 animals born during 1900–1999 ($F = 2930$, $M = 2220$) come from 39 timber extraction areas within 10 of the 14 regions in Myanmar: Chin ($N = 21$)/Rakhine ($N = 148$) (joined due to low sample size from Chin), Ayeyarwady ($N = 92$), Bago ($N = 531$), Kachin ($N = 620$), Magway ($N = 607$), Mandalay ($N = 716$), Sagaing ($N = 1463$), Shan ($N = 531$), Tanintharyi ($N = 50$) and unknown ($N = 371$).

We used discrete time survival analyses inspired from a logistic regression modeling framework[70] and Siler's survival function[48] to investigate the effects of capture on survival until age 55. The approach we used allows: a detailed analysis of the effects of time-dependent variables on the elephant's probability of dying over discrete time intervals (years); individuals to enter the analyses at varying ages (birth or capture age); and inclusion of data for those individuals with missing exact death date (still alive or disappeared individuals as censored observations), thus avoiding a biased sample towards those dying at a young age or with complete records only. Captive-born elephants were incorporated into the analysis from their birth, and wild-caught elephants from their capture age onwards. This analysis allowed us to estimate the elephant's risk of dying in each year from age 0 to 55 years, while investigating the effects of capture status or methods and other variables in comparison to captive-born elephants. Survival was investigated until the age of 55 for males and 54 for females, because beyond this age data was insufficient for all capture methods or capture status (the maximum age of death/ censoring: captive-born 55 years; stockade 83 years; immobilization 62 years; milarshikar 56 years).

We predicted yearly probability of death according to the following non-linear model:

$$p = w_{1s} \times e^{(-b_{1s} \times \text{age})} + w_{2s} \times e^{(b_{s2} \times \text{age})} + w_{3stl}$$

$$+ I \times \left( (w_{4sc} - 1)e^{(-w_{5sc} \times \text{age})} + 1 \right) \times e^{\left( -b_{4sc}^{\left(1 + b_{5sc}^{\text{age}}\right)} \times \text{timecap} \right)} \tag{1}$$

This model is an extension from the model introduced by Siler[48] to specifically model the effect of covariates on mortality. The first line of the equation is similar to Siler's original mortality model. The only modification is that we here allow the

Siler's original coefficients (here called $w_1$, $b_1$, $w_2$, $b_2$, and $w_3$) to vary depending on the sex, region and birth cohort of the elephants. Within a given sex, region and birth cohort this part of the model is thus identical to the original formulation. This first equation represents what we call the baseline mortality, that is the mortality not accounting for the effects of capture (although it could be seen as the mortality of captive elephants, the parameters are estimated using information from all elephants, which is why we call it "baseline"). The baseline mortality model is made of 3 components. The first component—called the "mortality for immature animals" in Siler's paper—models the decrease in mortality rate during early life, the second one—called the "mortality for senescence"—models the increase in mortality rate at old age, and the third one—called the mortality for mature animals—models a constant mortality rate, independent from aging which we thus prefer calling "mortality independent from ageing". Importantly, all three components contribute to the mortality of elephants at all ages since the mortality is the sum of the three components; only the numerical values vary in a way that gives more weight to a given component at a given age. The original model was introduced by Siler[48] to efficiently account for and disentangle these three types of hazards shaping the survival of many organisms. In the particular case for which the "mortality for immature animals" and the "mortality independent from ageing" are null and only "mortality for senescence" is considered, the model reduces to the Gompertz equation[71]. If only the "mortality for immature animals" is null, the model corresponds to the modification of the Gompertz equation[70] proposed by Makeham[72,73].

The baseline mortality rate is thus described by 5 meta-parameters ($w_1$, $b_1$, $w_2$, $b_2$, and $w_3$). We considered that the first four parameters could differ between sexes (subscript "s" in the equation above, 2 levels: males, females). We also considered that the mortality for mature animal ($w_3$) differed between sexes (subscript "s"), birth cohort (subscript "t", 4 levels: 1940, 1960, 1980, 2000) and living region (subscript "l", 10 levels: unknown, Ayeyarwady, Bago, Kachin, Magway, Mandalay, Rakhine, Sagaing, Shan, Tanintharyi), and we considered that these three sources of variation were additive and did not interact leading to $2 + 4 + 10 = 16$ possible values for $w_3$. To fit the most complex model for baseline mortality, one must thus estimate $4 \times 2 + 16 = 24$ parameter values. We rely on the following half-life metrics to describe age effects: during immaturity the annual mortality rate decreases by half every $\log(2)/b_1$ years; during senescence the annual mortality rate doubles every $\log(2)/b_2$ years.

The second line of the equation describes the effect of capture and taming. The index $I$ is thus equal to 0 for captive-born and 1 for wild-born elephants. Then follows a product between two main terms. The first main term represents the short-term effect of capture. It models the effect at the year of capture. The mortality increase by $w_4$ for an elephant of age zero, then $w_5$ describes how the effect modeled by $w_4$ changes with the elephant age. If $w_5$ is positive this short-term mortality increases with elephant age, if it is negative the short term effect decreases, and if it is null the short-term effect is independent from the elephant's age. The second main term models how the short-term increase in mortality changes with time since the capture event. The initial mortality cost decreases if $b_4$ is positive, increases if $b_4$ is negative, and it remains constant if $b_4$ is null. The last meta-parameter—$b_5$—models how the elephant age influences the effect of $b_4$: if $b_5$ is positive the effect of $b_4$ increases with age, if $b_5$ is negative it decreases with age and if $b_5$ is null, the elephant age bears no effect of the long-term effect of capture. Each of the 4 coefficients of this second part of the mortality model ($w_4$, $w_5$, $b_4$, and $b_5$) could be considered as null, constant (1 parameter value for each of these 4 meta-parameters), vary according to the sex (2 parameter values for each), or vary according to the capture method (subscript "c" in the equation above; 3 parameter values for each). Only models for which those 4 meta-parameters were considered as null did not account for the effect of capture. All the other parameterizations do account for a possible effect of capture. Hence, meta-parameters $w_4$, $w_5$, $b_4$, and $b_5$ with subscripts "1" and "s" in Table 1 do consider the effect of capture as such but not differences between capture methods. To fit the effect of capture for the most complex model (model 1 in Table 1), one must thus estimate $4 \times 3 = 12$ parameter values. The most complex model thus contains a total of 36 parameters.

We estimated the parameter values by maximum likelihood using R v 3.4.4 (R Core Team, 2017[74]). Specifically, for each year from birth/capture age to 55 years, we coded the mortality of each animal as a binary variable (0 = alive vs. 1 = dead during the observation year), with each animal exiting the analyses at death or last known age alive (censored observations). Given a set of parameters, the mortality model can be used to predict the probability for each observed binary event and the product of these probabilities directly gives the likelihood of the model (as in a classical Bernoulli model)[75]. One can thus look for the parameter values maximizing such likelihood. We performed such estimation using a bound-constrained optimization by quadratic approximation (BOBYQA[76]) implemented by the package nloptr 1.0.4, an R interface to the NLopt library[77]. We assumed that all parameters of the model are positive and did not constrain their upper bound and confirmed that this assumption was correct based on estimation obtained. We approximated the standard errors (SE) on each parameter estimate, for each of the model presented in Table 1, by refitting each model 40 times after resampling the original data at the level of the individuals with replacement (i.e., non-parametric bootstrap). Those SE are presented in Supplementary Table 3.

We compared the predictive power[78] of 17 different fitted models (Table 1) based on their Akaike Information Criterion (AIC) and proceeded to a multi-model inference[47]. The different models correspond to different biological

hypotheses. Models 14–17 (Table 1) consider no effect of sex and no effect of capture. They differ in how mortality is assumed to change with age. The model 16 corresponds to the classical Gompertz mortality model[71]. All other models considered the effect of sex on the mortality, starting with the model 13 which considers the effect of sex but no effect of capture. Moreover, 3 groups of 4 models were considered to model the effect of capture: one with an effect of capture method on the effect of capture (models 1, 2, 3, 4), one with an effect of sex on the effect of capture (models 5, 6, 7, 8), and one with no effect of sex and capture method on the effect of capture (models 9, 10, 11, 12). Within each group, we consider 4 possible aging effects: an influence of ageing on both short-term and long-term mortality due to capture (models 1, 5, 9), an influence of ageing on short-term mortality only (models 2, 6, 10), no influence of ageing on both short-term and long-term mortality (models 3, 7, 11), and an influence of ageing on long-term mortality only (models 4, 8, 12).

To compute estimates accounting for the model selection uncertainty we performed a so-called model averaging by weighting the estimates of all models (Supplementary Table 1) by the AIC weight of each model[47]. The AIC weights estimate the probability of model selection; that is, the frequency at which a given model would be best across all samples, or equivalently the probability that the best model in a given random sample really is the best model in the population. We computed the weight as suggested by Burnham and Anderson[47]. We obtained 95% confidence intervals (CI) using the traditional Wald method after deriving the weighted average SE of each estimate following equation 5 in Lukacs et al. 2010[79]. Parameters not estimated for a given model were assumed to be null and were considered as such during the averaging.

To assess the influence of each method on elephant lifespan, we used the computation of median lifespans, the minimal age below which half of the individuals are predicted to be dead. We chose not to use full life expectancies because we did not estimate mortality after 55 years and thus predictions for life expectancies would be unreliable. Because our formula (equation 1) bears no simple relationship between hazard and survivorship, we estimated median lifespan numerically from the cumulative products of yearly survival probabilities. We then interpolated between these yearly measurements assuming piecewise constant hazards (i.e., $\ln S(k + t) = t \ln S(k + 1) + (1 - t) \ln S(k)$, with $k$ the age in years and $t$ the increments within years).

There are no ethical issues related to the demographic data on Asian elephants because it is collected from historical records and no experimental protocols were used in collecting the dataset.

**Data availability**. The computer code and small subset of data accompanying this study are provided as an R package available from https://github.com/courtiol/SileR. Whole data for re-analysis are available on request from Prof. Lummaa.

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

## Acknowledgements

We thank the Ministry of Natural Resources and Environmental Conservation, the Government of the Union of Myanmar for giving permission to work with Myanma Timber Enterprise (MTE), Myanma Timber Enterprise and all the vets and officers involved in data collection as well as the Myanmar Timber Elephant Project members for help and support. We also thank John Jackson for helpful comments on an earlier version of the manuscript and Marcus Rowcliffe for assistance during the initial phases of this study, European Research Council (VL), Academy of Finland (VL) and Kone Foundation (ML) for funding.

## Author contributions

K.U.M. collected the elephant dataset, M.L. and V.L. conceived and designed the paper. A.C. and M.L. performed the analyses. M.L., V.L., A.C., and K.U.M wrote the paper. All authors read and approved the final manuscript.
