## [Peer Review File · Nature Communications]

Reviewers' comments:

Reviewer #1 (Remarks to the Author):

This paper reports an original analysis of the influence of capture on age-specific survival of Asian elephants. From an exceptionally high quality dataset (> 5000 individuals, 50 years of monitoring), the authors reported several major findings including that (1) captured elephants from both sexes displayed lower survival at a given age than elephants born in captivity, especially when animals caught were immobilized, (2) the immediate extra-mortality of wild caught elephants relative to captive-born ones increased with age, and (3) the duration of the negative influence of capture on survival was shorter in young than in old elephants. These findings are timely and provide an important contribution to improve our knowledge of age-specific mortality and thereby the management and conservation of vertebrate populations.

However, I found a couple of problems that need to be solved.

First, the inclusion of a linear or quadratic effect of age in the models is not optimal. Moreover, the authors did not provide any evidence that a quadratic model reliably accounted for observed age-specific variation. A full age-dependent model should thus be fitted and compared explicitly to the quadratic model. The authors should also fit proper age-specific models of survival such as Gompertz or Weibull models that are commonly used to analyse age variation in survival of vertebrates (see e.g. Ricklefs 1998 *Am Nat*, Tidière et al. 2015 *Evolution*).

Second, too many tests are displayed in the Table 1 and it is especially difficult to identify the best model. A model selection procedure (based on some criterion like AIC, see Burnham & Anderson 2002's book on model selection) is required. In addition, the effect sizes (with SE) of the retained effects should be displayed in a table.

Lastly, elephants are especially slow-living species, such as the other case studies mentioned on l. 212. It might thus be that the negative influence of survival reported here could be mostly observed in species with very slow life histories. □ This needs to be discussed (see Tidière et al. 2016 *Scientific Reports* for recent evidence of the effect of the pace of life on the response of age-specific survival patterns to environmental conditions).

Minor comments:

l. 211: Remove "significantly"

l. 300: Remove "significantly"

l. 367-368: How was defined "individual temperament"? If available for all elephants captured, it might be worth to include that variable in the survival analysis.

The figures are not informative in absence of the raw data

J.M. Gaillard

Reviewer #2 (Remarks to the Author) (In addition, please see comments from reviewer #2 in the attachment):

1. The paper presents unique and highly valuable data based on individual records for working elephants from Myanmar.
2. The authors used the data to address important and critical questions about differences in survival between captive-born and wild-caught Asian elephants. They proposed to use this data set to investigate whether:
 - a. wild-captured individuals have compromised survival at any age compared to captive-born individuals
 - b. the effect of capture on survival depends on the age at capture,
 - c. the effect of capture on survival depends on the time since capture
 - d. there is variation in survival based on the capture method used
 - e. there is a sex difference in the effects of capture on survival.
3. The methods and statistical analysis used are sound and indeed the authors report very important and original findings demonstrating that survival is significantly different between these groups of elephants, with lower survival for wild-caught males, females, etc.
4. I agree with the authors that their findings demonstrate that the practice of capturing elephants for sustaining captive populations is not only detrimental because it reduces wild elephant populations. It also doesn't provide a viable solution for sustaining captive populations.
5. The main problem I see with this work is that the authors link survival back to the capture and capture method, rather than to the substantial trauma and injuries wild-caught elephants experience during the breaking and taming period or the differences in management of wild-caught and captive-born elephants. For example:
 - a. Wild-caught elephants suffer much longer and more traumatic breaking and taming periods. Practices include:
 1. long periods of confinement to crushes and cradles;
 2. animals often are tied at their feet and forced into very painful physical positions. Ropes used to tie them cause deep and festering wounds on all four extremities which often get infected. The scars from these wounds remain for life.
 3. periods of starvation alternated with offering food and water;
 4. shouting at the animal, beating, poking with sharpened sticks, and many other forms of mistreatment.

b. Wild-caught animals generally do not adapt as well to captivity and logging work as captive-born. As a consequence, they tend to be punished more and often are treated more harshly than captive-born animals.

6. Similarly, the authors assume that differences in survival are directly related to the capture method (e.g. immobilization, stockade, and melashika) and do not consider that many of the observed differences may have to do with differences in a) immediate capture mortality which may result in selection the healthiest and most adaptable elephants via stockade; b) the population and individuals targeted by difference techniques; and c) potential for pre-capture trauma and injury from human-elephant conflict suffered by individuals captured via immobilization. The authors mention some of these factors in the discussions but because the hypotheses, methods and results are set up to test for capture rather than management, these arguments don't stand out and are not explained or discussed sufficiently. This may have long-lasting impacts on future elephant management recommendations.

For example, the uninitiated reader will assume that immobilization is much worse than stockade as a capture technique. However, stockade has a much, much higher immediate mortality, much of which probably is never reported. Stockade will result in the increased capture of older individuals as well as in the capture of more bull elephants because it is unselective. However, because mortality is so high, stockade capture may select for the hardiest and most adaptable animals to be transferred into captivity, the others likely perish during the process.

Melashika likely is the “softest” capture technique (as the authors rightly report), because it specifically targets young and docile animals. Older and more rambunctious animals are nearly impossible to catch with this technique. These elephants will be easier to break and be less likely to suffer long-term injuries as a result.

Immobilization also is very specific but it allows for the capture and removal of “problem” elephants that are engaged in crop raiding and other types of human-elephant conflict. Removal of these “problem” elephants and transfer to captivity is a relatively common practice in Myanmar. Many of these animals may already be severely stressed, traumatized, and injured prior to the capture, because in crop-raiding conflicts local people throw rocks, torches, large china crackers, at these elephants.

While it is possible that the drugs used for immobilization have long-term consequences for survival, it seems much more likely that pre-capture trauma and injury, breaking and taming, and differential treatment in working camps play a significant role in reduced survival of these elephants. Controlled study of the use of immobilization drugs on elephants in captivity and the wild should be possible given the number of animals that are treated this way. Also, one would

think that dosage and application are important for these considerations.

7. It seems that males may have higher mortality because they generally perform the most difficult and dangerous work, pushing and dragging logs in extreme terrain. Consequently they have a much higher risk for injury and mortality. Additionally, I believe pregnant females and young mothers have reduced workloads and are either not used at all or are used to transport lighter loads.

Immobilization/capture are critical tools for elephant management, specifically for capturing and treating injured elephants (e.g. elephants in Asian frequently get injured by wildlife snares), relocation of problem elephants, and satellite tracking to better study their ecology. Some of these techniques are also critical in anti-poaching activities. It will be critical to better understand whether immobilization has significant negative effects on these wild elephants and to balance them with the potential benefits from the immobilization. Thus, it is important to sufficiently discuss the findings on immobilization from this research and ensure that the findings are not the result of other effects that were not sufficiently measured or described.

8. Wild-capture of elephants is a huge conservation problem and likely has led to significant declines in wild elephant populations. The authors acknowledge this but it would be useful to see more discussion of this topic. Much of this practice was meant to supplement captive populations that are not self-sustained. The results in this paper shows that such practices are very wasteful because of a) the mortality associated with capture and b) the decreased survival of these wild-caught elephants in captivity.

Reviewer #3 (Remarks to the Author):

This is a very interesting (and disturbing) paper, based on an unusual dataset and using sophisticated statistical methods.

Our two main critical comments are to do with the context and modern-day relevance of the work. First it sounds as though the practice of catching Asian elephants from the wild in Myanmar ceased by 2000, and although later the authors attempt to generalize from this animal to other species it is not clear to which such data can really be extrapolated, partly as elephant capture methods seem unusually harsh because of these animals' great size, and also because capture from the wild is being phased out by some bodies (for example, the catching of wild animals for research in Europe). Can the authors be more precise about specific, current practices their data apply to? Are these data truly relevant today? If so, in which countries and to which species?

Second, we already know of substantial birth origin effects on elephant survivorship (not least the highly adverse effects of being captive-bred in zoos), and of effects of capture from the wild that elevate mortality for some years: such results feature in two papers in which the lead author is an author (Clubb et al. 2008, their ref. 25; also Clubb et al. 2009 in *Animal Welfare* which discusses potential mechanisms for such effects; plus also Dr. Mar's PhD thesis). So the Introduction as written is quite strange as it makes this paper look totally novel, and as though the authors are writing on a blank slate (which isn't so!) Please make the Introduction more accurately, transparently representative of the current state of knowledge of birth origin effects, including what we already know about elephants. Then what this particular study specifically adds that is novel should be spelled out more clearly.

Minor comments:

The first few paragraphs of the Discussion currently read more like an introduction than a wrap up of findings. If more of the details presented here were instead in the Introduction – with emphasis on the topics highlighted above - that would give the reader a better sense of why the authors did this work. Relatedly, why is there historical information in the Methods section? Since Methods come after the Discussion, there needs to be more of this background -- e.g on the capture methods -- in the Introduction section instead, otherwise the meaning of the results are not clear.

Lines 108-109: how can data from captive born animals predate (since starting 1925) data from wild-caught animals (which start 1951)?

Line 440: How was age at capture assessed for wild animals? Crucial if age is to reliably be included in the models.

Line 170-172: "There were however few differences..." this sentence was unclear. What is meant by few differences? Elaborate on the effect of milarshikar on survival.

Lines 211-213: Suggest checking the stats in these cited papers as some of the cetacean work in particular is not very good...

Lines 218-220: Not entirely clear why changes in cognitive function are being talked about here. On the other hand if you want an overview of how stress affects morbidity and mortality, this is good and may well be useful:

Walker, M. D., G. Duggan, N. Roulston, A. Van Slack, and G. Mason. Negative affective states and their effects on morbidity, mortality and longevity. *Animal Welfare* 21, no. 4 (2012): 497-509.

Line 369: “The taming undoubtedly induces stress and compromises welfare...” how so? What indicates this? Is perhaps in danger of appearing subjective.

Lines 458-459: How were the hierarchical models created and what objective model selection criteria were used (e.g. AIC)?

Figure 2: Make it obvious in the description that WC is wild captured and CB is captive born. Also maybe just state that colors represent sex throughout all the graphs since the color legend is only in A.

Point-by-point responses to reviewer comments

We have revised our manuscript according to the reviewers' comments as outlined below. The comments by the all referees are in normal font, while ours follow in italics.

Reviewers' comments:

Reviewer #1 (Remarks to the Author):

This paper reports an original analysis of the influence of capture on age-specific survival of Asian elephants. From an exceptionally high quality dataset (> 5000 individuals, 50 years of monitoring), the authors reported several major findings including that (1) captured elephants from both sexes displayed lower survival at a given age than elephants born in captivity, especially when animals caught were immobilized, (2) the immediate extra-mortality of wild caught elephants relative to captive-born ones increased with age, and (3) the duration of the negative influence of capture on survival was shorter in young than in old elephants. These findings are timely and provide an important contribution to improve our knowledge of age-specific mortality and thereby the management and conservation of vertebrate populations. However, I found a couple of problems that need to be solved.

We are pleased to hear that the reviewer appreciates the quality of our dataset and finds our results timely and important. We want to particularly thank for the very helpful comments and advice we obtained. Following the Reviewer's comment we have extensively revised our statistical analysis, but this new analysis confirms each of the 3 main findings highlighted above.

First, the inclusion of a linear or quadratic effect of age in the models is not optimal. Moreover, the authors did not provide any evidence that a quadratic model reliably accounted for observed age-specific variation. A full age-dependent model should thus be fitted and compared explicitly to the quadratic model. The authors should also fit proper age-specific models of survival such as Gompertz or Weibull models that are commonly used to analyse age variation in survival of vertebrates (see e.g. Ricklefs 1998 Am Nat, Tidière et al. 2015 Evolution).

We are grateful for the reviewer's remark which led us to revise our statistical approach. Our new approach is more accurate than the previous one; it no longer depends on the assumption of a quadratic effect for age; it recovers all three main findings mentioned above; and it allows for a simpler presentation of our results.

We had formerly assumed a quadratic effect for age because the data show high mortality at both early and old age, and because an approach based on Generalized Linear Models (GLM) implies to consider polynomials (the quadratic function being a polynomial). The choice for relying on GLM had been itself motivated by the need for performing a survival analysis which could tackle left and right censored data via the estimation of yearly mortality rate, and the need to model time varying effects. Traditional implementation of survival analyses are not well suited to combine these difficulties and earlier attempts using BaSTA (an R package implementing flexible MCMC based inference to tackle such kind of complexity) were not promising (lack of convergence after many days of runtime).

The approach we are now using is a full age-dependent model as you proposed, which considers that the baseline mortality rate of individuals follows a model introduced by Siler (1979). As we explain in our methods, this model allows to efficiently account for high early and late mortality, without assuming that these rises in mortality are symmetrical (quadratic functions do make this strong assumption). Siler's model is a generalization of the Gompertz equation. Siler's model is also more general than the modification of the Gompertz equation proposed by Makeham. (As such Siler's model can reduce to these two classical survival functions as particular cases.)

We extended Siler's model to model the effects of capture on mortality. As in our initial GLM-based approach, we fitted the new model to annual observations to account for left and right censorship (see methods for details). We fitted this non-linear model by maximum likelihood using a general optimization library.

We tried different functions to extend Siler's model so to include capture effects and obtained qualitatively similar results. As mentioned above we also retrieved all three main findings from our former GLM based approach. We are thus confident that our results are robust and decided to only present the approach that seems best to us for motivations of clarity and scope. We hope you agree and now find our modelling approach suitable.

Second, too many tests are displayed in the Table 1 and it is especially difficult to identify the best model. A model selection procedure (based on some criterion like AIC, see Burnham & Anderson 2002's book on model selection) is required.

We agree with the reviewer and we are now using an AIC based approach to analyze our model.

The new statistical approach we rely on requires to estimate many parameters. Minimizing the risk of overfitting the data is therefore another argument for choosing an AIC based approach, as proposed by the Reviewer.

We thus fitted a set of candidate models corresponding to clearly defined hypothesis. We then chose to perform a model averaging of all the candidate models (weighted by Akaike weights). This multi-model inference methodology recommended by Burnham and Anderson, combines the benefits of model selection without the costs. Indeed, like model selection our approach does not draw inference on overfitted models. But contrary to model selection, model averaging allows to take model selection uncertainty into account, and it does not suffer from an increase in false discovery rate rising from multiple testing.

In addition, the effect sizes (with SE) of the retained effects should be displayed in a table.

We now provide all parameter estimates for all candidate models and the parameter estimates obtained by model averaging in a supplementary table.

Unfortunately, we cannot provide SE for our parameter estimates. Due to the redundancy in parameters, inverting the Hessian matrix to obtain asymptotic SE turned out to be particularly difficult and did not succeed for all models. Besides, it is not clear to which

extent an asymptotic approximation would make sense. An alternative would be to compute SE by non-parametric bootstraps or to build confidence intervals by likelihood profiling. However, due to the size of our dataset, refitting our models multiple times was not possible within the time frame available (each model would take many days of running time on a super computer) and the ecological cost associated with this operation does not seem justifiable. We hope you find our revised tables adequate and informative enough; we remain open to further suggestions for improvement of course.

Lastly, elephants are especially slow-living species, such as the other case studies mentioned on l. 212. It might thus be that the negative influence of survival reported here could be mostly observed in species with very slow life histories. This needs to be discussed (see Tidiere et al. 2016 Scientific Reports for recent evidence of the effect of the pace of life on the response of age-specific survival patterns to environmental conditions).

Thank you for this point, we agree and have included this citation to lines 42-47 and end of discussion, lines 376-379.

Minor comments:

l. 211: Remove "significantly"

Removed.

l. 300: Remove "significantly"

Removed.

l. 367-368: How was defined "individual temperament"? If available for all elephants captured, it might be worth to include that variable in the survival analysis.

Comment concerning lines 428-431. Thank you, this would be an interesting further analysis, however we do not have individual personality or temperament recorded for historical study elephants that are now deceased, only for some of the currently living animals. This precludes from conducting a survival analysis on the effects of temperament at least for the time being.

The figures are not informative in absence of the raw data

We appreciate this suggestion and as a consequence, we have now added raw data on the main figure showing how mortality varies with age, but we chose not to display the raw data on other figures.

We are not convinced that showing raw data is really a good thing when predictions and observations are not directly comparable. In our case, predictions and observations are not directly comparable for several reasons. First of all, the process being analyzed is probabilistic but the observed outcome is binary. Because of this, representing mortality using raw data implies to divide the dataset into small subsets, for which estimations of the probability of occurrence of very rare events are highly unreliable. Second, predictions are controlled for certain covariates (here birth cohort, living region), while raw data are not. Finally, the left-censorship prevents to draw meaningful empirical survival curves. Of course, one can start playing with raw data to circumvent these difficulties. For example, to

represent the direct increase in mortality at capture, we could subtract proportions computed on captive individuals to those computed on captured individuals, and we could consider a couple of years after capture to increase the sample size and thus reduce the noise in the estimation. However, doing so implies to process the data in a crude way which no longer reflects either raw data or precise prediction from models. Even these technicalities aside, showing raw data separately for each capture method in Figure 2 and 3 would produce too busy images. We hope the new Figure 1 with raw data by age is therefore sufficient for these reasons.

J.M. Gaillard

Reviewer #2 (Remarks to the Author)

We wish to thank you for all your very helpful comments, which we have extensively used to thoroughly revise this ms. We address your first set of comments here (see below). Please see other attachment “Response to reviewer 2 manuscript comments” for addressing the comments which were included in the manuscript pdf-version.

1. The paper presents unique and highly valuable data based on individual records for working elephants from Myanmar.
2. The authors used the data to address important and critical questions about differences in survival between captive-born and wild-caught Asian elephants. They proposed to use this data set to investigate whether:
 - a. wild-captured individuals have compromised survival at any age compared to captive-born individuals
 - b. the effect of capture on survival depends on the age at capture,
 - c. the effect of capture on survival depends on the time since capture
 - d. there is variation in survival based on the capture method used
 - e. there is a sex difference in the effects of capture on survival.
3. The methods and statistical analysis used are sound and indeed the authors report very important and original findings demonstrating that survival is significantly different between these groups of elephants, with lower survival for wild-caught males, females, etc.
4. I agree with the authors that their findings demonstrate that the practice of capturing elephants for sustaining captive populations is not only detrimental because it reduces wild elephant populations. It also doesn't provide a viable solution for sustaining captive populations.

Thank you, we agree on all points and especially on elephant capture not being a viable solution to sustain captive populations.

5. The main problem I see with this work is that the authors link survival back to the capture and capture method, rather than to the substantial trauma and injuries wild-caught elephants experience during the breaking and taming period or the differences in management of wild-

caught and captive-born elephants. For example:

- a. Wild-caught elephants suffer much longer and more traumatic breaking and taming periods. Practices include:
 1. long periods of confinement to crushes and cradles;
 2. animals often are tied at their feet and forced into very painful physical positions. Ropes used to tie them cause deep and festering wounds on all four extremities which often get infected. The scars from these wounds remain for life.
 3. periods of starvation alternated with offering food and water;
 4. shouting at the animal, beating, poking with sharpened sticks, and many other forms of mistreatment.

Thank you for these illuminating examples, we fully agree with you that differences between wild-caught and captive-born animals result from a whole host of differences starting from capture and accumulating subsequently e.g. due to harsher taming methods used, and this was not clear enough in the original submission. This does not reduce the value or implications of our findings that the survival of wild-caught and captive-born animals differ, and we have now said more clearly in the whole manuscript that the breaking and taming are likely much harsher for wild-caught elephants than captive-born elephants and that the management can differ in some cases for wild-born and captive-born animals. See for example Introduction, lines 62-64 and Methods, lines 441-444. We have also checked our wording throughout the Manuscript to ensure we address your concern and use clear language. We still wish to stress that our study on the differences between wild-caught and captive-born elephants is the best which has been conducted so far in Asian elephants as both wild-born and captive-born elephants are subjected to many similar management practices in the Myanmar population and share exactly the same environment, diet, disease load and exercise possibilities irrespective of their birth-origin (see lines 119-130).

- b. Wild-caught animals generally do not adapt as well to captivity and logging work as captive-born. As a consequence, they tend to be punished more and often are treated more harshly than captive-born animals.

Thanks for pointing this out, we now acknowledge this point too in the manuscript, see page 4, lines 60-64.

6. Similarly, the authors assume that differences in survival are directly related to the capture method (e.g. immobilization, stockade, and melashika) and do not consider that many of the observed differences may have to do with differences in a) immediate capture mortality which may result in selection the healthiest and most adaptable elephants via stockade; b) the population and individuals targeted by difference techniques; and c) potential for pre-capture trauma and injury from human-elephant conflict suffered by individuals captured via immobilization. The authors mention some of these factors in the discussions but because the hypotheses, methods and results are set up to test for capture rather than management, these arguments don't stand out and are not explained or discussed sufficiently. This may have long-lasting impacts on future elephant management recommendations.

We have now mentioned points a and b straight in the beginning of the introduction (lines 64-66) and discuss all these three points in the discussion, lines 323-336. About point c, many elephants which have been captured after 90s have indeed been captured to mitigate human

elephant conflict (HEC), and we now say that, see lines 110-112. However, there is no certainty that all problem elephants in Myanmar would have been captured by immobilization as Leimgruber et al. 2011 (Current Status of Asian Elephants in Myanmar, Gajah 35: 76-86) says that majority of problem elephants were captured by Kheddah (stockade) in at least 2003/2004 in one township in Myanmar (page 79: “To reduce people-elephant conflict, MTE captured 41 elephants in 2003/2004 in this township (36 by Keddah and 5 via immobilization)” Also, after rerunning all the models following the suggestion of Referee 1, we do not detect big differences between the capture methods on mortality anymore and therefore decided not to go very deep into discussing about each capture method in the manuscript.

For example, the uninitiated reader will assume that immobilization is much worse than stockade as a capture technique. However, stockade has a much, much higher immediate mortality, much of which probably is never reported. Stockade will result in the increased capture of older individuals as well as in the capture of more bull elephants because it is unselective. However, because mortality is so high, stockade capture may select for the hardiest and most adaptable animals to be transferred into captivity, the others likely perish during the process.

We agree with these points and now discuss selective sampling and mortality following capture in the discussion, lines 323-336 and say also that because of these reasons it is challenging to predict capture effects or especially compare the capture methods. We also give estimates on capture-related immediate mortality in many points of the manuscript for the three different methods (lines 107-110 & 199-202) and explain how the three methods differ in which elephants are being targeted (lines 199-204 & 326-329). In any case, the results from our new statistical analysis do not support large differences between capture methods.

Melashika likely is the “softest” capture technique (as the authors rightly report), because it specifically targets young and docile animals. Older and more rambunctious animals are nearly impossible to catch with this technique. These elephants will be easier to break and be less likely to suffer long-term injuries as a result.

Yes, we agree and report these points, see lines 203-204, 326-329 and 422-426.

Immobilization also is very specific but it allows for the capture and removal of “problem” elephants that are engaged in crop raiding and other types of human-elephant conflict. Removal of these “problem” elephants and transfer to captivity is a relatively common practice in Myanmar. Many of these animals may already be severely stressed, traumatized, and injured prior to the capture, because in crop-raiding conflicts local people throw rocks, torches, large china crackers, at these elephants.

Thank you, we say that capture has targeted specific elephants to mitigate human elephant conflict (HEC), see lines 110-112. However, our new results show that immobilization did not differ substantially from other capture methods in terms of long-term survival.

While it is possible that the drugs used for immobilization have long-term consequences for survival, it seems much more likely that pre-capture trauma and injury, breaking and taming, and differential treatment in working camps play a significant role in reduced survival of

these elephants. Controlled study of the use of immobilization drugs on elephants in captivity and the wild should be possible given the number of animals that are treated this way. Also, one would think that dosage and application are important for these considerations.

We agree that further studies are needed for drugs used for immobilization (lines 253-256) and we hope our findings will inspire such studies.

7. It seems that males may have higher mortality because they generally perform the most difficult and dangerous work, pushing and dragging logs in extreme terrain. Consequently they have a much higher risk for injury and mortality. Additionally, I believe pregnant females and young mothers have reduced workloads and are either not used at all or are used to transport lighter loads.

We agree. In general, as we show now (Figure 1), males have higher mortality through all ages in our study population. Some of these differences during working ages are potentially because of the reasons you mentioned, but there is a sex difference present already among newborn and calves under taming age which do not work and live in comparative freedom. Similarly, a sex difference prevails after retirement age. Such general differences between the sexes are in line with what is being documented for most mammals in the wild, and also for humans.

Immobilization/capture are critical tools for elephant management, specifically for capturing and treating injured elephants (e.g. elephants in Asian frequently get injured by wildlife snares), relocation of problem elephants, and satellite tracking to better study their ecology. Some of these techniques are also critical in anti-poaching activities. It will be critical to better understand whether immobilization has significant negative effects on these wild elephants and to balance them with the potential benefits from the immobilization. Thus, it is important to sufficiently discuss the findings on immobilization from this research and ensure that the findings are not the result of other effects that were not sufficiently measured or described.

Thank you, we now say in the discussion that there are some conditions when the capture is necessary for elephants, see lines 370-372, but more in-depth analysis on the need for capturing elephants or very detailed discussion on each capture method in the light of our new results is beyond the scope of this study and not possible due to strict length limits imposed by the journal.

8. Wild-capture of elephants is a huge conservation problem and likely has led to significant declines in wild elephant populations. The authors acknowledge this but it would be useful to see more discussion of this topic. Much of this practice was meant to supplement captive populations that are not self-sustained. The results in this paper shows that such practices are very wasteful because of a) the mortality associated with capture and b) the decreased survival of these wild-caught elephants in captivity.

We agree and now also emphasize in lines 253-256 & 352-356 that both short-term (capture-related and/or taming-related) as well as longer-term effects of capture should be taken into account in further studies. We also say that elephant capture is likely to be an unstable strategy and alternative methods should be sought to boost captive populations (lines 376-379, see also 256-258).

Reviewer #3 (Remarks to the Author):

This is a very interesting (and disturbing) paper, based on an unusual dataset and using sophisticated statistical methods.

Our two main critical comments are to do with the context and modern-day relevance of the work. First it sounds as though the practice of catching Asian elephants from the wild in Myanmar ceased by 2000, and although later the authors attempt to generalize from this animal to other species it is not clear to which such data can really be extrapolated, partly as elephant capture methods seem unusually harsh because of these animals' great size, and also because capture from the wild is being phased out by some bodies (for example, the catching of wild animals for research in Europe). Can the authors be more precise about specific, current practices their data apply to? Are these data truly relevant today? If so, in which countries and to which species?

Thank you for your helpful comments. We welcome the opportunity to expand on this important topic. First, although legislation is now more strict and the wild-capture of certain species has recently been reduced by some parties and purposes such as for research animals in Europe, it still applies to many species even in Europe. In many other parts of the world the wild-capture of numerous species is still extensive, not as controlled and for example the majority of wild-life trade concerns wild-caught animals (e.g. Nijman 2010 Biodiver Conserv 19:1101-1114). Furthermore, a significant proportion of species such as elephants in captive facilities continue to be wild-caught and wild-caught animals continue to be used in research (although in decreasing numbers), in pet trade and farming operations. All these activities can have a critical impact on the population sizes in the wild (Mason Anim Behav 2013).

Second, wild elephants are still today captured extensively (legally or illegally) (e.g. Mikota, S. et al. Sumatran elephants in crisis: time for change, 361–380 in Elephants and ethics: toward a morality of coexistence (eds Wemmer, C. & Christen, C, 2008)) and wild-capture is known to impose a very negative pressure on Asian elephant population growth rates (Leimgruber et al. 2008). Capture continues in Myanmar, too, although it usually focuses on elephants involved in human- elephant conflict (lines 110-112). Similarly, translocation of elephants, requiring animal capture, continues for diverse reasons (e.g. for veterinary or conservation activities) in both Asia and Africa (e.g. Fernando et al. (2012). Problem-Elephant Translocation: Translocating the Problem and the Elephant? PLoS ONE, 7(12), e50917; Pinter-Wollman et al. Assessing translocation outcome: Comparing behavioral and physiological aspects of translocated and resident African elephants (Loxodonta africana). Biol. Cons. 142, 1116–1124 (2009)).

We have revised our ms based on your advice to now say more clearly that elephant capture is still evident in many places and concerns both Asian and African elephants, and more generally, that other species are also being captured in significant numbers each year (see lines 36-37, 68-69, 243-245 and 347-349).

As long-term consequences of capture have only rarely been studied in any animals or endangered populations and animals with slow life-history are shown to suffer most from captivity (Tidière, M. et al. Sci. Rep. 6:36361, 2016) and might also be more prone to negative effects of capture, we think this study is very much needed and our results call for further studies on different species and responses to capture (immediate and long-term).

Second, we already know of substantial birth origin effects on elephant survivorship (not least the highly adverse effects of being captive-bred in zoos), and of effects of capture from the wild that elevate mortality for some years: such results feature in two papers in which the lead author is an author (Clubb et al. 2008, their ref. 25; also Clubb et al. 2009 in *Animal Welfare* which discusses potential mechanisms for such effects; plus also Dr. Mar's PhD thesis). So the Introduction as written is quite strange as it makes this paper look totally novel, and as though the authors are writing on a blank slate (which isn't so!) Please make the Introduction more accurately, transparently representative of the current state of knowledge of birth origin effects, including what we already know about elephants. Then what this particular study specifically adds that is novel should be spelled out more clearly.

Thank you for pointing out this unintentional omission in our writing. We have now tried to be more specific about the novelty points of this manuscript compared to previous research on capture effects on survival in elephants (see Introduction and lines 82-92). Both of the Clubb et al. papers focused on females only and did not compare the male survival in relation to being captive- or wild-born. Neither of these previous papers analysed whether the age at capture and capture method had an effect on the mortality pattern and whether the time since capture effect depended on these variables in same environment in both sexes. We now also mention previous studies in the discussion, lines 303-307. In addition we now explain how our approach differs from birth origin effects in captivity, lines 69-80. We also discuss about zoo elephant survivorship in Discussion, lines 356-363. We hope these clarifications now illustrate how our analysis adds importantly to the existing literature.

Minor comments:

The first few paragraphs of the Discussion currently read more like an introduction than a wrap up of findings. If more of the details presented here were instead in the Introduction – with emphasis on the topics highlighted above - that would give the reader a better sense of why the authors did this work.

We hope our re-write of the Introduction based on your comment above, as well as modifications to the Discussion, now better illustrate the relevance and novelty points of our study compared to previous research.

Relatedly, why is there historical information in the Methods section? Since Methods come after the Discussion, there needs to be more of this background -- e.g on the capture methods -- in the Introduction section instead, otherwise the meaning of the results are not clear.

We included most of the information concerning capture methods in the Methods section because of the length restrictions imposed by Nature Communications on the Introduction (max 1000 words which we already exceed by 275 words). We, too, would prefer to present this information earlier on and we have now followed your advice and moved some materials into the Results section to make the results more understandable. We are happy to further expand our discussion on capture methods already in the Introduction, should the Editor find this acceptable.

Lines 108-109: how can data from captive born animals predate (since starting 1925) data from wild-caught animals (which start 1951)?

We excluded wild-caught individuals captured before 1951 because only limited records were available prior to then. We have now said this in the Methods section, lines 467-470.

Line 440: How was age at capture assessed for wild animals? Crucial if age is to reliably be included in the models.

We agree and have now expanded our explanation on how the age is estimated for wild-born elephants, see lines 402-410. Majority (>70%) of the wild-captured elephants in this study were under age of 20 when captured and the error in these estimates is unknown, but likely to be within a couple of years for young elephants that still grow. While errors could be a problem for demographic forecasting, they do not represent a major limitation in the context of our study. Indeed, looking at our new figures clarifies why errors of a few years would not impede our main conclusions:

- males show a higher mortality rate than females at all ages, so shifts in the x-axis would not impact this finding

- capture increases immediate mortality at all ages, so again shifts in the x-axis would not impact this finding

- error in age estimates could only impact the absolute value of such increase in immediate mortality but not the sign of the slope, and only the latter shows that the impact of capture gets worse with age

- how fast elephants recover from the stress correlating with capture mostly depend on the time since capture, which is a relative duration that would not change if age estimates are wrong.

Line 170-172: “There were however few differences...” this sentence was unclear. What is meant by few differences? Elaborate on the effect of milarshikar on survival.

This sentence has been now removed from the revised version of the manuscript.

Lines 211-213: Suggest checking the stats in these cited papers as some of the cetacean work in particular is not very good...

OK, we have now modified the sentence to express more caution regarding the strength of this mentioned study. We would still like to cite it, because very few other studies exist that are anyhow comparable to ours. See lines 263-266.

Lines 218-220: Not entirely clear why changes in cognitive function are being talked about here. On the other hand if you want an overview of how stress affects morbidity and mortality, this is good and may well be useful:

Walker, M. D., G. Duggan, N. Roulston, A. Van Slack, and G. Mason. Negative affective states and their effects on morbidity, mortality and longevity. *Animal Welfare* 21, no. 4 (2012): 497-509.

Thank you for the useful reference, we have now modified the sentence and cited the Walker et al. article, see lines 279-280.

Line 369: “The taming undoubtedly induces stress and compromises welfare...” how so? What indicates this? Is perhaps in danger of appearing subjective.

We have now clarified the sentence, see lines 431-432.

Lines 458-459: How were the hierarchical models created and what objective model selection criteria were used (e.g. AIC)?

We have now revised the model comparison and discuss explicitly this topic in the methods, starting from line 555.

Figure 2: Make it obvious in the description that WC is wild captured and CB is captive born. Also maybe just state that colors represent sex throughout all the graphs since the color legend is only in A.

Thank you, we have now modified the legends and figures accordingly.

Reviewer #2 Comments in manuscript file

-Title: Better: Differences in age-specific survival between wild-caught and captive-born Asian elephants. The authors are investigating differences in survival between wild-caught and captive-born elephants. Differences in survival in these long-lived animals are more likely caused by differences in management applied to wild-caught vs. captive-born elephants.

We changed it accordingly to “Differences in age-specific mortality between wild-caught and captive-born Asian elephants”.

-Lines 16-17: This seems a bit too sweeping statement. There may be research questions that are not affected by changes in longevity caused by capture trauma.

We mean that there can also be other kind of consequences in addition to increased mortality if long-term effects of capture have not been acknowledged more widely across animals. We have now modified the sentence, see lines 13-15.

-Lines 23-24: There are many other factors that need to be considered. These capture methods are used on different individuals and also have different rates of capture-related mortality, which likely bias the assessments over the long-term.

Thank you, we changed the abstract in light of the new results. There isn't space in the abstract to discuss more in detail the capture methods, but to address your concerns we now discuss the sampling bias and selective appearance/disappearance in other parts of the manuscript, see lines 64-66 and 323-336.

-Line 64: It should be possible to derive indicators for at least some of these parameters from Dr. Khine U Mar's extensive database

We agree that such a study building on the results presented here would be very interesting. The historical mortality dataset analysed in our ms does not contain data on the behavior, physiology or immunity of the several thousand individuals that are now deceased; instead this large demographic dataset includes information on all animals on their life events, births, deaths, capture ages and times, and calving information. See lines 393-410 for more details on the information we have available. We hope our results will inspire more studies on the behavioral, physiological and immunity-related factors underlying the mortality differences reported here.

-Line 65: This needs to be quantified for your data.

We agree that quantifying capture-related injuries would be illuminating, but such individual-level data on the historical animals analysed here is not recorded and available for this or probably any other population worldwide (we say that more clearly now in the manuscript, lines 107-108 & 340-343). Again, hopefully our study will inspire such important research topics.

-Lines 70-71: Even more surprising that the author's did not include some of this information from their extensive database. Rather they reduced their analysis to capture method. Capture

method covers a relatively short time interval compared to "breaking-in", training, post-capture transport, use in logging, and overall management.

We totally agree, however there is no existing individual-specific dataset on the length of the taming/breaking, transport and variation in work-load that concerns the historical, already deceased animals to address these questions. We are somewhat puzzled why this Reviewer thinks so, and we have now carefully checked the ms to prevent further misunderstandings.

-Line 76: But this is not related to capture, but rather management. Another argument why the authors shouldn't reduce their analysis in the way they chose.

We mentioned the zoos to give an example of how captivity affects elephant survival. We did not intend to draw parallels to capture-effects on survival; sorry if it was unclearly said, we have now modified the sentences on lines 71-73.

-Line 82: repetitive word choice/confusing

Changed, lines 77-78.

-Lines 97-98: Reference missing or numbers need to be provided from studbook. Based on simple population models we found that breeding reported from this population would not be sufficient to sustain a population of 5,000 animals and that in fact the number of wild caught animals is under-reported in this database. Given existing problems with elephant registration throughout the region, this needs to be taken into account.

Ok sorry we meant that the current population includes 80% of captive-born elephants but we agree this was unclear in the light of the historical trends and we have now removed the sentence to avoid confusion.

-Lines 101-105: It's apparent that official off-take numbers are under-reporting actual off-take.

We now mention that, see lines 103-106.

and potentially higher as indicated by both reference 31 and 33. Supervision of Keddah capture has been very poor and often was conducted by private elephant owners. Likely off-takes as well as mortalities were higher than recorded in official numbers especially for the stockade method. Additionally, there may be mortality associated with animals that were darted but ran off and could not be located after darting.

Thank you, we changed the sentence, see lines 107-110.

-Lines 107-108: Capture continued well into the 2000s, including keddah operations by private elephant owners.

Yes, we say that smaller-scale capture continued after banning it in 90s (lines 110-112).

-Lines 119-120: Above, based on reference 33, the authors report that more than 2,000 elephants were captured between 1970 and 1993. Presumably the number of wild-caught animals in the database must be much larger than 2,000 (more like 4,000) for the 50 year

period? This also fits better with models reported in reference 35. The question here is: a) where are the missing wild-caught elephants from this database? b) is it possible that a large number of elephants reported as captive born, are in fact wild-caught? c) Does this bias the analysis presented, if there are a lot of missing wild-caught elephants?

Actually we say that we have records on 2000+ elephants that were captured between 1951-2000 (lines 119-122). Our demographic data is based on records maintained by the Myanmar Timber Enterprise on individual elephants as explained in the Methods. It may not include animals which records could not be linked to later-life events; we now acknowledge that capture numbers may be underestimates. In any case, the exact number of captured animals is not critical for the aims of this ms, that focuses on the longitudinal comparison of the life-course of known captive-born and wild-captured animals, NOT on population modelling. We have no reason to suspect that a large number of captive-born animals would be wild-caught; MTE has carefully recorded the origin of each animal as well as detailed maternal (and sometimes paternal) information on all captive-born animals, and our detailed cross-check of these maternal reproductive histories has not revealed inconsistencies that would suggest mis-maternities (wrongly assigning wild-captured animals to captive mothers). Our in-depth analysis of such maternal reproductive histories is published elsewhere (e.g. Hayward et al. 2014 JEB 27, 772–783; Lahdeperä et al. 2014 Front. Zool. 11,54) and shows that the reproductive patterns are consistent with data published on other populations.

-Lines 121-123: This implies that management for the two subsets are identical. This is not true for several reasons: 1. Captive-born elephants are weaned and separated from their mothers at an early age. Their training follows fairly natural. 2. Wild-caught elephants are transferred to a crush where they are behaviorally broken. This is an extremely violent and aggressive form of management that includes periods of starvation, beating the elephants into submission, and confinement to the crush and enclosures for prolonged periods of time. The process is extremely traumatic. It is longer and more violent the older the animal, with adult males rarely surviving. 3. Elephants are used as work and draft animals, with large males required to do the most difficult and dangerous work. I.e. risk of injury is biased towards male and middle-aged animals. 4. Pregnant females are transferred to rest camps where they stay during parts of the pregnancy and early infancy of their calves.

Actually to our knowledge also captive-born elephants have traditionally been transferred to a crush or cradle around age 4-5 to undertake a taming similar to the one described by the Reviewer; prior to this age captive-born calves are not used to human touch or command. We also cite an article about the training methods used for captive-born calves in the manuscript now to clarify the issue (Zaw Min-Oo 2010 Gajah, lines 439-440). We also say in the manuscript that the training is likely harsher for wild-born individuals than captive-born individuals and wild-captured elephants are likely to go through more stressful psychological and physical trauma during capture and taming (lines 441-444). In addition, following your advice, we now mention that older elephants and male elephants take longer to tame than females and younger elephants. See lines 428-431 & 285-287. However, as said also captive-born calves are tamed and it can also be traumatizing and cause injuries (Zaw Min-Oo 2010 Gajah). We also say now that females of both origin are given rest from mid-pregnancy until the calf is one year old and then given light work but allowed to nurse the calf on demand (lines 456-458). Indeed, the capture and initial training aside, in many important ways, the subsequent management of elephants is quite similar for wild-borns and captive-borns, as they both live in the same environment in mixed-groups with similar food, disease exposure and health care available to them, and e.g. females from both groups are allowed to get a

“maternity leave” and same treatment when having “calves-at-heel” (although some individuals may be subjected to differential management because for example behavioural differences, see lines 62-64). Hence, the population offers an exceptional opportunity to compare the long-term survival trends of elephants born in the wild vs. in captivity.

-Lines 125-127: This really covers a very short period of life. I believe the studbooks contain relevant veterinary information on injuries, health conditions of the individual elephants. Additionally it would be useful to know how often elephants were moved, where they worked (e.g. which timber region).

As said previously, we do not have any individual-specific dataset on all individuals on their veterinary treatments, injuries and health as the large demographic dataset only includes information on all animals on their life events, births, deaths, capture ages and times, calving information etc. Also, the same concerns movements between camps. However, we have controlled for the latest living region in all our models. See lines 393-410 and 508-512 for more details.

-Lines 134-138: How was age established in wild-caught elephants? Figure 1 shows that there are significant differences in the subpopulations of wild elephants that are captured by different methods. This may have important effects on mortality:

1. Stockade/keddah method is unspecific, and results in the capture of entire elephant herds/social groups. Generally it results in many more adult elephants captured and mortality rate during capture and breaking in period are extremely high. I think you will find that most experts consider this the most traumatic and damaging form of elephant capture. 2. Both lasso and immobilization techniques are much more targeted and specific. Individual animals are chosen and then pursued until captured. Lasso is most specific in that almost only young animals are captured this way. From a management perspective young animals are most valuable because they can easily be trained, are easier to manage, and have lower mortalities. They also represent the least risk to the capture party. 3. Immobilization is often used to capture problem animals. I.e. if there are individual elephants that cause conflict with people. Older animals that are immobilized mostly because they are problem animals. These animals likely have experienced significant other trauma during crop-raiding conflicts, possibly physical injuries from home made guns and china crackers.

Thank you, we have now expanded our explanation on how the age is estimated for wild-born elephants, see lines 402-410. And we totally agree the capture methods differ in how selective the sample of elephants is (see lines e.g. 64-66, 199-204 and 323-336) and have therefore chosen the statistical models which take into account the elephants (captured with different methods or being captive-born) entering the population at different ages (captive born elephants at birth but wild-captured elephants at age at capture).

-Lines 145-146: capture in the sense that different subpopulations were captured with different techniques. However, this seems overly simplified because how do we know this is not a reflection of differences in management?

We now discuss the management effects in the introduction, lines 60-66 and discussion lines 273-297 and admit these are likely to have an effect on the different mortality between wild-born and captive-born animals. We also say in lines 374-376 that captive-born elephants may be easier to handle than wild-captured elephants.

-Lines 150-151: this makes perfect sense However, the authors fail to explain this adequately.

Reviewer comment concerning sentence: “Wild-caught individuals had a higher immediate mortality risk within the first year of capture, and this immediate risk increased with age in both sexes (Fig. 2a).” We have now increased discussion on this results further, see lines 273-297 in discussion. See also the same paragraph in the discussion for explaining the result that older animals suffer more from capture than younger ones.

-Line 153: It's difficult to determine the exact age of a wild caught animal. How sensitive is this analysis to errors in these estimates?

As we say, the errors in the estimates are likely to be within few years for animals younger than 20 when captured which form the majority of wild-captured animals in the study (>70%, lines 408-410). While errors could be a problem for demographic forecasting, they do not represent a major limitation in the context of our study. Indeed, looking at our new figures makes it clear why errors of a few years would not impede our main conclusions:

- males show a higher mortality rate than females at all ages, so shifts in the x-axis would not impact this finding

- capture increases immediate mortality at all ages, so again shifts in the x-axis would not impact this finding

- error in age estimates could only impact the absolute value of such increase in immediate mortality but not the sign of the slope, and only the latter shows that the impact of capture gets worse with age

- how fast elephants recover from the stress correlating with capture mostly depend on the time since capture, which is a relative duration that would not change if age estimates are wrong.

-Lines 154-155: It's very strange that immediate mortality is higher for females than for males. Observationally, males resist training and breaking in much more than females and tend to receive more injuries.

This result is no longer present in the light of our revised analysis.

-Lines 178-180: Is this different from the previous way to predict survival rate? This needs more explanation?

These predictions are indeed different from the previous ones. We have now clarified that in the legend of Figure 4 and in the text (lines 227-231). The difference is that here we do not assume all elephants to be born in the same location and in the same birth cohort. Instead, we build for each observation the prediction corresponding to the characteristics of the elephant behind that observation. Then those prediction are being averaged among the elephant of the same age and of the same capture status.

-Line 208: Asian

Word “timber” replaced by “Asian” as suggested, line 245-247.

-Lines 211-213: This is an overly general statement. Can you please be specific what the benefits are to welfare specialists, veterinarians, and ecologists? Are you saying that

immobilization through sedation increases mortality risk? Even this would be an overly general statement, unless you can provide more detail on the drugs and dosages being used.

We have hopefully stated now more clearly in the manuscript that there are many factors to consider when comparing different capture methods. Our take-home message is that the capture (and subsequent training etc.) clearly increases elephant mortality and this is not a sustainable tactic to supplement captive populations world-wide for this endangered animal and long-term effects of capture should be studied also in other animals (which has been only rarely done so far). The welfare specialists, veterinarians and ecologists will benefit from knowing of these adverse effects on elephants by e.g. allowing them to avoid unnecessary capture and trauma for elephants, forecasting how wild-capture affects captive and wild population sizes, potentially planning more elephant-friendly capture methods (when it cannot be avoided) and improving management practices so as to best support captured animals during the most critical periods after capture (see also lines 249-253). These results also call for further studies on capture methods e.g. how certain drugs or dosages affect different animals short- and long-term) or specific studies on taming/training-related mortality (or injuries) of each method (see lines 253-256).

Lines 218-219: -This is only true if one accepts your statement that management and treatment of wild-caught vs. captive-born elephants in these camps is identical. Considering the significant differences in breaking in and training, as well as the possible history of traumatic conflict with people prior to capture, this cannot be assured.

Reviewer comment concerning sentence: “Elephants captured from the wild had higher mortality rates than captive-born elephants at all ages, and such effects were more detrimental if the elephant was older at the time of capture”. We are now saying more clearly that the capture effects as whole might be partly due to differences in breaking, taming and subsequent training. However, in general wild-captured elephants and captive-born elephants are treated similarly as they share the same environment, work-loads, disease sources, breeding patterns and social interactions. They are also kept in mixed groups of both wild-born and captive-born elephants.

-Line 227: Here you are omitting the fact that timber elephants sometimes are moved around over long distances. This can include transport by truck or boat.

Reviewer comment concerning sentence: “Moreover, inter-zoo transfers reduce Asian elephant survivorship, an effect lasting up to four years after the initial transfer”. We agree, however, the same concerns both captive-born and wild-captured elephants.

-Lines 231-232: You are completely omitting the fact that these elephants are first broken by being tied into a crush, starved and beaten.

We now discuss these effects in the whole manuscript, see e.g. lines 273-297 in discussion.

-Line 267: This statement completely ignores the fact that capture-related mortality is 3-4 times higher for the stockade method (and is probably severely underreported).

Reviewer comment on sentence “Although there is variation in the effects of the three capture methods on mortality depending on the animal age at capture as well as the time elapsed since capture, in general, immobilization appeared to be the most damaging capture method

in both sexes.” We agree, however our study does not focus on mortality during capture operations, but rather on the subsequent survival patterns of the captured elephants kept in the same logging camps with captive-born counterparts (see lines 107-110). Our main message of capture methods is that all capture methods are detrimental either in the short- or long-term (or both), and capturing wild elephants thus poses a substantial risk on their survival across several years (lines 343-345).

-Lines 268-269: This is possible. Were these injuries not recorded in the books? Can you provide numbers how often this happens?

As said already previously, we do not have records available of the injuries or mortality during the capture operation, mentioned on lines 107-110.

-Lines 270-271: This observation may lend support to the fact that differences in work load, types or work performed, and rest periods, may be more important in driving mortality differences between males and females.

The sex-specific mortality is a general mammalian pattern and holds even at ages when elephants are not engaged in work activities, see our reply to previous comment.

-Lines 274-276: The real question here is: Whether is what are the differential contributions of a) pre-capture traumatic experience and injuries from human-elephant conflict. b) capture related mortality, injury, and risk; c) post-capture injuries and trauma from breaking in and training.

We cannot separate these effects in our analyses but acknowledge all these points in the revised manuscript. Our analyses focus on comparisons during post-capture period and show that capture increased mortality also beyond the known immediate increase in mortality associated with the capture operation itself (lines 169-170).

-Lines 278-279: Please also discuss how different capture methods may result in the capture of individuals that are very different in terms of pre-existing trauma, injury, adaptability to captivity, and general health.

Indeed, we discuss these points in the manuscript, see e.g. lines 62-66, 273-276, 323-326 and methods where we explain how each method selects captured elephants.

-Lines 347-349: Can you provide some kind of sensitivity analysis? How much would your results change if wild-caught elephants are consistently aged older or younger?

This is unlikely to alter our conclusion as argued above.

-Line 363: It's my understanding that the drives are unspecific? I.e. all elephants in an area are driven. The resulting groups may not necessarily be family units.

Thank you, changed accordingly on lines 412-413.

-Line 364: How about crushes?

We talk about using crushes in the next paragraph, starting from line 428.

- Line 364: Can you provide more detail on taming procedure.

We now talk about taming in the next paragraph, beginning from line 428. See also Zaw Min-Oo 2010 Gajah 58-61 for further information on training methods, which we now cite in the manuscript.

-Line 385: This regularly results in horrendous festering wounds, with scars that are visible for the rest of their lives.

Yes, we agree and say in the same paragraph “Wild-captured elephants are likely to go through stressful psychological and physical trauma during capture/taming... and training of captive-born elephants is likely to be less stressfull/harmful”, lines 441-444.

Reviewers' comments:

Reviewer #1 (Remarks to the Author):

This revised version nicely accounts for most problems I identified when reading the previous draft. In particular, I really enjoyed the new survival analysis based on a generalization of the Siler model. That being said, I still have several minor concerns, mostly about the presentation of the new statistical analyses:

l. 212: « lowest » instead of « best »

l. 212-213: Unclear what you measured the « probability of model selection ». A delta-AIC (not DAIC as stated) of 0.9 should lead to retain the model with the lowest number of parameters based on the principle of parsimony.

l. 307: Remove « fits »

l. 501-503: The exact differences between the Siler model and the model used here should be made explicit. The Siler model only has 3 « w » terms and it is unclear to me why the third « w » term and not the second « w » term was assumed to be constant. The simple extrapolation of the Siler model should have considered both w_2 and w_3 as constant terms. There is nothing wrong with the model but a better justification is required.

l. 504: This statement is not correct. The Siler model includes 5 parameters and should correspond to:

$$p = w_1 \cdot e^{-b_1 \cdot \text{age}} + w_2 + w_3 \cdot e^{b_2 \cdot \text{age}}$$

This is quite different from what it is reported in the first line of the equation!

l. 506-507: In the Siler model, the second term is a constant mortality of prime-aged adults and the third term describes the mortality increase with age, not the reverse.

l. 511: This statement is not correct. There is no need to assume a null mortality at any age. The Gompertz model provides a model of age-specific changes from a specific age (usually assumed to be the age at first reproduction) so that immature individuals are excluded and there is no prime-age stage with a constant mortality rate.

l. 515: « age-specific changes of mortality rate », not « baseline »

l. 530: « describes » instead of « describe »

l. 531-532: Rewrite the sentence.

l. 544-545: Provide the exact package and the reference

l. 548-550: Provide a reference to justify this metric.

560-563: Estimates without associated errors are not very useful. That getting confidence intervals would be computer intensive is a poor justification!

l. 565: The AIC does not allow estimating the « predictive power » of a model.

l. 570: Remove « baseline »

l. 576: « effects » instead of « effect »

l. 584: « model » instead of « models »

l. 592: « chose » instead of « choose »

l. 604-605: This statement is far too vague. Instead, all the code used and data analyzed in that work should be made available to readers.

l. 797: Remove « Baseline »

l. 841: « the best among the candidate models fitted » instead of « the best »

J.M. Gaillard

Reviewer #3 (Remarks to the Author):

This paper is much improved, but there are still two problems with the way it is framed. First, the authors exaggerate how relevant their data and findings are to other types of wild-caught animal; and second, they do not properly discuss how these data and findings relate to translocated and zoo elephants.

When Asian elephants are caught to work in the logging industry, the capture and move to captivity are then followed by several weeks of intense and harsh taming/breaking/training. The authors do acknowledge this, and one of the other referees highlights it too. However, the authors downplay this in the language they use (referring only to the effects of ‘capture’ throughout their paper), and also in the over-zealous way they claim that their results apply to the members of other species caught from the wild (in the Abstract, Introduction and Discussion). The authors repeat these claims about generality to other species several times (e.g. twice in the Abstract alone), and yet are necessarily vague with them too, as it’s really not clear that any other types of wild animal undergo anything like this (do they? If yes be specific). So, we would request two types of editing to acknowledge the uniqueness of the ‘Asian-elephant-captured-to-work’ scenario. First, please only suggest that these results apply to other species once and once only (not repeatedly), and also make it clear that this is just a hypothesis. Second, please use the phrase ‘capture and taming’ (or similar) in the Abstract, Discussion and all sub-titles (e.g. at lines 164 and 194) throughout the paper, as it is clear that the effects of capture per se cannot be parsed out from those of taming/training, and so the two must be presented as a package (such that calling these ‘capture effects’ is simplistic and potentially misleading).

Second, how the authors compare and relate these to other elephant populations still needs some work. It’s these comparisons that are most interesting and relevant, and yet the authors do not do them justice. Other elephants are not even mentioned in the opening paragraph of the Introduction for example!

Thus in terms of the wild-capture of Asian elephants that still happens to this day (e.g. lines 112-113), just how relevant are these data: are today’s wild-caught animals still subject to the same

breaking and taming processes? Please make it clear. And in terms of the cessation of capture for logging in the 90's/2000, was a sense that wild-caught animals did not fare well part of the reason for the decision? Turning to translocated animals, again are the authors' data really relevant to them, since translocated elephants again are not subject to training and taming? Make it clear. Lastly, comparing these findings to those from zoo populations (and the reference populations used to assess zoo performance), it's here that the authors are perhaps weakest, as will be detailed next.

The previous work comparing Asians in zoos with MTE animals (Clubb et al. 2008) found that wild-caught MTE elephants have elevated risks of mortality throughout their juvenile years (1-9 years of age), compared to both wild-caught zoo animals and captive-bred MTE animals, but such effects were not evident by adulthood (10-70 years of age): an apparent difference to the effects reported here. Clubb et al. 2008 also found that when animals were transferred between zoos, mortality risks were elevated for 4 years post-move, but after this period such an adverse effect could no longer be detected (obviously pretty relevant to this paper). Thus the authors' implication in their Introduction that their study is the first to follow animals for decades really does need some damping down. Furthermore, that "the early maternal environment of those born in captivity is typically different from wild-captured animals" (lines 76-78) is also not special to zoos, but applies to MTE animals too (who are unlikely to be living in true matrilineal groups).

In addition, the Clubb et al. 2008 paper (built on further in 2009) found that zoo-bred Asian elephants had much higher mortality risks than wild-caught Asian elephants also housed in zoos. This is obviously completely the opposite to what Clubb et al. 2008 and this manuscript find for MTE animals, and it seems odd to downplay this remarkable contrast with a mere "(vice versa in zoos)" cryptic clause, in parentheses, in the Introduction (line 84). A fuller acknowledgement of this paradox is surely warranted. Why the zoo and MTE population differ in the direction of their birth origin effect is raised at lines 366-368, but almost as a rhetorical question with no credible suggestions offered as to potential explanations. I think the authors can do better than this, especially as the lead author was a co-author of the Clubb et al. 2009 paper that identified heavy infant birthweights (perhaps indicating neonatal obesity) as a significant difference between zoo and MTE captive-bred calves, and that also then suggested the high nutritional plane of zoo elephants as a possible cause of the poor survival of captive-bred elephants in zoo populations. This idea is surely still a plausible, testable one, no?

Minor comments:

The phrase "before and after capture" (line 85) also needs toning down too, since the authors do not assess wild Asian elephants before capture and so should not imply that this is a unique angle of their paper!

Lines 99-100 say that around 15,000 Asian elephants live in captivity, but lines 371-372 say about 1000 are: needs editing to clarify so not inadvertently confusing (I think the latter applies to non-timber animals only?).

Figure 1: make it clear that these are data for all MTE animals?

Reviewer #4 (Remarks to the Author):

The manuscript summarizes survivorship using an exceptional data set on Asian elephants. Insights derived regarding differences between mortality in captive born and wild capture approaches are valuable and interesting. Writing is awkward in places, and close editing is needed. The analysis approach is interesting, but I have a few concerns I raise on the modeling approach and the lack of ability to derive CI on estimated trends. This can be alleviated by presenting the raw data for each of the relationships inferred through the model. I would also present survivorship curves using cox regression as an alternative modeling approach that allows incorporation of covariates and will enable presentation of uncertainty with outputs.

Intro

Line 68: the authors conflate translocation with wild capture in their references and framing of the motivation for the paper. Looking at translocation impacts and survivorship is substantially different given that the mortality occurs only during initial capture and transport. If I am correct, that period was not analyzed in the presented data. As such, I would suggest separating these two in the framing of the manuscript.

Line 122: second clause appears to refer to males and females, but not explicitly stated

Line 122-130 is a summary of data and methods – this is not typically included in the intro.

Line 130-137 provides a results and concluding statement not typically in intro – This paragraph reads more like material that would normally be presented in the abstract.

Results:

Line 155-162: Why parameterize birth cohort as 0 mortality initially? What is sensitivity of this parameterization – i.e., could it drive result of differences between captive and wild? See comments on methods about concerns with approach

Line 184: Is recovering the correct term? Seems like you are describing filtering of individuals not able to handle captivity? The model outputs seem to be homogenized by approach. It would be more valuable to see raw data here rather than model estimates without any error estimates.

Discussion

Line 256: This is very specific to elephant context. I think the intro and discussion attempts to

generalize these results to broadly. Elephants are such a unique species in many respects, that I am not sure how generalizable these results are to other species that are not similar in size, feeding ecology and requirements and social structure.

Line 266: detrimental may not be correct word

Line 298-302: The comparison is problematic. Social and demographic costs may differ across age classes, such that one manifests for one group more than others. Ultimately, mortality is a powerful measure of costs, but young may be impacted in other ways.

Line 329: This is a critical point. It is likely capture and successive mortality is selecting for different individual characteristics– acting as a type of filter such that only those that persist can survive in captivity. The captive born have already been selected for traits conducive to captivity given their mothers survived and successfully breed.

Line 337: Not sure robust is the correct word – really looking at traits conducive to captivity

Line 343: can you be more explicit about confounding variables. There is much more heterogeneity in caretaking than presented here. Certainly, all handlers are not the same and this can impact stress, the breaking process varies, ect.

Methods

The first two paragraphs are narratives that could be greatly condensed or removed – simply refer to this historical context.

Line 396: What is meant by natural – impression from wild data suggest breeding of captive may be depressed in general, so not equal to natural rates

Line 415: Aging from dentition is more accurate and could be accomplished for survivors.

Line 437: The “breaking” time would be a really valuable covariate to include in the models looking at different capture methods. Concern remains about the influence of initial capture conditions on longer term survival.

Line 447: remove “also”

Line 454: what is meant by role as subordinates? To people?

Model averaging has been controversial in the literature of late. I could not find the model covariate estimates and CI for the top model and closely ranked models in the SI. These should be presented in the SI. It was not clear to me if the capture technique was informative on mortality, though the authors have worked to include it (seeing raw survivorship and covariate estimates would be more informative).

The approach used is an interesting analytical framework, apparently used in human literature. Coming from the wildlife conservation disciplines, I am not familiar with the approach. My concern is that it appears the authors were unable to derive error estimates on their mortality estimates. If my interpretation is wrong, it would be valuable to have the error presented in the figure (Fig 1, 2 and 4). If not, I wondered if using a Cox regression approach would provide more direct inference? This would allow the same analytical framework – regression with covariates – but provide insight to uncertainty. If Cox regression is used, I would suggest building different models for the different capture approaches and comparing survival directly.

While I am not familiar with the approach used, I am concerned that the model structure, i.e., the parameterization of the three lifestages, results in a homogenization of the mortality rates across subgroups. The assumptions required for this three stage model parameterization should be assessed in a sensitivity analysis (i.e., if you do not assume 0 infant mortality as the baseline, do results shift?). I would prefer to contrast survival curves for each group directly, with error on the survival estimates. However, I realize the authors have already put substantial work into this analysis and approach and there may be perfectly solid arguments to proceed with the approach employed. At a minimum, I would suggest presenting survivorship curves for the capture methods and captive born individuals.

Fig. 1, 2, and 4: Could you present the raw data rather than yearly average mortality rates and model predictions? This would offer better insight to the variability in the data.

Supplementary material: I could not find a supplementary material file besides the two included tables. The lack of information makes it hard to understand the two tables. I assume table 1 is the parameter fit for the models, but CIs are missing. I could not understand table 2.

Several reviewers ask to see the raw data used in the analyses. The authors have only presented the data in Fig. 1. It is important to have the raw data in the other figures, particularly given the authors were not able to estimate confidence intervals in the presented predictions. Otherwise, readers cannot get an idea of uncertainty in the results presented.

Discussion of difference in captive and wild caught breaking does not explicitly state that wild-caught tend to go through a much rougher process. It would be valuable to mention some of the mechanisms causing the higher probability of complications for wild-caught individuals. These certainly could relate to some of the sex differences recorded, even if there is a biological difference in survival between the sexes that is unrelated to capture. The reduction of the emphasis on capture methods given the results and lack of covariate data is sufficient.

Reviewer #3 (Remarks to the Author) and responses back

The two main issues still remain: the authors still seem to claim in some places (but without evidence) that their results can be applied to other wild caught species. In response to our comments about this, they really haven't given any specific/justified examples of which other species/circumstances where these results are relevant. The authors also continue to state that they assessed the effects of capture, rather than capture *and* taming/breaking. Thus even though they acknowledge the possible influence of taming [see their response to another referee for example], their interpretation still revolves around 'capture' effects ... presumably to support their claim that these data are relevant beyond elephants.

When Asian elephants are caught to work in the logging industry, the capture and move to captivity are then followed by several weeks of intense and harsh taming/breaking/training. The authors do acknowledge this, and one of the other referees highlights it too. However, the authors downplay this in the language they use (referring only to the effects of 'capture' throughout their paper), and also in the over-zealous way they claim that their results apply to the members of other species caught from the wild (in the Abstract, Introduction and Discussion). The authors repeat these claims about generality to other species several times (e.g. twice in the Abstract alone), and yet are necessarily vague with them too, as it's really not clear that any other types of wild animal undergo anything like this (do they? If yes be specific). So, we would request two types of editing to acknowledge the uniqueness of the 'Asian-elephant-captured-to-work' scenario. First, please only suggest that these results apply to other species once and once only (not repeatedly), and also make it clear that this is just a hypothesis. Second, please use the phrase 'capture and taming' (or similar) in the Abstract, Discussion and all sub-titles (e.g. at lines 164 and 194) throughout the paper, as it is clear that the effects of capture per se cannot be parsed out from those of taming/training, and so the two must be presented as a package (such that calling these 'capture effects' is simplistic and potentially misleading).

While we agree with many suggestions by the reviewer, it is important to recall that captive-born elephants also undergo taming with the same aim and objective as the wild-caught elephants: to obey human commands and work efficiently in timber industry. Captive-born elephants are tamed around age 4-5 years generally using the same personnel, infrastructure and methods as is used for the wild-caught elephants. Before that, calves are not used to human-handling, commands or caretaking and could as well be considered as being "wild". This fact makes our study so unique, and also means that the alternative terminology proposed by the reviewer is not suited to our study design. It is true that the taming and breaking are harsher for wild-born individuals to a degree that also depends on the age, sex and personality. We say this explicitly in the paper (lines 133-137, 462-465, 475-478) and cite an article (line 465) which

gives details about the taming methods used for captive-born elephants: Zaw Min Oo 2010 The training methods used in Myanma Timber Enterprise 2010 Gajah 33:58-61). It is noteworthy that the taming can also be harsh for the captive-born calves, regularly leading to accidental deaths as recorded by the stated causes of death (see lines 311-314 and citation to Mar et al. 2012 PlosOne 7(3):e32335). Also, due to the intense training, the baby elephants may get wounds, injuries, and stress. We say on lines 60-64 (second paragraph of the whole article) that taming/breaking can have effects on post- capture lifespan and there may be differences in the management of captive-born and wild- born animals. We also say now in the end of the introduction, lines 137-139, that part of the effects seen in wild-caught elephants and captive-born elephants may be due to differences during taming period and from now on we refer to all these effects as capture effects. Just to be sure, we repeat this in each section (Results 182-183, Discussion 297-300,309-314, Methods 462-465, 475-478). For all these reasons and for keeping the wording as light and lively as possible we prefer to stick to the term “capture” instead of using the “capture and taming” which we find more misleading than our original formulation.

It is very good that authors acknowledge the potential confounds between capture and taming/breaking. But it is problematic that they then still present the results as though they were truly able to separate the capture and taming effects in their current study. Whether intentional or not, this implies that capture plays a bigger role than taming and the other acknowledged factors, and there is no evidence to support this.

(And lines 348-350 appear to be the only area of the MS where the effect of capture is separated from the effects of training and captivity, but the authors call it “partial evidence” and “weak”, which further makes it seem poorly justified to use “capture” and “capture effects” throughout).

Also, be careful about implying that the process is similar between wild-caught and captive-born animals; as here is a quote from one of the authors a few years ago (Mar, Khyne U. "Birth sex ratio and determinants of fecundity in female timber elephants of Myanmar." *Gajah* 38, 2013): 8-18):

“Wild-caught females take some years to recover from the stress of capture and/or taming so that their reported age of first calving in captivity is older than captive-born females. For captive-born calves, breaking normally takes only a few days, whereas the breaking period for wild-caught animals lasts a minimum of 2 weeks to a maximum of 8 weeks depending on temperament. The longer the taming process, the harsher the punishments and more unpleasant the process”

So, we repeat: please use the phrase ‘capture and intensive taming’ (or similar, e.g. ‘capture and breaking’) in the Abstract, and in all sub-titles, when referring

to what happens to wild-caught animals. This will keep things clear and honest, while not impeding the flow of the writing.

We are somewhat puzzled why the Reviewer has the perception that we claim the results to apply to other species as such. We have carefully checked the ms and have found no evidence of such claims. Nevertheless, to settle this argument, we have now aimed to be more accurate with the sentences saying how our results apply to other species. In the abstract we do not claim that the results apply to the members of other species caught from the wild (and did not say previously even one time), instead we say that our results are timely because elephants as well as many other animal species are still being captured to supplement captive populations even though there have been alarming declines in wild populations globally. In the introduction, we did not repeatedly make this claim, but we said in the end of the introduction that “The long-term differences between captive-born and wild-captured animals shown by our study are currently rarely considered in research and conservation programs”. Now we have changed this sentence to concern only elephants. In the beginning of discussion, we say that “Large numbers of animals across a range of species are routinely captured from the wild for diverse purposes, but surprisingly little is known of the consequences of such experiences for the subsequent long-term performance of those individuals”, which is true. In the same paragraph we say that these results have more general implications and are timely (although our population is a very unique case) but have modified the sentences to say exactly how and that also more elephant studies are needed from different contexts (translocation and zoo populations too) “Although capturing elephants for logging industry is a very restricted form of exploiting wild populations, our results have general implications beyond the study system. For example, such findings emphasize the need for animal welfare specialists, veterinarians and ecologists to identify the consequences of wild-capture on the success of individuals and populations and when relevant, to improve conservation and management practices so as to best support captured animals during the most critical periods after capture. Our results also stress the need for detailed further studies in elephants to identify potential costs of wild capture in different contexts (such as translocation and zoo environments) and more specific studies on taming/training related mortalities (and injuries) or long-term effects of sedatives.” In the last paragraph of discussion we say that capture effects should be studied in other animals too to find out the capture-related as well as long-term effects and also say that ignoring the potential effects MAY lead to erroneous conclusions in some studies. “Although our study population is unique and the use of elephants for logging is not a situation that applies to many other species, capture of elephants continues for legal or illegal purposes e.g.33-35 and capture of various other species from the wild is practised for diverse purposes each year e.g.1,2,63. Therefore, our results are timely and have three main implications. First, long-term effects of capture are currently not considered in research design and conservation programs, but our results show that capture can negatively influence animal performance for several years at least in species such as elephants. Therefore, using wild-captured animals to supplement medical trial populations e.g.64 or as reference groups for species-typical parameter values e.g.3,4 may lead to erroneous conclusions and both immediate (capture-related) as well as long-term effects of capture should be taken into account in further studies.” Finally, in the end of discussion, we say that our results imply that capture is costly for elephants and POTENTIALLY more generally among species, especially among

species with slow life- histories, which have been shown to also suffer more from zoo environments or in which captive-borns have shown to have higher survival than wild-borns. “Our study implies that capturing wild individuals in elephants and potentially more generally across species (especially among species with slow life-histories 11, 48-50), is costly for individual longevity and alternative methods should be sought to boost captive populations in order to avoid further capture from endangered wild populations.”.

The authors still start both the abstract and the discussion with a sentence about other species *and* then also clearly say that results can potentially be generalized to other species (in highlighting above). Lines 410-414: The final sentences of the discussion also refer to generalizing results across species. See also the section of their response highlighted in green below.

This does not seem well supported, for reasons we’ve already raised. *Which* other species for example? Do the authors have specific examples of cases where wild animals from other species are caught and then ‘broken’ for several weeks? Maybe raptors? Wild horses? Or orcas??

It’s also clear from the zoo studies (see below) that the effects of early experience can be the *opposite* to those reported here, with captive-born animals instead sometimes being at most risk of an early death.

Thus *taken as a whole*, the fascinating corpus of data from diverse elephant populations reveal the profound potential effects of early experience on lifetime survivorship. Capture and breaking have longterm adverse effects, *but so too does being zoo-born*. Both of these patterns highlight the importance of looking for similar effects in other species. But to suggest that capture *per se* might confer risks on other species is clearly weirdly narrow and myopic.

Second, how the authors compare and relate these to other elephant populations still needs some work. It’s these comparisons that are most interesting and relevant, and yet the authors do not do them justice. Other elephants are not even mentioned in the opening paragraph of the Introduction for example!

We have now been more precise about what implications our results have for different elephant populations (e.g. lines 277-280, 386-399). Note however that our focus isn’t in zoo populations or translocated populations (small minority of elephants kept in captivity) but we cite these studies because there aren’t many studies available on other similar populations of elephants (for example studies on wild Asian elephants are very rare). Concerning the opening paragraph, we prefer to keep it focused on the capture of animals in general and the shortage of long-term

studies about capture effects on survival across species. This is probably more relevant for the readership of a general journal. We do talk about elephants later after we have explained how capture effects can impact animals in general.

Thus in terms of the wild-capture of Asian elephants that still happens to this day (e.g. lines 112-113), just how relevant are these data: are today's wild-caught animals still subject to the same breaking and taming processes? Please make it clear. And in terms of the cessation of capture for logging in the 90's/2000, was a sense that wild-caught animals did not fare well part of the reason for the decision? Turning to translocated animals, again are the authors' data really relevant to them, since translocated elephants again are not subject to training and taming? Make it clear. Lastly, comparing these findings to those from zoo populations (and the reference populations used to assess zoo performance), it's here that the authors are perhaps weakest, as will be detailed next.

All animals captured nowadays in Myanmar go through similar taming process which we have described in the paper. We say this in the paper when we say that "All captured elephants go through taming..." (lines 462-465). Also, wild-capture still happens in other range countries too like in Indonesia together with similar taming and training periods (e.g. Mikota et al. 2008, cited in the paper). The reasons for the decision to ban the capture is unclear for us but the reasons aren't however having an effect on our study results or interpretation anyhow.

In line with this Reviewer's suggestion we have now removed one translocation reference but we prefer to keep the rest as they may give insights into the reasons behind higher mortality of wild-captured individuals compared to captive-born (lines 314-317) or show that the effect of translocation can also have long-term effects on animal wellbeing (lines 341-343, 303- 304)(see also our response to Reviewer 4 question).

We want to make it clear that although discussing similarities and differences to zoo elephants is interesting, our main focus isn't in zoo elephants; there are numerous other factors specific to zoo-conditions which are likely to have an effect on elephant survival therein. In Myanmar, wild-caught and captive-born elephants live, forage and work alongside one another, and the same governmental regulations apply for both capture types concerning data recording, workload and rest periods. The elephants are not provisioned, but instead forage unsupervised in forests at night, and the same basic veterinary care is available to all individuals. There are therefore many differences in our study compared to the previous studies with reference populations used to assess zoo performance. Our aim is not to provide specific comparisons but to highlight general discrepancies.

We're not requesting a general comparison between zoo and MTE elephants: we're highlighting that effects of early experience have also been found in zoo populations, but in the opposite direction to those reported here. Thus in zoos, captive-bred animals have shorter lives, while in MTE, they have longer lives. This contrast is surely interesting? One potential reason is that the wild-caught animals in zoos have already survived the breaking period; and another is that

captive-bred calves in zoos appear to be over-weight (perhaps not an issue in MTW?).

And thus in terms of the nameless other wild species that the authors desperately want to extrapolate to, it also suggests that sometimes captive-bred animals can be more vulnerable to e.g. health problems than are subjects caught from the wild. Again, surely this is worth acknowledging and highlighting when making suggestions for future work?

The previous work comparing Asians in zoos with MTE animals (Clubb et al. 2008) found that wild-caught MTE elephants have elevated risks of mortality throughout their juvenile years (1-9 years of age), compared to both wild-caught zoo animals and captive-bred MTE animals, but such effects were not evident by adulthood (10-70 years of age): an apparent difference to the effects reported here.

We thank the reviewer for bringing up a point which we wish to clarify. We do say in the paper that in captive populations in Asia wild-born elephants suffer from increased mortality compared to captive-born individuals and cite Clubb et al. 2008 and 2009 articles (lines 83- 84). We chose to not go into details however, because the analytical decisions made in the Clubb et al. study prevents direct comparisons. First, they used different analyses compared to ours which do not allow as accurate age-specific comparisons. In particular, Clubb et al. 2008 analyzed the effects in separate age categories (infants, juveniles and adults) which leads to juvenile survival appearing better in zoo populations as compared to MTE populations, although infant mortality is much higher in zoos compared to MTE elephants (Clubb et al. Supplementary Table 1). Therefore, the result during juvenile years could at least partially be because of unaccounted selective disappearance (weaker infants dying early in zoo populations and only more robust ones surviving beyond 1st year) and actually more juveniles surviving beyond 1st living year in MTE populations than in zoo populations.

*Second, the issue with adults is that the effects of capture depends, as we show it for the first time, on both the current age of the elephant and the age at capture. While a capture at old age increases mortality, time spent in captivity after capture leads to a decrease in the cost of capture (see results). Therefore, old elephants that have been captured at very young age present only a minor difference in mortality as compared to elephants born in captivity. **Since the Clubb et al. study does not distinguish between these two effects, we can only compare their results to ours at very general level.** The overall finding that MTE captive borns have higher survival than MTE wild caught elephants is actually evident in the Clubb et al. paper and this comparison is also significant (see Supplementary Table 2, WC all mortality vs CB mortality). The comparison is non-significant when comparing MTE wild born mortality from natural causes to MTE captive born mortality but this comparison isn't very relevant to the question posed in our article because considering only natural mortality in MTE wild-caught group, at least in part, removes the capture effects. (This has been done by first estimating that the capture and taming decreases survival up to eight years after capture and then the first eight years of each wild-caught elephant histories were left- censored in the subsequent analyses.)*

We would prefer not dwelling on such differences in study design and analytical power in our article, because it is unnecessary for the important points we want to get across with our study.

Clubb et al. 2008 also found that when animals were transferred between zoos, mortality risks were elevated for 4 years post-move, but after this period such an adverse effect could no longer be detected (obviously pretty relevant to this paper). Thus the authors' implication in their Introduction that their study is the first to follow animals for decades really does need some damping down.

We have actually mentioned this finding in the discussion, lines 333-334 “Interestingly, inter-zoo transfers also reduce Asian elephant survivorship, an effect detected up to four years after the initial transfer³⁶. ”Also, we mention the even more important finding from Clubb et al. 2008 paper that the capture effects lasted 8 years (lines 331-333).

OK good

Concerning the novelty of this study, we have been careful to say on lines 95-97 that our study is the first to study age-specific mortality effects of capture in a long-lived mammal.

Not capture though: capture and breaking

This is accurate and true, because previous studies do not truly compare age-specific differences in yearly mortalities with confounding time-varying variables such as time since capture effects like we do. We also say in the discussion, lines 339-341, that “To our knowledge, there are no comparable studies investigating capture effects on mortality for decades in any species (but see Saraux et al. 56 on effects of tagging)”. This is because Clubb et al investigated whether the capture effects could last for 0-14 years, not longer (see Supplementary material), and concluded the effect lasts 8 years after capture and did not investigate whether the capture effects differ in duration among younger and older elephants.

True

We investigated the possibility that the capture effects can last decades and that the results can differ with elephant's age and capture age. Of course they investigated elephants aged between 0-70+ years but this is not the same as studying how long the capture effect lasts and how the time since capture variable modifies the outcome (range in our study 0-45 years). Given the surprising and interesting results concerning age and time spent in captivity, we consider our novelty statement justified and hope that you agree with this.

Furthermore, that “the early maternal environment of those born in captivity is typically different from wild-captured animals” (lines 76-78) is also not special to zoos, but applies to MTE animals too (who are unlikely to be living in true matrilineal groups).

This paragraph (lines 68-81) is specifically focused on explaining the different management factors that lead to higher documented mortality among zoo elephants as compared to WILD

elephants, as we state in the second sentence. We are hence a bit puzzled how the MTE animals are relevant here, given they are neither wild nor live in zoos. Yes it is true that MTE animals do not necessarily have all genetically related individuals around as calves compared to calves born in the wild, but the calves are always allowed to be with the mothers until taming around age 4 or 5 and they usually also have at least one or more allomothers. Their social environment includes several different aged and sex elephants and they can also meet their true relatives when they are released to forests during nights. Also grandmothers are present for many calves increasing their survival (Lahdenperä et al. Sci Rep 2016), and this all makes the early environment in timber camps very different to that experienced in zoos. Here we specifically talk about multi-generational family-groups, which are very rare in zoo environments compared to MTE population. As said, the point of the paragraph is however to illustrate differences between zoos and wild Asian elephants, so we have left this sentence as it is.

OK

In addition, the Clubb et al. 2008 paper (built on further in 2009) found that zoo-bred Asian elephants had much higher mortality risks than wild-caught Asian elephants also housed in zoos. This is obviously completely the opposite to what Clubb et al. 2008 and this manuscript find for MTE animals, and it seems odd to downplay this remarkable contrast with a mere “(vice versa in zoos)” cryptic clause, in parentheses, in the Introduction (line 84). A fuller acknowledgement of this paradox is surely warranted. Why the zoo and MTE population differ in the direction of their birth origin effect is raised at lines 366-368, but almost as a rhetorical question with no credible suggestions offered as to potential explanations. I think the authors can do better than this, especially as the lead author was a co-author of the Clubb et al. 2009 paper that identified heavy infant birthweights (perhaps indicating neonatal obesity) as a significant difference between zoo and MTE captive-bred calves, and that also then suggested the high nutritional plane of zoo elephants as a possible cause of the poor survival of captive-bred elephants in zoo populations. This idea is surely still a plausible, testable one, no?

To address this, we have now expanded discussing the differences in the origin effects in the Discussion.

OK great

We do not want to raise these points in the Introduction, given our own results are not clear to the readers yet at that point and thus the comparison to zoo results cannot be made.

OK

Furthermore, Nature Communications imposes length restrictions on the Introduction which we already exceed. We now say that “In contrast to the situation in our semi-captive Myanmar population, captive(zoo)-born Asian elephants in European zoos have poorer survivorship than wild-captured animals” and cite both Clubb et al. papers (lines 387-389).

OK great

We also say that these comparisons and differences in the origin effects in zoos and timber camps reflect the many problems that zoo elephants face. Please accept that our focus here isn't the zoo population mortality in relation to birth origin effects but rather, timber camp elephant mortalities in wild-caught and captive born animals. We cannot test the nutritional intake of zoo vs timber elephants since our article only focuses on the latter, or of captive- born and wild-born MTE elephants since both forage independently in the forest at night unobserved and no data exists on nutritional status of the historical animals throughout their lives. However we have now added a sentence saying that the different mortality pattern in zoos call for further studies and the higher nutritional plane and stress are two potential reasons for this difference, see lines 396-399.

Minor comments:

The phrase "before and after capture" (line 85) also needs toning down too, since the authors do not assess wild Asian elephants before capture and so should not imply that this is a unique angle of their paper!

Corrected.

Lines 99-100 say that around 15,000 Asian elephants live in captivity, but lines 371-372 say about 1000 are: needs editing to clarify so not inadvertently confusing (I think the latter applies to non-timber animals only?).

We thank the reviewer for noticing these confusing sentences. Yes, we meant that the 15,000 elephants includes all Asian elephants living in captivity in range countries, including timber elephants, elephants in temples and zoos and those owned by private people. The second number includes only those elephants living in zoos, safari parks and circuses world-wide. We have tried to clarify the sentences in the text, see lines 99-101 and 399-400.

Figure 1: make it clear that these are data for all MTE animals?

We thank the reviewer, clarification added to the Figure 1 legend.

OK great.

One last little thing:

Lines 114 & 220 refer to capture of elephants involved in human-elephant conflict. We believe these animals are captured for translocation? But if this is the case they should clearly state it.

OVERALL:

I THINK WE ARE CONVERGING ON A MUTURALLY ACCEPTABLE MS, AND WE HAVE NOW DOUBT THAT THIS IS IMPORTANT AND EXCITING WORK. WE'RE LOOKING FORWARD TO SEEING IT PUBLISHED.

TO REITERATE AND SUM OUR LAST REQUESTS:

- 1) IN THE ABSTRACT, AND IN ALL SUB-TITLES, PLEASE USE THE PHRASE 'CAPTURE AND INTENSIVE TAMING' OR 'CAPTURE AND BREAKING' TO DENOTE WHAT IS HAPPENING TO WILD-BORN ANIMALS IN MTE. THAT WILL KEEP THINGS CLEAR AND UNAMBIGUOUS.**

- 2) IN THE DISCUSSION, MAKE IT CLEAR THAT BEING CAUGHT FROM THE WILD AND BROKEN IS *NOT THE ONLY RISK FACTOR WITH POTENTIAL RELEVANCE FOR OTHER SPECIES*: SINCE BEING ZOO-BORN HAS ALSO BEEN SHOWN TO PREDISPOSE ELEPHANTS TO PREMATURE DEATHS, THE RICH DATASETS AVAILABLE FOR DIVERSE ELEPHANTS *TOGETHER* SHOW THAT EARLY EXPERIENCE CAN HAVE PROFOUND *AND UNPREDICTABLE EFFECTS* ON THE HEALTH OF WILD ANIMALS KEPT IN CAPTIVITY. PERHAPS CAPTURE IS SOMETIMES THE PROBLEM, PERHAPS CAPTURE AND TAMING/BREAKING; BUT IT'S CLEAR TOO THAT SOMETIMES OTHER FACTORS MAKE THE F1 CAPTIVE GENERATION INSTEAD EVEN MORE VULNERABLE! FUTURE RESEARCHERS SHOULD BE MINDFUL OF *ALL* OF THESE IMPORTANT AND INTRIGUING POSSIBLE EFFECTS.**

Point-by-point responses to reviewer comments

We have revised our manuscript according to the reviewers' comments as outlined below. The comments by the all referees are in normal font, while ours follow in italics.

Reviewers' comments:

Reviewer #1 (Remarks to the Author):

This revised version nicely accounts for most problems I identified when reading the previous draft. In particular, I really enjoyed the new survival analysis based on a generalization of the Siler model. That being said, I still have several minor concerns, mostly about the presentation of the new statistical analyses:

We thank J.M. Gaillard for his enthusiasm and thorough review. His suggestions throughout this review process have truly improved this article. Most of his remaining criticisms were stemming from lack of confidence intervals, unclear passages in the text and from the fact that we did not provide the source code documenting each step of the analysis. We have now addressed all of these remaining 3 points.

l. 212: « lowest » instead of « best »

Corrected, line 226.

l. 212-213: Unclear what you measured the « probability of model selection ». A delta-AIC (not DAIC as stated) of 0.9 should lead to retain the model with the lowest number of parameters based on the principle of parsimony.

Thanks for pointing out that we did not explain this well enough. We deliberately chose not to retain a single most parsimonious model. This is because we share the view that it is better practice to take model selection uncertainty into account instead of neglecting it. When such uncertainty is low, then relying on a single model (the one with lowest AIC) may be acceptable, but as we shall see, the model selection uncertainty is at times large in our case.

*The idea behind the approach we chose -- multi-model inference (see e.g. Chapter 4 in *Model Selection and Multimodel Inference* by Burnham and Anderson 1998 2nd edition) -- is to recognize that the most parsimonious model may actually differ among different samples drawn from the same population; and the goal is to include this uncertainty into parameter estimation (inference given a model set instead of inference conditional to a single model). We have now clarified that we follow such an approach on lines 155-156 & 595-596.*

We used the term “probability of (model) selection” to refer to such model selection uncertainty. We define the “probability of model selection” as the frequency at which a given model would be best among all samples, or equivalently as the probability that the best model in a given random sample really is the best model in the population. We have now defined that on lines 611-614. A single sample does not prevent the computation of “probability of

model selection”. Equivalently to non-parametric bootstrap which is used to draw inference on between samples variation from a single sample, it is possible (under some assumption) to compute the probability of model selection from a single sample. Different methods have been proposed in the literature and identifying the best one remains a debated question in statistics. We thus relied on the common usage which considers that the probability of model selection can be approximated by the relative Akaike weight of the model (see section 6.4.5 Akaike Weights as Bayesian Posterior Model Probabilities in Burnham and Anderson.) This information is now present in the manuscript (lines 614-617).

Because our top three models have relative weights of respectively 0.38, 0.30, 0.14, there is almost 50% chance that another dataset from the same population would have led us to identify our second or third best model as the best one. Albeit this being an a posteriori result, this is in itself the best justification for why we used multi-model inference. We have now added this information in the manuscript (lines 225-229).

When we wrote “the model with the lowest AIC presents a probability of selection only 1.27 times higher than the second best fit” (lines 226-227), the value 1.27 thus directly corresponds to the ration between the probabilities of model selection of the two models. Since we computed this ratio as the ratio between the relative weights of the two models, this ratio is also the so-called “evidence ratio” (Burnham and Anderson, page 78). We have now mentioned this correspondence in the manuscript (line 226-227) since more readers may be familiar with this term despite it being perhaps more abstract (it has been introduced as the ratio between likelihoods).

It is unclear from the reviewer’s comment if he had issues with how we measured the “probability of model selection” or just with the terminology itself. We think that our edits should have now covered both possibilities. For your information, we decided to stick to the terminology “probability of (model) selection” because we find it clearer to understand than possible alternatives. Burnham and Anderson tend to simply use “probability of model” (section 2.9.1) but they later admit that this lacks clarity: “Saying ‘ p_i is the probability of model g_i ’ we must be referring to the probability that this model is the target model of the selection procedure” p348. Further, they also sometimes use an expression that is synonymous to our choice: “model selection probabilities” (section 4.5.2)

We fixed the typo where D was used instead of delta (line 227).

1. 307: Remove « fits »

Removed, line 330.

1. 501-503: The exact differences between the Siler model and the model used here should be made explicit. The Siler model only has 3 « w » terms and it is unclear to me why the third « w » term and not the second « w » term was assumed to be constant. The simple extrapolation of the Siler model should have considered both w_2 and w_3 as constant terms. There is nothing wrong with the model but a better justification is required.

Contrary to Siler’s original model, w_1 , w_2 and w_3 can all depend on the sex, so none of these parameters are considered as being constant. This is now indicated in the manuscript (line 530-531).

l. 504: This statement is not correct. The Siler model includes 5 parameters and should correspond to:

$$p = w1 \cdot e(-b1.age) + w2 + w3.e(b2.age)$$

This is quite different from what it is reported in the first line of the equation!

Our original statement was unclear but it was correct. Within a given sex, birth cohort and location, our model is (for individuals born in captivity): $p = w1 \cdot e(-b1.age) + w3 + w2.e(b2.age)$, which is strictly identical to Siler's original model. We have now clarified this in the manuscript (lines 531-532). We present Siler's original second term in third position and vice versa. This is because we prefer to define $w3$ as the "mortality independent from aging" (line 540), while Siler defined it as the "mortality for mature animals". We prefer our presentation because all individuals (including young and very old ones) suffer the so-called "mortality for mature animals". This is only a question of semantic and presentation but again the mathematical model remains the same. Since not many readers will know about Siler's model we prefer to define things in the clearest possible way and find it unnecessary to dwell about why our notation slightly differ superficially from Siler's one, given the journal limits on the paper length. Of course, if the Editor so wishes, all these details can be added but in any case the entire R code is now accessible to readers (see below).

l. 506-507: In the Siler model, the second term is a constant mortality of prime-aged adults and the third term describes the mortality increase with age, not the reverse.

Ok, but a mathematical definition does not constrain any order among the terms belonging to a sum. We deliberately reordered the terms to be clearer than Siler's original publication. (see previous comment).

l. 511: This statement is not correct. There is no need to assume a null mortality at any age. The Gompertz model provides a model of age-specific changes from a specific age (usually assumed to be the age at first reproduction) so that immature individuals are excluded and there is no prime-age stage with a constant mortality rate.

This is a misunderstanding. The "mortality for immature animals" is the term used by Siler to refer to $w1.e(-b1.age)$ and the "mortality for mature animals" to $w3$. So if those two terms (which we clearly defined in the text) are null, then only $w2.e(b2.age)$ remains. This latter term is precisely the Gompertz equation for mortality, and what we indeed say in the ms - we are not claiming that data cannot be shaped in a way that would allow the Gompertz model to be applied if mortality was not null. We think that the reviewer got confused because he may have interpreted "mortality for immature animals" and "mortality for mature animals" outside the context of their mathematical definition. We have now clarified this in the manuscript (lines 536-548).

l. 515: « age-specific changes of mortality rate », not « baseline »

We refer to "baseline mortality" because this mortality is perceived by all elephants (captive born or captured individuals). We cannot call the first part of our extended Siler model "age-specific changes for mortality rate" because components of the second part such as $w5.age$ and $b5^{age}$ also define "age-specific changes of mortality rate". So we stick to our terminology because baseline is the term that we could come up with that most accurately

represents the biological meaning of the first part of the model. We have now clarified this in the manuscript (lines 532-535 & 565-568).

1. 530: « describes » instead of « describe »

Corrected.

1. 531-532: Rewrite the sentence.

Corrected, lines 566-568.

1. 544-545: Provide the exact package and the reference

The package and reference was (and still is) in the manuscript (line 588).

1. 548-550: Provide a reference to justify this metric.

We have now cited the chapter 7 of the book by Allison (2010), line 586. While this chapter does not detail the computation of the likelihood of the binary logistic regression (which is available in any statistical text book, Wikipedia and so on), it explains quite well both the fundamental connection between survival analysis and logistic regression and that left censorship is correctly accounted for by simply considering only the actually observed intervals for the computation of the likelihood.

560-563: Estimates without associated errors are not very useful. That getting confidence intervals would be computer intensive is a poor justification!

We are now providing confidence intervals for all parameters (Supplementary Table 1). This has been possible because we recoded the entire R code into small efficient functions (as a R package), coded the computing bottleneck into C++ (using Rcpp), and implemented parallel computation for the bootstrap (see R package). For information, despite the considerable gain in computing speed, fitting the bootstrapped datasets still took 2500 CPU hours (which would represent more than 100 days on a computer with a single processor).

1. 565: The AIC does not allow estimating the « predictive power » of a model.

Actually it does and this is perhaps the most useful feature of the AIC despite that this fact is not known by most biologists. Technically, the AIC is an estimator the Kullback-Leibler divergence, i.e. a measure of distance between two probability distributions, the fitted model and the true unobserved reality (i.e., the distribution of new observations to be predicted) (Akaike, 1973). The less information is lost the closer we are to the true data generating process and thus the greater is the ability of the model to predict new observations. The equivalence is not just semantical: among a set of candidate models, the model with the smallest AIC has been shown (asymptotically) to be the one with the highest predictive power (as measured by residual sum of squares) in the context of the leave-one-out cross validation. This results has been demonstrated multiple times and for different kind of models (e.g. Stone 1977: An Asymptotic Equivalence of Choice of Model by Cross-Validation and Akaike's Criterion. J. Royal Stat. Soc. B 39(1), 44-47; Fang 2011: Asymptotic equivalence between

cross-validations and Akaike information criteria in mixed-effects models. J. Data Sci. 9(1), 15-21.). In practice, it means that the model with the smallest AIC is likely to be the one that predicts best the value of a new observation from its associated covariates. It is thus precisely a measure of the predictive power and we recommend that it should be interpreted as such (we find this more helpful than the alien entropy based arguments from information theory with which it is usually introduced). We have now added a reference to Stone's paper (line 595) to clarify that we are not making things up.

l. 570: Remove « baseline »

Corrected, line 600.

l. 576: « effects » instead of « effect »

Corrected.

l. 584: « model » instead of « models »

Corrected.

l. 592: « chose » instead of « choose »

Corrected.

l. 604-605: This statement is far too vague. Instead, all the code used and data analyzed in that work should be made available to readers.

We are now providing all the code (and a small subset of the data) as an R package which reproduces the results of this paper. The whole dataset is available for re-analysis only on request from Professor Lummaa due to privacy restrictions (the data is Myanmar Government owned and Myanmar Timber Enterprise does not allow us to share the data publicly on a server – but Prof. Lummaa can provide it for those interested).

l. 797: Remove « Baseline »

Like we already explained earlier we prefer to keep the word “baseline” as this mortality is perceived by all elephants (captive born or captured individuals)(Figure legend 1).

l. 841: « the best among the candidate models fitted » instead of « the best »

Corrected, Table 1 legend.

J.M. Gaillard

Reviewer #3 (Remarks to the Author):

This paper is much improved, but there are still two problems with the way it is framed. First, the authors exaggerate how relevant their data and findings are to other types of wild-caught

animal; and second, they do not properly discuss how these data and findings relate to translocated and zoo elephants.

We thank the reviewer for having made many suggestions for how we should discuss the relevance of our findings in the context of other elephant populations or other species. These suggestions have truly improved the ms over the entire review process. We have made further modifications as detailed below, but we must respectfully disagree about how to best frame the paper. Any remaining generalizations to other species are worded only so as to inspire further research, and the relevance of our findings to zoo elephants is discussed on multiple occasions as detailed below.

When Asian elephants are caught to work in the logging industry, the capture and move to captivity are then followed by several weeks of intense and harsh taming/breaking/training. The authors do acknowledge this, and one of the other referees highlights it too. However, the authors downplay this in the language they use (referring only to the effects of ‘capture’ throughout their paper), and also in the over-zealous way they claim that their results apply to the members of other species caught from the wild (in the Abstract, Introduction and Discussion). The authors repeat these claims about generality to other species several times (e.g. twice in the Abstract alone), and yet are necessarily vague with them too, as it’s really not clear that any other types of wild animal undergo anything like this (do they? If yes be specific). So, we would request two types of editing to acknowledge the uniqueness of the ‘Asian-elephant-captured-to-work’ scenario. First, please only suggest that these results apply to other species once and once only (not repeatedly), and also make it clear that this is just a hypothesis. Second, please use the phrase ‘capture and taming’ (or similar) in the Abstract, Discussion and all sub-titles (e.g. at lines 164 and 194) throughout the paper, as it is clear that the effects of capture per se cannot be parsed out from those of taming/training, and so the two must be presented as a package (such that calling these ‘capture effects’ is simplistic and potentially misleading).

While we agree with many suggestions by the reviewer, it is important to recall that captive-born elephants also undergo taming with the same aim and objective as the wild-caught elephants: to obey human commands and work efficiently in timber industry. Captive-born elephants are tamed around age 4-5 years generally using the same personnel, infrastructure and methods as is used for the wild-caught elephants. Before that, calves are not used to human-handling, commands or caretaking and could as well be considered as being “wild”. This fact makes our study so unique, and also means that the alternative terminology proposed by the reviewer is not suited to our study design. It is true that the taming and breaking are harsher for wild-born individuals to a degree that also depends on the age, sex and personality. We say this explicitly in the paper (lines 133-137, 462-465, 475-478) and cite an article (line 465) which gives details about the taming methods used for captive-born elephants: Zaw Min Oo 2010 The training methods used in Myanma Timber Enterprise 2010 Gajah 33:58-61). It is noteworthy that the taming can also be harsh for the captive-born calves, regularly leading to accidental deaths as recorded by the stated causes of death (see lines 311-314 and citation to Mar et al. 2012 PlosOne 7(3):e32335). Also, due to the intense training, the baby elephants may get wounds, injuries, and stress. We say on lines 60-64 (second paragraph of the whole article) that taming/breaking can have effects on post-capture lifespan and there may be differences in the management of captive-born and wild-born animals. We also say now in the end of the introduction, lines 137-139, that part of the effects seen in wild-caught elephants and captive-born elephants may be due to differences

during taming period and from now on we refer to all these effects as capture effects. Just to be sure, we repeat this in each section (Results 182-183, Discussion 297-300,309-314, Methods 462-465, 475-478). For all these reasons and for keeping the wording as light and lively as possible we prefer to stick to the term “capture” instead of using the “capture and taming” which we find more misleading than our original formulation.

We are somewhat puzzled why the Reviewer has the perception that we claim the results to apply to other species as such. We have carefully checked the ms and have found no evidence of such claims. Nevertheless, to settle this argument, we have now aimed to be more accurate with the sentences saying how our results apply to other species. In the abstract we do not claim that the results apply to the members of other species caught from the wild (and did not say previously even one time), instead we say that our results are timely because elephants as well as many other animal species are still being captured to supplement captive populations even though there have been alarming declines in wild populations globally. In the introduction, we did not repeatedly make this claim, but we said in the end of the introduction that “The long-term differences between captive-born and wild-captured animals shown by our study are currently rarely considered in research and conservation programs”. Now we have changed this sentence to concern only elephants. In the beginning of discussion, we say that “Large numbers of animals across a range of species are routinely captured from the wild for diverse purposes, but surprisingly little is known of the consequences of such experiences for the subsequent long-term performance of those individuals”, which is true. In the same paragraph we say that these results have more general implications and are timely (although our population is a very unique case) but have modified the sentences to say exactly how and that also more elephant studies are needed from different contexts (translocation and zoo populations too) “Although capturing elephants for logging industry is a very restricted form of exploiting wild populations, our results have general implications beyond the study system. For example, such findings emphasize the need for animal welfare specialists, veterinarians and ecologists to identify the consequences of wild-capture on the success of individuals and populations and when relevant, to improve conservation and management practices so as to best support captured animals during the most critical periods after capture. Our results also stress the need for detailed further studies in elephants to identify potential costs of wild capture in different contexts (such as translocation and zoo environments) and more specific studies on taming/training related mortalities (and injuries) or long-term effects of sedatives.” In the last paragraph of discussion we say that capture effects should be studied in other animals too to find out the capture-related as well as long-term effects and also say that ignoring the potential effects MAY lead to erroneous conclusions in some studies. “Although our study population is unique and the use of elephants for logging is not a situation that applies to many other species, capture of elephants continues for legal or illegal purposes e.g.33-35 and capture of various other species from the wild is practised for diverse purposes each year e.g.1,2,63. Therefore, our results are timely and have three main implications. First, long-term effects of capture are currently not considered in research design and conservation programs, but our results show that capture can negatively influence animal performance for several years at least in species such as elephants. Therefore, using wild-captured animals to supplement medical trial populations e.g.64 or as reference groups for species-typical parameter values e.g.3,4 may lead to erroneous conclusions and both immediate (capture-related) as well as long-term effects of capture should be taken into account in further studies.” Finally, in the end of discussion, we say that our results imply that capture is costly for elephants and POTENTIALLY more generally among species, especially among species with slow life-histories, which have been shown to also suffer more from zoo environments or in which

captive-borns have shown to have higher survival than wild-borns. “Our study implies that capturing wild individuals in elephants and potentially more generally across species (especially among species with slow life-histories¹¹, 48-50), is costly for individual longevity and alternative methods should be sought to boost captive populations in order to avoid further capture from endangered wild populations.”.

Second, how the authors compare and relate these to other elephant populations still needs some work. It's these comparisons that are most interesting and relevant, and yet the authors do not do them justice. Other elephants are not even mentioned in the opening paragraph of the Introduction for example!

We have now been more precise about what implications our results have for different elephant populations (e.g. lines 277-280, 386-399). Note however that our focus isn't in zoo populations or translocated populations (small minority of elephants kept in captivity) but we cite these studies because there aren't many studies available on other similar populations of elephants (for example studies on wild Asian elephants are very rare). Concerning the opening paragraph, we prefer to keep it focused on the capture of animals in general and the shortage of long-term studies about capture effects on survival across species. This is probably more relevant for the readership of a general journal. We do talk about elephants later after we have explained how capture effects can impact animals in general.

Thus in terms of the wild-capture of Asian elephants that still happens to this day (e.g. lines 112-113), just how relevant are these data: are today's wild-caught animals still subject to the same breaking and taming processes? Please make it clear. And in terms of the cessation of capture for logging in the 90's/2000, was a sense that wild-caught animals did not fare well part of the reason for the decision? Turning to translocated animals, again are the authors' data really relevant to them, since translocated elephants again are not subject to training and taming? Make it clear. Lastly, comparing these findings to those from zoo populations (and the reference populations used to assess zoo performance), it's here that the authors are perhaps weakest, as will be detailed next.

All animals captured nowadays in Myanmar go through similar taming process which we have described in the paper. We say this in the paper when we say that “All captured elephants go through taming...” (lines 462-465). Also, wild-capture still happens in other range countries too like in Indonesia together with similar taming and training periods (e.g. Mikota et al. 2008, cited in the paper). The reasons for the decision to ban the capture is unclear for us but the reasons aren't however having an effect on our study results or interpretation anyhow.

In line with this Reviewer's suggestion we have now removed one translocation reference but we prefer to keep the rest as they may give insights into the reasons behind higher mortality of wild-captured individuals compared to captive-born (lines 314-317) or show that the effect of translocation can also have long-term effects on animal wellbeing (lines 341-343, 303-304)(see also our response to Reviewer 4 question).

We want to make it clear that although discussing similarities and differences to zoo elephants is interesting, our main focus isn't in zoo elephants; there are numerous other factors specific to zoo-conditions which are likely to have an effect on elephant survival therein. In Myanmar, wild-caught and captive-born elephants live, forage and work alongside one another, and the same governmental regulations apply for both capture types

concerning data recording, workload and rest periods. The elephants are not provisioned, but instead forage unsupervised in forests at night, and the same basic veterinary care is available to all individuals. There are therefore many differences in our study compared to the previous studies with reference populations used to assess zoo performance. Our aim is not to provide specific comparisons but to highlight general discrepancies.

The previous work comparing Asians in zoos with MTE animals (Clubb et al. 2008) found that wild-caught MTE elephants have elevated risks of mortality throughout their juvenile years (1-9 years of age), compared to both wild-caught zoo animals and captive-bred MTE animals, but such effects were not evident by adulthood (10-70 years of age): an apparent difference to the effects reported here.

We thank the reviewer for bringing up a point which we wish to clarify. We do say in the paper that in captive populations in Asia wild-born elephants suffer from increased mortality compared to captive-born individuals and cite Clubb et al. 2008 and 2009 articles (lines 83-84). We chose to not go into details however, because the analytical decisions made in the Clubb et al. study prevents direct comparisons. First, they used different analyses compared to ours which do not allow as accurate age-specific comparisons. In particular, Clubb et al. 2008 analyzed the effects in separate age categories (infants, juveniles and adults) which leads to juvenile survival appearing better in zoo populations as compared to MTE populations, although infant mortality is much higher in zoos compared to MTE elephants (Clubb et al. Supplementary Table 1). Therefore, the result during juvenile years could at least partially be because of unaccounted selective disappearance (weaker infants dying early in zoo populations and only more robust ones surviving beyond 1st year) and actually more juveniles surviving beyond 1st living year in MTE populations than in zoo populations.

*Second, the issue with adults is that the effects of capture depends, as we show it for the first time, on both the current age of the elephant and the age at capture. While a capture at old age increases mortality, time spent in captivity after capture leads to a decrease in the cost of capture (see results). Therefore, old elephants that have been captured at very young age present only a minor difference in mortality as compared to elephants born in captivity. **Since the Clubb et al. study does not distinguish between these two effects, we can only compare their results to ours at very general level.** The overall finding that MTE captive borns have higher survival than MTE wild caught elephants is actually evident in the Clubb et al. paper and this comparison is also significant (see Supplementary Table 2, WC all mortality vs CB mortality). The comparison is non-significant when comparing MTE wild born mortality from natural causes to MTE captive born mortality but this comparison isn't very relevant to the question posed in our article because considering only natural mortality in MTE wild-caught group, at least in part, removes the capture effects. (This has been done by first estimating that the capture and taming decreases survival up to eight years after capture and then the first eight years of each wild-caught elephant histories were left-censored in the subsequent analyses.)*

We would prefer not dwelling on such differences in study design and analytical power in our article, because it is unnecessary for the important points we want to get across with our study.

Clubb et al. 2008 also found that when animals were transferred between zoos, mortality risks were elevated for 4 years post-move, but after this period such an adverse effect could no longer be detected (obviously pretty relevant to this paper). Thus the authors' implication in

their Introduction that their study is the first to follow animals for decades really does need some damping down.

We have actually mentioned this finding in the discussion, lines 333-334 “Interestingly, inter-zoo transfers also reduce Asian elephant survivorship, an effect detected up to four years after the initial transfer³⁶.” Also, we mention the even more important finding from Clubb et al. 2008 paper that the capture effects lasted 8 years (lines 331-333).

Concerning the novelty of this study, we have been careful to say on lines 95-97 that our study is the first to study age-specific mortality effects of capture in a long-lived mammal. This is accurate and true, because previous studies do not truly compare age-specific differences in yearly mortalities with confounding time-varying variables such as time since capture effects like we do. We also say in the discussion, lines 339-341, that “To our knowledge, there are no comparable studies investigating capture effects on mortality for decades in any species (but see Saraux et al. 56 on effects of tagging)”. This is because Clubb et al investigated whether the capture effects could last for 0-14 years, not longer (see Supplementary material), and concluded the effect lasts 8 years after capture and did not investigate whether the capture effects differ in duration among younger and older elephants. We investigated the possibility that the capture effects can last decades and that the results can differ with elephant’s age and capture age. Of course they investigated elephants aged between 0-70+ years but this is not the same as studying how long the capture effect lasts and how the time since capture variable modifies the outcome (range in our study 0-45 years). Given the surprising and interesting results concerning age and time spent in captivity, we consider our novelty statement justified and hope that you agree with this.

Furthermore, that “the early maternal environment of those born in captivity is typically different from wild-captured animals” (lines 76-78) is also not special to zoos, but applies to MTE animals too (who are unlikely to be living in true matrilineal groups).

This paragraph (lines 68-81) is specifically focused on explaining the different management factors that lead to higher documented mortality among zoo elephants as compared to WILD elephants, as we state in the second sentence. We are hence a bit puzzled how the MTE animals are relevant here, given they are neither wild nor live in zoos. Yes it is true that MTE animals do not necessarily have all genetically related individuals around as calves compared to calves born in the wild, but the calves are always allowed to be with the mothers until taming around age 4 or 5 and they usually also have at least one or more allomothers. Their social environment includes several different aged and sex elephants and they can also meet their true relatives when they are released to forests during nights. Also grandmothers are present for many calves increasing their survival (Lahdenperä et al. Sci Rep 2016), and this all makes the early environment in timber camps very different to that experienced in zoos. Here we specifically talk about multi-generational family-groups, which are very rare in zoo environments compared to MTE population. As said, the point of the paragraph is however to illustrate differences between zoos and wild Asian elephants, so we have left this sentence as it is.

In addition, the Clubb et al. 2008 paper (built on further in 2009) found that zoo-bred Asian elephants had much higher mortality risks than wild-caught Asian elephants also housed in zoos. This is obviously completely the opposite to what Clubb et al. 2008 and this manuscript find for MTE animals, and it seems odd to downplay this remarkable contrast with a mere “(vice versa in zoos)” cryptic clause, in parentheses, in the Introduction (line 84). A fuller

acknowledgement of this paradox is surely warranted. Why the zoo and MTE population differ in the direction of their birth origin effect is raised at lines 366-368, but almost as a rhetorical question with no credible suggestions offered as to potential explanations. I think the authors can do better than this, especially as the lead author was a co-author of the Clubb et al. 2009 paper that identified heavy infant birthweights (perhaps indicating neonatal obesity) as a significant difference between zoo and MTE captive-bred calves, and that also then suggested the high nutritional plane of zoo elephants as a possible cause of the poor survival of captive-bred elephants in zoo populations. This idea is surely still a plausible, testable one, no?

To address this, we have now expanded discussing the differences in the origin effects in the Discussion. We do not want to raise these points in the Introduction, given our own results are not clear to the readers yet at that point and thus the comparison to zoo results cannot be made. Furthermore, Nature Communications imposes length restrictions on the Introduction which we already exceed. We now say that “In contrast to the situation in our semi-captive Myanmar population, captive(zoo)-born Asian elephants in European zoos have poorer survivorship than wild-captured animals” and cite both Clubb et al. papers (lines 387-389). We also say that these comparisons and differences in the origin effects in zoos and timber camps reflect the many problems that zoo elephants face. Please accept that our focus here isn’t the zoo population mortality in relation to birth origin effects but rather, timber camp elephant mortalities in wild-caught and captive born animals. We cannot test the nutritional intake of zoo vs timber elephants since our article only focuses on the latter, or of captive-born and wild-born MTE elephants since both forage independently in the forest at night unobserved and no data exists on nutritional status of the historical animals throughout their lives. However we have now added a sentence saying that the different mortality pattern in zoos call for further studies and the higher nutritional plane and stress are two potential reasons for this difference, see lines 396-399.

Minor comments:

The phrase “before and after capture” (line 85) also needs toning down too, since the authors do not assess wild Asian elephants before capture and so should not imply that this is a unique angle of their paper!

Corrected.

Lines 99-100 say that around 15,000 Asian elephants live in captivity, but lines 371-372 say about 1000 are: needs editing to clarify so not inadvertently confusing (I think the latter applies to non-timber animals only?).

We thank the reviewer for noticing these confusing sentences. Yes, we meant that the 15,000 elephants includes all Asian elephants living in captivity in range countries, including timber elephants, elephants in temples and zoos and those owned by private people. The second number includes only those elephants living in zoos, safari parks and circuses world-wide. We have tried to clarify the sentences in the text, see lines 99-101 and 399-400.

Figure 1: make it clear that these are data for all MTE animals?

We thank the reviewer, clarification added to the Figure 1 legend.

Reviewer #4 (Remarks to the Author):

The manuscript summarizes survivorship using an exceptional data set on Asian elephants. Insights derived regarding differences between mortality in captive born and wild capture approaches are valuable and interesting. Writing is awkward in places, and close editing is needed. The analysis approach is interesting, but I have a few concerns I raise on the modeling approach and the lack of ability to derive CI on estimated trends. This can be alleviated by presenting the raw data for each of the relationships inferred through the model. I would also present survivorship curves using cox regression as an alternative modeling approach that allows incorporation of covariates and will enable presentation of uncertainty with outputs.

We thank the new reviewer for the positive view on the importance of our dataset, analyses and findings. We have now edited the text and improved the justification of our modelling approach, also discussing why the proposed Cox approach is not sufficient. Last but not least we now provide CIs on all parameter estimates in the paper, and the proposed Cox regression curves below for your information. We hope that these revisions now make our article suitable for publication in Nature Communications.

Intro

Line 68: the authors conflate translocation with wild capture in their references and framing of the motivation for the paper. Looking at translocation impacts and survivorship is substantially different given that the mortality occurs only during initial capture and transport. If I am correct, that period was not analyzed in the presented data. As such, I would suggest separating these two in the framing of the manuscript.

We thank the reviewer, we agree that the translocation does not totally correspond to the capture event although capture always precedes translocation and both can certainly have long-term consequences that we examine here for the capture. You are correct that we do not have data on initial capture-related mortality which we acknowledge in the paper many times (lines 109-111, 185-187, 216-219 and 371-373, 482-485); instead we focus on the interesting long-term consequences. It is noteworthy that also the translocation can have longer-term effects on animals, such as increased stress levels after prolonged periods (refs on zebra & rhinoceros in the paper, Franceschini et al. Anim Conserv. 2008; Capiro et al. Zoo Bio. 2014, lines 341-343 and review Teixeira et al. Anim. Behav. 2007, lines 300-301). We have now removed one reference concerning elephant translocations (lines 68 and 378) in line with the Reviewer suggestion. However, as there are hardly any capture studies conducted so far on large mammals which would have studied the duration of capture effects on survival, and some reasons for the lower performance might concern both translocated elephants and wild-captured elephants (lines 314-317; e.g. competition with locals in new areas; lines 300-301: long-term stress), we decided to keep some of these translocation references and subsequent discussion in the paper. We have now clarified that we discuss them in the long-term downstream effect context, rather than in terms of immediate mortality during the capture and transport operation.

Line 122: second clause appears to refer to males and females, but not explicitly stated

We have now clarified the sentence, lines 121-124.

Line 122-130 is a summary of data and methods – this is not typically included in the intro. Line 130-137 provides a results and concluding statement not typically in intro – This paragraph reads more like material that would normally be presented in the abstract.

We thank the reviewer for these comments, however, these parts were added to meet the journal's requirements.

Results:

Line 155-162: Why parameterize birth cohort as 0 mortality initially? What is sensitivity of this parameterization – i.e., could it drive result of differences between captive and wild? See comments on methods about concerns with approach

This is a misunderstanding: this parameter was not parameterized at 0 but instead, it was estimated at ~ 0 by the fitting procedure. The near zero estimates imply that for the best cohort and location the mortality is adequately predicted by a model which only contains age dependent mortality terms. We have now made this more clear by indicating in the legend of supplementary table 2 that there are more generally the two possible sources of zero (estimated as such or parameter constrained to be null). We refer the reader to Table 1 to discriminate between these cases based on the model structure. No sensibility analysis is needed because the model averaging includes a model with no parameter constrained to zero and it was possible for such a model to be the best.

Line 184: Is recovering the correct term? Seems like you are describing filtering of individuals not able to handle captivity? The model outputs seem to be homogenized by approach. It would be more valuable to see raw data here rather than model estimates without any error estimates.

We have rephrased this result and no longer mention “recovery” (lines 197-198). We do not see however how raw data could illustrate our result nor how they could inform on the distinction between selective disappearance or true recovery: raw data would just show that some elephants die while others survive, this is a binary event. We recall that instead we now have added error estimates (please see our response to Reviewer 1).

Discussion

Line 256: This is very specific to elephant context. I think the intro and discussion attempts to generalize these results to broadly. Elephants are such a unique species in many respects, that I am not sure how generalizable these results are to other species that are not similar in size, feeding ecology and requirements and social structure.

We thank the Reviewer, we have now modified the paragraph accordingly (lines 273-277).

Line 266: detrimental may not be correct word

We have now reorganized the paragraph and removed word detrimental from here (lines 285-286).

Line 298-302: The comparison is problematic. Social and demographic costs may differ across age classes, such that one manifests for one group more than others. Ultimately, mortality is a powerful measure of costs, but young may be impacted in other ways.

We thank the reviewer, we agree with this point. We have now clarified that the effects we are talking here are the survival costs for elephants (lines 323-325).

Line 329: This is a critical point. It is likely capture and successive mortality is selecting for different individual characteristics— acting as a type of filter such that only those that persist can survive in captivity. The captive born have already been selected for traits conducive to captivity given their mothers survived and successfully breed.

We agree with the Reviewer and that's why we have included this section into the discussion about selective disappearance. We also say twice in the same paragraph that the animals may have been differently adapted to captivity (lines 353-356, 361-363). We had not previously considered that the effect of selective disappearance could cross over to the next generation. This is indeed an interesting idea. Unfortunately, demographic data are not sufficient to assess the extent to which selective disappearance could exert effects both within and across generations. While close monitoring of the physical and mental health of elephants could help to disentangle between selective disappearance and true recovery after capture, establishing whether captive born elephants have been influenced by the capture event undergone by the mother would be very difficult. The hypothetical experiment would be to release some captive mothers into the wild, track their offspring, capture them and compare how those handle the stress of capture relative to those captured from mothers from which the lineage has never been captured. This cannot be done in practice.

Line 337: Not sure robust is the correct word – really looking at traits conducive to captivity

We thank the reviewer, we have modified the sentence (lines 361-363).

Line 343: can you be more explicit about confounding variables. There is much more heterogeneity in caretaking than presented here. Certainly, all handlers are not the same and this can impact stress, the breaking process varies, etc.

This is true but the goal is to highlight here (lines 366-369) why our two groups (captive born and wild caught) are more similar than, say, comparing zoo elephants with wild ones. It is thus deliberate that we focus on similarities and not on the myriad of factors that could still differ between the two groups. Besides, each elephant has one handler, so we don't see how this could create systematic differences between the two compared groups. The confounding variables that we do address are listed in full in the Methods section.

Methods

The first two paragraphs are narratives that could be greatly condensed or removed – simply refer to this historical context.

We thank the reviewer. However, we feel that we have to describe the population and data structure so readers can understand what measures we have and where the data comes from.

Line 396: What is meant by natural – impression from wild data suggest breeding of captive may be depressed in general, so not equal to natural rates

We mean here that the breeding rates are natural in the sense that breeding rates are unmanaged by humans as elephants are not aided in mating or calving. We have now added clarification to the sentence, lines 423-425.

Line 415: Aging from dentition is more accurate and could be accomplished for survivors.

We agree, and elephant capturers and handlers do indeed also consider dentation among other traits (all elephants are easily trained to open their mouths for inspection).

Line 437: The “breaking” time would be a really valuable covariate to include in the models looking at different capture methods. Concern remains about the influence of initial capture conditions on longer term survival.

We fully agree but unfortunately, we don't have that kind of data available on the duration of taming procedure for each historical elephant. Our models are able to adjust for the capture method used, capture year and the estimated age at capture.

Line 447: remove “also”

Removed (line 473).

Line 454: what is meant by role as subordinates? To people?

Yes, we have now clarified the sentence, line 480.

Model averaging has been controversial in the literature of late. I could not find the model covariate estimates and CI for the top model and closely ranked models in the SI. These should be presented in the SI.

We have now reported estimates for the best model, provide the CI on model averaged estimates (see Supplementary Table 1) and provide the SE for each parameter estimate in each model in Supplementary Table 3.

It was not clear to me if the capture technique was informative on mortality, though the authors have worked to include it (seeing raw survivorship and covariate estimates would be more informative).

The effect of capture is very clear (Figure 2, Figure 4, Table 1, Supplementary Table 1, 2, 3) and on par with the effect size of sex. The effect of the capture method is more uncertain as indicated by the similar AIC values between models distinguishing between methods or not. The figure 3 already presented raw data about capture methods. Showing raw survivorship curve would fail to account for the fact that different capture methods have targeted elephants captured at different ages and from different region and birth cohorts, which our analysis accounts for.

The approach used is an interesting analytical framework, apparently used in human literature. Coming from the wildlife conservation disciplines, I am not familiar with the approach. My concern is that it appears the authors were unable to derive error estimates on their mortality estimates.

*The Siler's model on which our method is based has been used in several non-human studies including wildlife (e.g.: Stolen, M. K. and Barlow, J. (2003). A model life table for bottlenose dolphins (*Tursiops truncatus*) from the Indian River Lagoon System, Florida, USA. *Marine Mammal Science* 19(4): 630-649.; Hayman, D. T. S., & Peel, A. J. (2016). Can survival analyses detect hunting pressure in a highly connected species? Lessons from straw-coloured fruit bats. *Biological Conservation*, 200, 131–139.). We have now addressed the issue of error estimates and provide them in Supplementary table 1 & 3.*

If my interpretation is wrong, it would be valuable to have the error presented in the figure (Fig 1, 2 and 4).

We have gone through considerable effort to provide uncertainty estimates on parameter estimates (Supplementary Table 1 & 3). Projecting them onto predictions would be a hard problem which requires to develop new statistical methods and that is beyond the scope of this paper.

We would like to recall that we had starting this paper by a much simpler approach but reviewers called for something significantly more complex. We do not wish to increase complexity ad infinitum.

Besides, on figure 4 it would not be possible with any approach to add errors because those are averaged survivorship curves across elephants spanning different covariate values as indicated in legend of the figure.

If not, I wondered if using a Cox regression approach would provide more direct inference? This would allow the same analytical framework – regression with covariates – but provide insight to uncertainty. If Cox regression is used, I would suggest building different models for the different capture approaches and comparing survival directly.

As mentioned above, we now provide measurements of the uncertainty with our method. Generally speaking, there are reasons why we did not rely on classical survival analysis such as Gompertz (see former review) or Cox. And the same reasons justify why we would prefer not to add in our paper the results of crude survival analyses and lengthy out-of-scope discussion of why they are not appropriate.

The main reason is that the data and the question are quite complex and call for non-mainstream methods whichever approach one wishes to use. We discussed why the Gompertz model is not appropriate in the rebuttal letter of the previous version of the paper and the reviewer 1 seem to have agreed with our arguments.

We will thus now discuss why the Cox approach is not a good alternative to our methodology. There are two main reasons. First, the Cox proportional hazard model does not allow to study age specific mortality when data are both right and left censored. So one

can either dismiss the effect of left censorship or dismiss a time varying effect of age (we do the latter below). Omitting left censorship would produce very biased estimates of survival among young elephants. Omitting a time varying effect of age imposes huge limitations in our context (e.g. we cannot measure the effect of the age at capture).

Second, current implementations of the Cox model do not handle well covariates that do not concern all individuals (represented for example as an interaction without main terms). Since the time since capture cannot be defined for captive born individuals (and considering they have been captured at birth would not help because of the collinearity with age). Because of this, a single model including all elephants cannot lead to the estimation of all estimates we need, which makes comparison between the two group (captive versus non-captive) problematic and estimation of the difference between them with appropriate prediction covariance impossible.

For more information, here are the results of a Cox analysis on our data (dismissing an explicit time-varying effect of age):

Call:

```
coxph(formula = Surv(time1, time2, status) ~ CaptureMethod +
  TimeSinceCapture:CaptureMethod + Sex + strata(Region) + strata(BirthCohort) +
  cluster(ID), data = SurvEles)
```

	coef	exp(coef)	se(coef)	robust se	z	p
CaptureMethodIMM	1.11839	3.05991	0.11683	0.12373	9.04	< 2e-16
CaptureMethodMILARSHI	1.01566	2.76120	0.22603	0.24037	4.23	2.4e-05
CaptureMethodSTOCKADE	0.74247	2.10112	0.09755	0.09968	7.45	9.4e-14
Sexmales	0.34169	1.40732	0.04518	0.04572	7.47	7.8e-14
CaptureMethodCAPTIVE:TimeSinceCapture	NA	NA	0.00000	0.00000	NA	NA
CaptureMethodIMM:TimeSinceCapture	-0.05738	0.94424	0.01206	0.01316	-4.36	1.3e-05
CaptureMethodMILARSHI:TimeSinceCapture	-0.04931	0.95189	0.01540	0.01644	-3.00	0.0027
CaptureMethodSTOCKADE:TimeSinceCapture	-0.02718	0.97319	0.00542	0.00553	-4.92	8.9e-07

Likelihood ratio test=173.6 on 7 df, p=<2e-16
n= 83026, number of events= 2259

Using the predictions from a slightly modified version of this model which dismisses the effect of time since capture (which when considered prevents the function predict.coxph to work, because there is no estimate for captive individuals), we reproduced the figure 4 and obtained results very similar to ours:

It is also true that we can use the model shown above to test the effect of the method of capture:

```
> AIC(fit) ## AIC of the fit above
[1] 20622.37
```

(Notice that the AIC shows that our method fits the data better than this approach)

```
> AIC(update(fit, . ~ . - CaptureMethod - TimeSinceCapture:CaptureMethod + Captured +
TimeSinceCapture:Captured)) ## AIC of a model with Capture but no capture method
[1] 20624
```

As you can see, as in our paper, the Cox method shows that the effect of the method of capture is uncertain (the delta AIC is lower than 2). This is also confirmed by prediction plots performed after refitting the model without the captive born individuals to allow the estimation of the effect of the time since capture (fine grey lines represent the results for each combination of sex and capture method, bold line represents a given combination with CI in grey around it):

All of this thus confirms that the results produced by the Cox analysis are in agreement with the ones produced by our methodology. It does not solve the uncertainty stemming from the data which is that the effect of the capture method is unclear. We have thus now briefly mentioned in the paper that a Cox survival analysis led to results in agreement with our approach (lines 230-233).

However, if using Cox allows to retrieve some of our results, many other results cannot be investigated using the Cox approach. We cannot use the Cox model to measure the immediate effect of capture, the delayed effect, the speed to which the increase in mortality returns to the baseline and, as mentioned above, we cannot even compare captive versus non-captive individuals in the same model when time since capture is considered.

We could of course re-implement the whole Cox tool box from scratch to cope with some (not all) limitations but other issues would remain despite the work that this would represent. Also, the Cox model above does not fulfill the assumption of proportional hazard for 8 estimates and it leads to a poorer fit of the data than our method.

Also we recall that we have already worked out two nice original ways to analyze our data (one based on GLMM corresponding to the first version of the paper) and one based on the Siler's model (current version). Both of those methods supported the same findings and produce the elements necessary for our discussion. Both have limitations too but it would

also be the case for any alternative method. We thus hope that the reviewer will consider that this is sufficient and recognize that the methodological effort behind this paper is already way beyond the norm. We also hope that the fact that the highly similar results that can be obtained from the three different methods we tried show that our results are particularly robust.

While I am not familiar with the approach used, I am concerned that the model structure, i.e., the parameterization of the three life stages, results in a homogenization of the mortality rates across subgroups. The assumptions required for this three stage model parameterization should be assessed in a sensitivity analysis (i.e., if you do not assume 0 infant mortality as the baseline, do results shift?).

If we had defined three age categories with age thresholds, then, we agree that a sensitivity analysis would be called for. But again, no sensitivity analysis is required because there is no such thing as the definition of three life stages. There are three main mortality components for which the names labelled by Siler refer to different life stages but it is important to understand that at any age an individual is subjected to all three mortality components. It only turns out that the weight that each component bears on mortality varies with age (e.g. the first one impacts more young individuals than old ones) but the transition is continuous, not discrete. This is clear from equation 1, but we have now mentioned it as text on lines 541-543.

I would prefer to contrast survival curves for each group directly, with error on the survival estimates. However, I realize the authors have already put substantial work into this analysis and approach and there may be perfectly solid arguments to proceed with the approach employed. At a minimum, I would suggest presenting survivorship curves for the capture methods and captive born individuals.

We are already providing the survival curve for each capture method and captive individuals on figure 4. As mentioned above the error cannot be computed for such curves. We would prefer not to provide the curve shown above with CI because as mentioned, a Cox alternative is problematic in many ways.

Fig. 1, 2, and 4: Could you present the raw data rather than yearly average mortality rates and model predictions? This would offer better insight to the variability in the data.

Raw data would be only zeros or ones and would thus offer no insight what-so-ever. Proportion (yearly average mortality rates) would be also uninformative because the number of deaths for each "age-capture category-time since capture-sex-birth location and cohort" is usually zero. Then one would need to start lumping different categories and different ages together which is going toward replicating model prediction in a more inaccurate way. Only the figure 1 allows for us to represent raw data (albeit imperfectly as predictions are controlled for effects of birth location and cohort but raw data are not split among these categories) and we have already done so.

Supplementary material: I could not find a supplementary material file besides the two included tables. The lack of information makes it hard to understand the two tables. I assume

table 1 is the parameter fit for the models, but CIs are missing. I could not understand table 2.

The legends of the supplementary tables are located in the main text. Our belief is that upon acceptance the journal will join those legends to the tables they refer to. CI are now included in Supplementary Table 1 and underlying SE for each model in in Supplementary Table 3.

Several reviewers ask to see the raw data used in the analyses. The authors have only presented the data in Fig. 1. It is important to have the raw data in the other figures, particularly given the authors were not able to estimate confidence intervals in the presented predictions. Otherwise, readers cannot get an idea of uncertainty in the results presented.

We repeat the justification of the raw data on some figures: the model comparison based on AIC readily provides some information about uncertainty; we have also now added CI on parameter estimates; presenting raw data for binary events is not informative and representing probabilities independently derived from particular model estimate is inadequate because it neglects covariances between parameter estimates..

Discussion of difference in captive and wild caught breaking does not explicitly state that wild-caught tend to go through a much rougher process. It would be valuable to mention some of the mechanisms causing the higher probability of complications for wild-caught individuals. These certainly could relate to some of the sex differences recorded, even if there is a biological difference in survival between the sexes that is unrelated to capture. The reduction of the emphasis on capture methods given the results and lack of covariate data is sufficient.

We say in lines 475-478 that wild-caught elephants go through a harder process than captive-born elephants and they are more prone to suffer from psychological and physical trauma during taming. However, we did not find any sex-differences here in relation to capture effects. We are happy to hear that the handling of capture method results is sufficient.

Reviewers' Comments:

Reviewer #1 (Remarks to the Author):

This revised version accounts for most problems of the previous version. I am convinced by the survival analysis based on the Siler-modified model and I have no doubt about the robustness of the findings (especially obvious when using the simplest metrics). However, there are still some contradictory statements in the manuscript and some errors that render the work difficult to follow.

I have thus two major concerns.

From the table of model selection, the model 10 should be selected (lowest AIC and lowest number of parameters, see e.g. Arnold et al. 2010 JWM). However, if we believe the legend, this model does not seem to include any influence of capture on survival parameters (although the Table S2 shows the contrary!). Moreover, still considering the information provided in the legend, 3 (model 10, model 9 and model 6) out of the 4 best models did not include any effect of capture on survival parameters. If this is correct (but I guess it is not!), it seems quite obvious that the statistical evidence supporting an effect of capture is weak at the best. One also could wonder whether intermediate models between Model 2 and Model 10 (i.e. models including a capture effect only in w4, only on w5, and only on b4 would be better than Model 10. From this table, we can really wonder about the statistical support for an effect of capture!

The second problem is that the confidence intervals reported in Supplementary tables do not help to conclude about the statistical support of capture effects. There are indeed obvious problems in the Table S1. For a substantial number of parameters, the mean estimate of parameters does not belong to the confidence interval (see e.g. w1.males, b1.males, w2.females, ...), which does not make any sense.

The authors should thus correct the problem when estimating confidence intervals and state more accurately what are the parameters included in each model so that readers can follow easily this great work.

In addition, I have two minor points:

l. 616-617: Make explicit that this corresponds to what is commonly described as model averaging.

l. 621-623: Simply state that you calculated AIC weights. There is no need to describe the formula (i.e. remove: « the exponential of half of the negative difference in AIC between a given model and the best model, and then rescaled the metrics obtained so that the total sum of weights across all models equals to one »)

J.M. Gaillard

Comments from Reviewer #3 can be found in the file attached to this email

Point-by-point responses to reviewer comments

We have revised our manuscript according to the reviewers' comments as outlined below. Our newest comments are in blue font.

REVIEWERS' COMMENTS:

Reviewer #1 (Remarks to the Author):

This revised version accounts for most problems of the previous version. I am convinced by the survival analysis based on the Siler-modified model and I have no doubt about the robustness of the findings (especially obvious when using the simplest metrics). However, there are still some contradictory statements in the manuscript and some errors that render the work difficult to follow.

We are extremely thankful to the reviewer for his close reading of the manuscript. We have now corrected all the problems he noticed.

I have thus two major concerns.

From the table of model selection, the model 10 should be selected (lowest AIC and lowest number of parameters, see e.g. Arnold et al. 2010 JWM). However, if we believe the legend, this model does not seem to include any influence of capture on survival parameters (although the Table S2 shows the contrary!). Moreover, still considering the information provided in the legend, 3 (model 10, model 9 and model 6) out of the 4 best models did not include any effect of capture on survival parameters. If this is correct (but I guess it is not!), it seems quite obvious that the statistical evidence supporting an effect of capture is weak at the best. One also could wonder whether intermediate models between Model 2 and Model 10 (i.e. models including a capture effect only in w_4 , only on w_5 , and only on b_4) would be better than Model 10. From this table, we can really wonder about the statistical support for an effect of capture!

This is a misunderstanding: the Table 1 legend states "The symbol '1' indicates that a single parameter value was estimated for a given model meta-parameter", thus the model 10 does include an effect of capture via its parameters w_4 and w_5 . It is just that the effect of capture is considered to be the same irrespectively of the capture method, but parameter values differ between captured and captive born individuals. Similarly, model 9 and 6 do consider the effect of capture but not the effect of the capture method.

While the effect of capture is strong and clear, it is true that the effect of the capture method is unclear. We have obtained this finding after applying each of the three statistical framework we employed (GLMM, Siler and Cox) as

mentioned in all previous submitted versions of the paper. In the current version this result is still clearly written: “Our model comparison reveals an important model selection uncertainty which prevents us from concluding unambiguously about the possible long-term differences between the capture methods in affecting mortality risk”, “the different capture methods have relatively similar effects on long-term mortality”, “all capture methods are associated with increased long-term mortality in both males and females in a similar way”, “the differential effect of the capture methods appears weak”...

To clarify further the distinction between the effect of capture as such (which is strong) and the effect of the capture method (which is weak), we have now reworked the sentence on lines 208-211: “We found some weak evidence that the clear increase in mortality associated with capture and taming actually differs depending on the capture method; this suggests that whichever method was used to capture an elephant had little influence on its survival past the event of capture and that all methods were associated with a similar long-term mortality cost.”. We have also introduced a new sentence in the method section (lines 572-576): “Only models for which those 4 meta-parameters were considered as null did not account for the effect of capture. All the other parameterisations do account for a possible effect of capture. Hence, meta-parameters w_4 , w_5 , b_4 , and b_5 with subscripts ‘1’ and ‘s’ in Table 1 do consider the effect of capture as such but not differences between capture methods”.

The second problem is that the confidence intervals reported in Supplementary tables do not help to conclude about the statistical support of capture effects. There are indeed obvious problems in the Table S1. For a substantial number of parameters, the mean estimate of parameters does not belong to the confidence interval (see e.g. $w_{1.males}$, $b_{1.males}$, $w_{2.females}$, ...), which does not make any sense.

The authors should thus correct the problem when estimating confidence intervals and state more accurately what are the parameters included in each model so that readers can follow easily this great work.

Apologies. There was indeed a mistake in our Supplementary Table S1. Although we had estimated the estimates and associated SE correctly, we had made a mistake in the code generating the Wald confidence intervals. Those intervals are being computed as estimates $\pm 1.96 * SE$ and we had made a mistake by calling the wrong function ($pnorm$ instead of $qnorm$) to compute the precise value of the 1.96 (i.e the 0.975 quantile of the standard normal distribution).

The new sentence aforementioned, which we included in the method section, should now allow the reader to track which parameters are being used in each model from the reading of table 1.

In addition, I have two minor points:

I. 616-617: Make explicit that this corresponds to what is commonly described as model averaging.

Done (lines 640-642)

I. 621-623: Simply state that you calculated AIC weights. There is no need to describe the formula (i.e. remove: « the exponential of half of the negative difference in AIC between a given model and the best model, and then rescaled the metrics obtained so that the total sum of weights across all models equals to one »)

We removed the sentence.

J.M. Gaillard

Reviewer #3 (Remarks to the Author) and responses back

The two main issues still remain: the authors still seem to claim in some places (but without evidence) that their results can be applied to other wild caught species. In response to our comments about this, they really haven't given any specific/justified examples of which other species/circumstances where these results are relevant. The authors also continue to state that they assessed the effects of capture, rather than capture *and* taming/breaking. Thus even though they acknowledge the possible influence of taming [see their response to another referee for example], their interpretation still revolves around 'capture' effects ... presumably to support their claim that these data are relevant beyond elephants.

We thank the referee for the in-depth concern in these issues. We have now reduced any generalizations in the whole ms to satisfy the Reviewer and Editor. Please note that our opening paragraph of the Introduction simply introduces the state-of-art in the field and states the well-evidenced fact that thousands of species are captured each year but the long-term effects of capture have rarely been examined in any species; this paragraph does not in any way discuss our obtained results or aim to generalize them. Animals captured from the wild experience all kind of handling and management practices after capture varying for example between and within species, places or humans involved and whether the animals are tamed or not. This

strengthens the need for capture effects and related changes in handling to be studied in detail in the future also in other species besides elephants.

Beside these important and well-justified points that we wish to make when outlining the need for our study, we now merely refer to other species to suggest that similar work should be done in other species (e.g. lines 270-273 in the beginning of Discussion: “For example, such findings emphasize the need for animal welfare specialists, veterinarians and ecologists to identify the consequences of wild-capture on the success of individuals and populations and WHEN RELEVANT, to improve conservation and management practices so as to best support captured animals during the most critical periods after capture.”). We also say that if studied, these effects could potentially be seen especially in species with slow life-history, lines 405-408. This point was requested to be added by Reviewer 1. We recall that Reviewer 1 said during the first round of revision that these effects could be mostly detected in these kind of species which also benefit less from captivity in terms of their longevity and in which wild-born individuals have been detected to have higher mortality compared to captive-born individuals (we cite these studies too in lines 407 & 289. Reviewer 1 comment: “Lastly, elephants are especially slow-living species, such as the other case studies mentioned on l. 212. It might thus be that the negative influence of survival reported here could be mostly observed in species with very slow life histories. This needs to be discussed (see Tidiere et al. 2016 Scientific Reports for recent evidence of the effect of the pace of life on the response of age-specific survival patterns to environmental conditions).”.

Please see the next point for a detailed response concerning the capture/taming issue and how we have now taken this point into account in the revised version of the ms.

When Asian elephants are caught to work in the logging industry, the capture and move to captivity are then followed by several weeks of intense and harsh taming/breaking/training. The authors do acknowledge this, and one of the other referees highlights it too. However, the authors downplay this in the language they use (referring only to the effects of ‘capture’ throughout their paper), and also in the over-zealous way they claim that their results apply to the members of other species caught from the wild (in the Abstract, Introduction and Discussion). The authors repeat these claims about generality to other species several times (e.g. twice in the Abstract alone), and yet are necessarily vague with them too, as it’s really not clear that any other types of wild animal undergo anything like this (do they? If yes be specific). So, we would request two types of editing to acknowledge the uniqueness of the ‘Asian-elephant-captured-to-work’ scenario. First, please only suggest that these results apply to other species once and once only (not repeatedly), and also make it clear that this is just a hypothesis. Second, please use the phrase ‘capture and taming’ (or similar) in the Abstract, Discussion and all sub-titles (e.g. at lines 164 and 194) throughout the paper,

as it is clear that the effects of capture per se cannot be parsed out from those of taming/training, and so the two must be presented as a package (such that calling these 'capture effects' is simplistic and potentially misleading).

While we agree with many suggestions by the reviewer, it is important to recall that captive-born elephants also undergo taming with the same aim and objective as the wild-caught elephants: to obey human commands and work efficiently in timber industry. Captive-born elephants are tamed around age 4-5 years generally using the same personnel, infrastructure and methods as is used for the wild-caught elephants. Before that, calves are not used to human-handling, commands or caretaking and could as well be considered as being "wild". This fact makes our study so unique, and also means that the alternative terminology proposed by the reviewer is not suited to our study design. It is true that the taming and breaking are harsher for wild-born individuals to a degree that also depends on the age, sex and personality. We say this explicitly in the paper (lines 133-137, 462-465, 475-478) and cite an article (line 465) which gives details about the taming methods used for captive-born elephants: Zaw Min Oo 2010 The training methods used in Myanmar Timber Enterprise 2010 Gajah 33:58-61). It is noteworthy that the taming can also be harsh for the captive-born calves, regularly leading to accidental deaths as recorded by the stated causes of death (see lines 311-314 and citation to Mar et al. 2012 PlosOne 7(3):e32335). Also, due to the intense training, the baby elephants may get wounds, injuries, and stress. We say on lines 60-64 (second paragraph of the whole article) that taming/breaking can have effects on post-capture lifespan and there may be differences in the management of captive-born and wild-born animals. We also say now in the end of the introduction, lines 137-139, that part of the effects seen in wild-caught elephants and captive-born elephants may be due to differences during taming period and from now on we refer to all these effects as capture effects. Just to be sure, we repeat this in each section (Results 182-183, Discussion 297-300, 309-314, Methods 462-465, 475-478). For all these reasons and for keeping the wording as light and lively as possible we prefer to stick to the term "capture" instead of using the "capture and taming" which we find more misleading than our original formulation.

It is very good that authors acknowledge the potential confounds between capture and taming/breaking. But it is problematic that they then still present the results as though they were truly able to separate the capture and taming effects in their current study. Whether intentional or not, this implies that capture plays a bigger role than taming and the other acknowledged factors, and there is no evidence to support this.

This is incorrect – we do not state anywhere in the ms that we are able to separate effects of capture and taming. On the very contrary, we state on lines 139-140 in the Introduction that any differences between wild-caught and captive-born elephants may stem from both the effects of capture and differential taming methods. In the Discussion we also explicitly say that our finding of high mortality increase after capture might be partly because of harsher taming and breaking in wild-born elephants (lines 312-315: "The

highest mortality increase in the year following capture is likely to be mainly related to capture-related injuries and trauma, as well as the subsequent harsh taming and breaking causing some of the recently captured animals that survived the capture-operation itself to die within a short time after entering captivity.”). In the same paragraph we also say that the older wild-captured elephants are likely to be subjected to harsher taming which could have impacted the finding that older wild-caught elephants have higher mortality after capture than younger ones. What comes to the effect of capture vs taming, we are unable to measure which factors play the biggest role, but as mentioned now repeatedly, both groups go through taming that is documented to increase mortality also in the captive-born elephants.

We are unclear why this referee continues to raise this issue despite it being dealt in the manuscript in several points, but to settle the matter we have now clearly labelled our results as being effects of capture and taming (lines 180 & 215-216). We have also included a specific section (“Capture and taming”) in the methods describing the capture and taming processes to aid readers with achieving full understanding and interpretation of our data and results. Also, we have now changed many sentences concerning our results in the whole ms to say “capture and taming” instead of only saying “capture” (see e.g. line 20-23 in the Abstract, also lines 97-99, 294-296, 402-404, Figure 2 and 4 legends). As we talk a lot about taming and management practices in our population in wild-born and captive-born elephants in the Discussion, it should now be obvious to the readers that the results may stem from both causes acting in concert (capture and taming differences between wild-borns and captive-borns).

(And lines 348-350 appear to be the only area of the MS where the effect of capture is separated from the effects of training and captivity, but the authors call it “partial evidence” and “weak”, which further makes it seem poorly justified to use “capture” and “capture effects” throughout).

We have deleted this problematic and unclear sentence, thanks for pointing it out.

Also, be careful about implying that the process is similar between wild-caught and captive-born animals; as here is a quote from one of the authors a few years ago (Mar, Khyne U. "Birth sex ratio and determinants of fecundity in female timber elephants of Myanmar." *Gajah* 38, 2013): 8-18):

“Wild-caught females take some years to recover from the stress of capture and/or taming so that their reported age of first calving in captivity is older than captive-born females. For captive-born calves, breaking normally takes only a few days, whereas the breaking period for wild-caught animals lasts a minimum of 2 weeks to a maximum of 8 weeks depending on temperament.

The longer the taming process, the harsher the punishments and more unpleasant the process”

So, we repeat: please use the phrase ‘capture and intensive taming’ (or similar, e.g. ‘capture and breaking’) in the Abstract, and in all sub-titles, when referring to what happens to wild-caught animals. This will keep things clear and honest, while not impeding the flow of the writing.

As mentioned above, we have now clearly labelled our results as being effects of capture and taming, including Abstract and sub-headings (when relevant) as proposed by the Reviewer. In addition, we have included a specific section in the methods describing the capture and taming processes to aid readers with achieving full understanding and interpretation of our data and results. Regarding the quoted remark, this is unfortunately incorrect. Please see Min-Oo, Z. The training methods used in Myanmar timber enterprise. Gajah 33, 58–61 (2010) for a more comprehensive description of taming methods used for captive-born calves and how long each step of the taming process lasts. In total “The duration of the training period in all methods is about one month.”. We cite this article on lines 135, 490, 492, 499 to provide the readers this background.

We are somewhat puzzled why the Reviewer has the perception that we claim the results to apply to other species as such. We have carefully checked the ms and have found no evidence of such claims. Nevertheless, to settle this argument, we have now aimed to be more accurate with the sentences saying how our results apply to other species. In the abstract we do not claim that the results apply to the members of other species caught from the wild (and did not say previously even one time), instead we say that our results are timely because elephants as well as many other animal species are still being captured to supplement captive populations even though there have been alarming declines in wild populations globally. In the introduction, we did not repeatedly make this claim, but we said in the end of the introduction that “The long-term differences between captive-born and wild-captured animals shown by our study are currently rarely considered in research and conservation programs”. Now we have changed this sentence to concern only elephants. In the beginning of discussion, we say that “Large numbers of animals across a range of species are routinely captured from the wild for diverse purposes, but surprisingly little is known of the consequences of such experiences for the subsequent long-term performance of those individuals”, which is true. In the same paragraph we say that these results have more general implications and are timely (although our population is a very unique case) but have modified the sentences to say exactly how and that also more elephant studies are needed from different contexts (translocation and zoo populations too) “Although capturing elephants for logging industry is a very restricted form of exploiting wild populations, our results have general implications beyond the study system. For example, such findings emphasize the need for animal welfare specialists, veterinarians and ecologists to identify the consequences of wild-capture on the success of individuals and populations and when relevant, to improve conservation and management practices so as to best support captured animals during the most critical periods after capture. Our results also stress the need for detailed further studies in elephants to identify potential costs of wild capture in different contexts (such as translocation and zoo

environments) and more specific studies on taming/training related mortalities (and injuries) or long-term effects of sedatives.” In the last paragraph of discussion we say that capture effects should be studied in other animals too to find out the capture-related as well as long-term effects and also say that ignoring the potential effects MAY lead to erroneous conclusions in some studies. “Although our study population is unique and the use of elephants for logging is not a situation that applies to many other species, capture of elephants continues for legal or illegal purposes e.g.33-35 and capture of various other species from the wild is practised for diverse purposes each year e.g.1,2,63. Therefore, our results are timely and have three main implications. First, long-term effects of capture are currently not considered in research design and conservation programs, but our results show that capture can negatively influence animal performance for several years at least in species such as elephants. Therefore, using wild-captured animals to supplement medical trial populations e.g.64 or as reference groups for species-typical parameter values e.g.3,4 may lead to erroneous conclusions and both immediate (capture-related) as well as long-term effects of capture should be taken into account in further studies.” Finally, in the end of discussion, we say that our results imply that capture is costly for elephants and POTENTIALLY more generally among species, especially among species with slow life-histories, which have been shown to also suffer more from zoo environments or in which captive-borns have shown to have higher survival than wild-borns. “Our study implies that capturing wild individuals in elephants and potentially more generally across species (especially among species with slow life-histories11, 48-50), is costly for individual longevity and alternative methods should be sought to boost captive populations in order to avoid further capture from endangered wild populations.”.

The authors still start both the abstract and the discussion with a sentence about other species *and* then also clearly say that results can potentially be generalized to other species (in highlighting above). Lines 410-414: The final sentences of the discussion also refer to generalizing results across species.

Our current abstract starts with statement: “Wild-capture of numerous species is common for diverse purposes, including medical experiments, conservation, veterinary interventions and research, but little objective data exists on its consequences”. This is not controversial but a well-documented fact and does not anyhow generalize our results. We have thus kept the sentence as it is.

Similarly, our discussion begins with the statement: “Large numbers of animals across a range of species are routinely captured from the wild for diverse purposes, but surprisingly little is known of the consequences of such experiences for the subsequent long-term performance of those individuals.” This is also all true and well-evidenced (see the Introduction and references from the first paragraph). Nevertheless, we really wish to see our work published now and we have removed this sentence totally following the reviewer remark, and we go straight to elephants in the Discussion.

As we explained above the last sentence of Discussion was added in response to Reviewer 1 requesting us to address this issue during the first round of revision. We have therefore kept it here as it does not state that our effects ARE similar in animals with slow life-history but that there could POTENTIALLY be similar effects in these species (which also suffer more from captivity than many other species).

See also the section of their response highlighted in green below. This does not seem well supported, for reasons we've already raised. Which other species for example? Do the authors have specific examples of cases where wild animals from other species are caught and then 'broken' for several weeks? Maybe raptors? Wild horses? Or orcas??

We are afraid this is based on confusion. The opening paragraph concerns the well-evidenced fact that thousands of species are captured each year. The paragraph does not aim to provide evidence how and why such capture may influence animals in the long-term (be it direct effect of capture or related breaking, for example) – as we state such long-term effects of capture have rarely been examined in any species and we simply do not know. As we say above, animals captured from the wild experience all kind of handling and management practices after capture varying for example between and within species, places or humans involved and whether the animals are tamed or not. The paragraph provides the state-of-art in the field for the current study. It does not generalize our results – indeed our results are not even presented yet at this part of the article.

It's also clear from the zoo studies (see below) that the effects of early experience can be the opposite to those reported here, with captive-born animals instead sometimes being at most risk of an early death. Thus taken as a whole, the fascinating corpus of data from diverse elephant populations reveal the profound potential effects of early experience on lifetime survivorship. Capture and breaking have longterm adverse effects, but so too does being zoo-born. Both of these patterns highlight the importance of looking for similar effects in other species. But to suggest that capture per se might confer risks on other species is clearly weirdly narrow and myopic.

We could not agree more with the referee – please see lines 66-67 and 421-423 about early-life effects and 309-310 for need of further studies in other animals.

Second, how the authors compare and relate these to other elephant populations still needs some work. It's these comparisons that are most interesting and relevant, and yet the authors do not do them justice. Other elephants are not even mentioned in the opening paragraph of the Introduction for example!

We have now been more precise about what implications our results have for different elephant populations (e.g. lines 277-280, 386-399). Note however that our focus isn't in zoo populations or translocated populations (small minority of elephants kept in captivity) but we cite these studies because there aren't many studies available on other similar populations of elephants (for example studies on wild Asian elephants are very rare). Concerning the opening paragraph, we prefer to keep it focused on the capture of animals in general and the shortage of long-term "Our study implies that capturing wild individuals in elephants and potentially more generally across species (especially among species with slow life-histories¹¹, 48-50), studies about capture effects on survival across species. This is probably more relevant for the readership of a general journal. We do talk about elephants later after we have explained how capture effects can impact animals in general.

Thus in terms of the wild-capture of Asian elephants that still happens to this day (e.g. lines 112-113), just how relevant are these data: are today's wild-caught animals still subject to the same breaking and taming processes? Please make it clear. And in terms of the cessation of capture for logging in the 90's/2000, was a sense that wild-caught animals did not fare well part of the reason for the decision? Turning to translocated animals, again are the authors' data really relevant to them, since translocated elephants again are not subject to training and taming? Make it clear. Lastly, comparing these findings to those from zoo populations (and the reference populations used to assess zoo performance), it's here that the authors are perhaps weakest, as will be detailed next.

All animals captured nowadays in Myanmar go through similar taming process which we have described in the paper. We say this in the paper when we say that "All captured elephants go through taming..." (lines 462-465). Also, wild-capture still happens in other range countries too like in Indonesia together with similar taming and training periods (e.g. Mikota et al. 2008, cited in the paper). The reasons for the decision to ban the capture is unclear for us but the reasons aren't however having an effect on our study results or interpretation anyhow.

In line with this Reviewer's suggestion we have now removed one translocation reference but we prefer to keep the rest as they may give insights into the reasons behind higher mortality of wild- captured individuals compared to captive-born (lines 314-317) or show that the effect of translocation can also have long-term effects on animal wellbeing (lines 341-343, 303- 304)(see also our response to Reviewer 4 question).

We want to make it clear that although discussing similarities and differences to zoo elephants is interesting, our main focus isn't in zoo elephants; there are numerous other factors specific to zoo-conditions which are likely to have an effect on elephant survival therein. In Myanmar, wild-caught and captive-born elephants live, forage and work alongside one another, and the same governmental regulations apply for both capture types concerning data recording, workload and rest periods. The elephants are not provisioned, but instead forage unsupervised in forests at night, and the same basic veterinary care is available to all individuals. There are therefore many differences in our study compared to the previous studies with reference populations used to assess zoo performance. Our aim is not to provide specific comparisons but to highlight general discrepancies.

We're not requesting a general comparison between zoo and MTE elephants: we're highlighting that effects of early experience have also been found in zoo populations, but in the opposite direction to those reported here. Thus in zoos, captive-bred animals have shorter lives, while in MTE, they have longer lives. This contrast is surely interesting? One potential reason is that the wild-caught animals in zoos have already survived the breaking period; and another is that captive-bred calves in zoos appear to be over-weight (perhaps not an issue in MTW?).

Yes, again we fully agree and this is explained (within the word limit we have) on lines 85-86 (the difference in our findings compared to zoo populations) as well as more detailed discussion on the topic on lines 408-418. On lines 418-421 we also say that the reasons could be because of higher nutritional plane in zoo elephants or early-life effects, suggested by Clubb et al. in their 2009 paper (cited there). We also say now that early-life effects can have effects on elephants kept in captivity, lines 421-423 and in animals general in lines 59-61.

And thus in terms of the nameless other wild species that the authors desperately want to extrapolate to, it also suggests that sometimes captive-bred animals can be more vulnerable to e.g. health problems than are subjects caught from the wild. Again, surely this is worth acknowledging and highlighting when making suggestions for future work?

We have now considered this possibility too as we say in the Introduction (lines 65-67) that there can be inherent differences (and differences in early-life) between captive-born and wild-born animals, such as susceptibility to diseases, which can further have an effect on their lifespan.

The previous work comparing Asians in zoos with MTE animals (Clubb et al. 2008) found that wild-caught MTE elephants have elevated risks of mortality throughout their juvenile years (1-9 years of age), compared to both wild-caught zoo animals and

captive-bred MTE animals, but such effects were not evident by adulthood (10-70 years of age): an apparent difference to the effects reported here.

We thank the reviewer for bringing up a point which we wish to clarify. We do say in the paper that in captive populations in Asia wild-born elephants suffer from increased mortality compared to captive-born individuals and cite Clubb et al. 2008 and 2009 articles (lines 83-84). We chose to not go into details however, because the analytical decisions made in the Clubb et al. study prevents directs comparisons. First, they used different analyses compared to ours which do not allow as accurate age-specific comparisons. In particular, Clubb et al. 2008 analyzed the effects in separate age categories (infants, juveniles and adults) which leads to juvenile survival appearing better in zoo populations as compared to MTE populations, although infant mortality is much higher in zoos compared to MTE elephants (Clubb et al. Supplementary Table 1). Therefore, the result during juvenile years could at least partially be because of unaccounted selective disappearance (weaker infants dying early in zoo populations and only more robust ones surviving beyond 1st year) and actually more juveniles surviving beyond 1st living year in MTE populations than in zoo populations.

*Second, the issue with adults is that the effects of capture depends, as we show it for the first time, on both the current age of the elephant and the age at capture. While a capture at old age increases mortality, time spent in captivity after capture leads to a decrease in the cost of capture (see results). Therefore, old elephants that have been captured at very young age present only a minor difference in mortality as compared to elephants born in captivity. **Since the Clubb et al. study does not distinguish between these two effects, we can only compare their results to ours at very general level.** The overall finding that MTE captive borns have higher survival than MTE wild caught elephants is actually evident in the Clubb et al. paper and this comparison is also significant (see Supplementary Table 2, WC all mortality vs CB mortality). The comparison is non-significant when comparing MTE wild born mortality from natural causes to MTE captive born mortality but this comparison isn't very relevant to the question posed in our article because considering only natural mortality in MTE wild-caught group, at least in part, removes the capture effects. (This has been done by first estimating that the capture and taming decreases survival up to eight years after capture and then the first eight years of each wild-caught elephant histories were left-censored in the subsequent analyses.)*

We would prefer not dwelling on such differences in study design and analytical power in our article, because it is unnecessary for the important points we want to get across with our study.

Clubb et al. 2008 also found that when animals were transferred between zoos, mortality risks were elevated for 4 years post-move, but after this period such an adverse effect could no longer be detected (obviously pretty relevant to this paper). Thus the authors' implication in their Introduction that their study is the first to follow animals for decades really does need some damping down.

We have actually mentioned this finding in the discussion, lines 333-334 “Interestingly, inter-zoo transfers also reduce Asian elephant survivorship, an effect detected up to four years after the initial transfer³⁶. ”Also, we mention the even more important finding from Clubb et al. 2008 paper that the capture effects lasted 8 years (lines 331-333).

OK good

Concerning the novelty of this study, we have been careful to say on lines 95-97 that our study is the first to study age-specific mortality effects of capture in a long-lived mammal.

Not capture though: capture and breaking

We have now modified the sentence to say “capture from wild and subsequent taming”, lines 98-99.

This is accurate and true, because previous studies do not truly compare age-specific differences in yearly mortalities with confounding time-varying variables such as time since capture effects like we do. We also say in the discussion, lines 339-341, that “To our knowledge, there are no comparable studies investigating capture effects on mortality for decades in any species (but see Saraux et al. 56 on effects of tagging)”. This is because Clubb et al investigated whether the capture effects could last for 0-14 years, not longer (see Supplementary material), and concluded the effect lasts 8 years after capture and did not investigate whether the capture effects differ in duration among younger and older elephants.

True

We investigated the possibility that the capture effects can last decades and that the results can differ with elephant’s age and capture age. Of course they investigated elephants aged between 0-70+ years but this is not the same as studying how long the capture effect lasts and how the time since capture variable modifies the outcome (range in our study 0-45 years). Given the surprising and interesting results concerning age and time spent in captivity, we consider our novelty statement justified and hope that you agree with this.

Furthermore, that “the early maternal environment of those born in captivity is typically different from wild-captured animals” (lines 76-78) is also not special to zoos, but applies to MTE animals too (who are unlikely to be living in true matrilineal groups).

This paragraph (lines 68-81) is specifically focused on explaining the different management factors that lead to higher documented mortality among zoo elephants as compared to WILD elephants, as we state in the second sentence. We are hence a bit puzzled how the MTE animals are relevant here, given they are neither wild nor live in zoos. Yes it is true that MTE animals do not necessarily have all genetically related individuals around as calves compared to calves born in the wild, but the calves are always allowed to be with the mothers until taming around age 4 or 5 and they usually also have at least one or more allomothers.

Their social environment includes several different aged and sex elephants and they can also meet their true relatives when they are released to forests during nights. Also grandmothers are present for many calves increasing their survival (Lahdenperä et al. Sci Rep 2016), and this all makes the early environment in timber camps very different to that experienced in zoos. Here we specifically talk about multi-generational family-groups, which are very rare in zoo environments compared to MTE population. As said, the point of the paragraph is however to illustrate differences between zoos and wild Asian elephants, so we have left this sentence as it is.

OK

In addition, the Clubb et al. 2008 paper (built on further in 2009) found that zoo-bred Asian elephants had much higher mortality risks than wild-caught Asian elephants also housed in zoos. This is obviously completely the opposite to what Clubb et al. 2008 and this manuscript find for MTE animals, and it seems odd to downplay this remarkable contrast with a mere “(vice versa in zoos)” cryptic clause, in parentheses, in the Introduction (line 84). A fuller acknowledgement of this paradox is surely warranted. Why the zoo and MTE population differ in the direction of their birth origin effect is raised at lines 366-368, but almost as a rhetorical question with no credible suggestions offered as to potential explanations. I think the authors can do better than this, especially as the lead author was a co-author of the Clubb et al. 2009 paper that identified heavy infant birthweights (perhaps indicating neonatal obesity) as a significant difference between zoo and MTE captive-bred calves, and that also then suggested the high nutritional plane of zoo elephants as a possible cause of the poor survival of captive-bred elephants in zoo populations. This idea is surely still a plausible, testable one, no?

To address this, we have now expanded discussing the differences in the origin effects in the Discussion.

OK great

We do not want to raise these points in the Introduction, given our own results are not clear to the readers yet at that point and thus the comparison to zoo results cannot be made.

OK

Furthermore, Nature Communications imposes length restrictions on the Introduction which we already exceed. We now say that “In contrast to the situation in our semi-captive Myanmar population, captive(zoo)-born Asian elephants in European zoos have poorer survivorship than wild-captured animals” and cite both Clubb et al. papers (lines 387-389).

OK great

We also say that these comparisons and differences in the origin effects in zoos and timber camps reflect the many problems that zoo elephants face. Please accept that our focus here isn't the zoo population mortality in relation to birth origin effects but rather, timber camp elephant mortalities in wild-caught and captive born animals. We cannot test the nutritional intake of zoo vs timber elephants since our article only focuses on the latter, or of captive-born and wild-born MTE elephants since both forage independently in the forest at night unobserved and no data exists on nutritional status of the historical animals throughout their lives. However we have now added a sentence saying that the different mortality pattern in zoos call for further studies and the higher nutritional plane and stress are two potential reasons for this difference, see lines 396-399.

Minor comments:

The phrase “before and after capture” (line 85) also needs toning down too, since the authors do not assess wild Asian elephants before capture and so should not imply that this is a unique angle of their paper!

Corrected.

Lines 99-100 say that around 15,000 Asian elephants live in captivity, but lines 371-372 say about 1000 are: needs editing to clarify so not inadvertently confusing (I think the latter applies to non-timber animals only?).

We thank the reviewer for noticing these confusing sentences. Yes, we meant that the 15,000 elephants includes all Asian elephants living in captivity in range countries, including timber elephants, elephants in temples and zoos and those owned by private people. The second number includes only those elephants living in zoos, safari parks and circuses world-wide. We have tried to clarify the sentences in the text, see lines 99-101 and 399-400.

Figure 1: make it clear that these are data for all MTE animals?

We thank the reviewer, clarification added to the Figure 1 legend.

OK great.

One last little thing:

Lines 114 & 220 refer to capture of elephants involved in human-elephant conflict. We believe these animals are captured for translocation? But if this is the case they should clearly state it.

No, this is incorrect – to our knowledge conflict elephants continue to be captured, tamed and subsequently worked.

OVERALL:

I THINK WE ARE CONVERGING ON A MUTURALLY ACCEPTABLE MS, AND WE HAVE NOW DOUBT THAT THIS IS IMPORTANT AND EXCITING WORK. WE'RE LOOKING FORWARD TO SEEING IT PUBLISHED.

TO REITERATE AND SUM OUR LAST REQUESTS:

1) IN THE ABSTRACT, AND IN ALL SUB-TITLES, PLEASE USE THE PHRASE 'CAPTURE AND INTENSIVE TAMING' OR 'CAPTURE AND BREAKING' TO DENOTE WHAT IS HAPPENING TO WILD-BORN ANIMALS IN MTE. THAT WILL KEEP THINGS CLEAR AND UNAMBIGUOUS.

Ok, added to abstract (line 20-23), and to the sub-headings when relevant (line 178). Also in many points of the ms we now say "capture and taming" (or use similar terms), see lines 98, 208-209, 282-284, 296, 363-364, 377, 442 for example and also Figure 2 & 4 legends. In lines 138-139 we also specifically say "Thus part of the effects between captive-born and wild-caught elephants can be due to differences during taming period".

2) IN THE DISCUSSION, MAKE IT CLEAR THAT BEING CAUGHT FROM THE WILD AND BROKEN IS *NOT THE ONLY RISK FACTOR WITH POTENTIAL RELEVANCE FOR OTHER SPECIES*: SINCE BEING ZOO-BORN HAS ALSO BEEN SHOWN TO PREDISPOSE ELEPHANTS TO PREMATURE DEATHS, THE RICH DATASETS AVAILABLE FOR DIVERSE ELEPHANTS *TOGETHER* SHOW THAT EARLY EXPERIENCE CAN HAVE PROFOUND AND UNPREDICTABLE EFFECTS ON THE HEALTH OF WILD ANIMALS KEPT IN CAPTIVITY. PERHAPS CAPTURE IS SOMETIMES THE PROBLEM, PERHAPS CAPTURE AND TAMING/BREAKING; BUT IT'S CLEAR TOO THAT SOMETIMES OTHER FACTORS MAKE THE F1 CAPTIVE GENERATION INSTEAD EVEN MORE VULNERABLE! FUTURE RESEARCHERS SHOULD BE MINDFUL OF ALL OF THESE IMPORTANT AND INTRIGUING POSSIBLE EFFECTS.

Thank you for this suggestion, we have now added your suggested sentence on lines 394-396 ("Rich datasets available for diverse elephants together show that early experience can have profound and sometimes unpredictable effects of wild animals kept in captivity.") We also discuss in the same paragraph about zoo elephant performance in general and compared to our findings and what should be further studied in these populations, lines 391-394. See also lines 273-276 where we specify what kind of studies are also needed in elephants. We totally agree with the Reviewer that there are different risk

factors for elephant survival which should be studied further in different elephant populations.